

**Optimization of Biological Production for Indian Ocean upwelling zones: Part – I:**
**Improving Biological Parameterization via a variable Compensation Depth**
Mohanan Geethalekshmi Sreeush[1,2,*].
Vinu Valsala[1],
Sreenivas Pentakota[1],
Koneru Venkata Siva Rama Prasad[2],
Raghu Murtugudde[3]
[1]Indian Institute of Tropical Meteorology, Pune, India
[2]Department of Meteorology and Physical Oceanography,  Andhra University, India
[3]ESSIC, University of Maryland, USA
*Corresponding author address:
Indian Institute of Tropical Meteorology,
Dr. Homibhabha Road, Pashan, Pune 411 008, India
E-Mail: sreeushmg@tropmet.res.in



**Abstract**
Biological modeling approach adopted by the Ocean Carbon Cycle Model Inter-comparison
Project (OCMIP-II) provided amazingly simple but surprisingly accurate rendition of the annual
mean carbon cycle for the global ocean. Nonetheless, OCMIP models are known to have
seasonal biases which are typically attributed to their bulk parameterization of 'compensation
depth'. Utilizing the principle of minimum solar radiation for the production and its attenuation
by the surface Chl-a, we have proposed a new parameterization for a spatially and temporally
varying 'compensation depth' which captures the seasonality in the production zone reasonably
well. This new parameterization is shown to improve the seasonality of $CO_2$ fluxes, surface
ocean $pCO_2$, biological export and new production in the major upwelling zones of the Indian
Ocean. The seasonally varying compensation depth enriches the nutrient concentration in the
upper ocean yielding more faithful biological exports which in turn leads to an accurate
seasonality in carbon cycle. The export production strengthens by ~70% over western Arabian
sea during monsoon period and achieved a good balance between export and new production in
the model. This underscores the importance of having a seasonal balance in model export and
new production for a better representation of the seasonality of carbon cycle over upwelling
regions The study also implies that both the biological and solubility pumps play an important
role in the Indian Ocean upwelling zones.
Keywords: Indian Ocean upwelling zones, Carbon cycle, Seasonal cycle - $CO_2$ flux and Oceanic
$pCO_2$, Biogeochemical model parameterization, Export production - New production balance.



## 1. Introduction

Among the world's oceans, Indian Ocean is characterized by a unique seasonally reversing
wind systems called monsoon winds. The monsoon winds are the major physical drivers for the
coastal and open ocean upwelling in the Indian Ocean. The major upwelling systems in the
Indian Ocean are (1) the western Arabian Sea (WAS; Ryther et. al., 1965, Smith et. al., 2001,
Sarma., 2004, Wiggert et. al., 2006, Murtugudde et. al., 2007, McCreary et. al., 2009, Prasanna
Kumar et. al., 2010, Naqvi et. al., 2010, Roxy et al., 2015) (2) the SriLanka Dome
(SLD;Vinayachandran et al., 1998, 2004), (3) Java and Sumatra coasts (SC; Murtugudde et al.,
1999, Susanto et. al., 2001, Osawa et. al., 2010, Xing et. al., 2012) and (4) the Seychelles-
Chagos thermocline ridge (SCTR; Dilmahamod., 2016, Figure 1). The physical and biological
processes and their variability over these key regions are inseparably tied to the strength of the
monsoon winds and associated nutrient dynamics. The production and variability in the coastal
upwelling systems are a key concern to the fishing community, since they affect the day-to-day
livelihood of the coastal populations (Harvell et. al., 1999, Roxy et. al., 2015, Praveen et al.,
2016). Coastal upwelling systems account for about 11% of the world oceanic biological
production (Prasanna Kumar et al., 2001, Wiggert et al., 2005, Levy et al., 2007, McCreary et al.,
2009, Liao et. al., 2016) and are especially important for the Indian Ocean rim countries due to
their developing countries status.
Arabian Sea is a highly productive coastal upwelling system characterized by phytoplankton
blooms both in summer (Prasanna Kumar et al., 2001, Naqvi et al., 2003, Wiggert et al., 2005)
and winter (Banse K. et. al., 1986, Schubert et. al., 1998, Wiggert et. al., 2000, Barber et. al.,
2001, Prasannakumar et al., 2001, Sarma., 2004). Arabian Sea is known for the second largest
Tuna fishing region among all oceans (Lee et al., 2005). The Somali and Omani upwelling





regions experience phytoplankton blooms that are prominent with Net Primary Production (NPP) exceeding 438.29 g C m$^{-2}$ yr$^{-1}$ (Liao et. al., 2016). On the other hand productivity over the SLD (Vinayachandran and Yamagata, 1998), in the sea of Sri Lanka is triggered by an open ocean Ekman suction. SLD shows strong Chl-a blooms during the summer monsoon (Murtugudde et al., 1999, Vinayachandran et al., 2004). Compared to the Arabian Sea this bloom lasts for more than four months due to the impact of biogeochemistry of the region (Vinayachandran et. al., 2004). During the winter monsoon, the southwest Bay of Bengal is also characterized by Chl-a blooms associated with the intense cyclonic activities (Vinayachandran and Mathew, 2003).

The SC upwelling is basically due to the stronger alongshore winds and its variation is associated with impact of equatorial and coastal Kelvin waves (Murtugudde et. al., 2000). The interannual variability associated with Java-Sumatra coastal upwelling is strongly coupled with ENSO (El-Nino Southern Oscillation) through Indonesian throughflow (Susanto. et. al., 2001, Valsala et al., 2011) and peaks in July through August with a potential new production of 1.02 x 10$^{14}$ g C yr$^{-1}$ (Xing et. al., 2012).

The SCTR region productivity has large spatial and interannual variability. The warmer upper ocean condition associated with El Nino reduce the amplitude of the subseasonal SST variability over the SCTR (Jung and Kirtman., 2016). The Chl-a concentration peaks in summer when the southeast trade winds induce mixing and initiate the upwelling of nutrient-rich water (Murtugudde. et. al., 1999, Wiggert et al., 2006, Vialard. J. et. al., 2009, Dilmahamod et al., 2016).

Understanding the biological production and variability in the upwelling systems are important because it gives us crucial information regarding marine species variability (Colwell,




1996, Harvell et al., 1999, Selina et al., 2015). The observations also provide vital insights into
physical and biological interactions of the ecosystem (Naqvi et al., 2010) although limitations of
sparse observations often force us to depend on models to examine the large spatio-temporal
variability of the ecosystem (Valsala et al., 2013). Simple to inter-mediate complex marine
ecosystem models have been employed by several of the previous studies (Sarmiento et. al.,
2000, Orr. et. al., 2001, Matsumoto et. al., 2008). However the representation of marine
ecosystem variability by proper parameterizations in models has always been a daunting task.
This is impediment to the accurate representation of biological primary and export productions in
models (Friedrichs et al., 2006, 2007) and the parameterization issues also impact the modeling
of upper trophics levels (Lehodey et al., 2010).
Biological production can be quantified with a better understanding of primary
production by phytoplankton. Primary production depends on water temperature, light and
nutrient availability (Brock. et. al., 1993, Moisan. et. al., 2002) and this became the key reason for
parameterizing the production in models as one or more combinations of these terms (Yamanaka
et. al., 2004). Any of these parameters can be tweaked to alter production in models. For
example the availability of nutrients and light determines the phytoplankton growth (Eppely et
al., 1972) or growth rate (Boyd et. al., 2013). Stoichiometry and carbon-to-Chl-a ratios are other
important factors to be considered in modeling (Christian et al., 2001, Wang et al., 2009) but we
will not consider them in this study.
In 1995, an initiative by the IGBP/GAIM (International Geosphere-Biosphere
program/Global Analysis, Integration and Modeling) and IGBP/JGOFS (Joint Global Ocean
Flux Study) to study carbon cycle referred to as the Ocean Carbon cycle Model Intercomparison
Project (OCMIP) greatly improved our understanding of global carbon cycle (Raymond Najjar



and James Orr, 1998). OCMIP-II further introduced a simple phosphate dependent production
term in biological models for long term simulations of carbon cycle (Najjar et al, 1998).
Although OCMIP – II is a very simplified model, it is surprisingly accurate in simulating the
annual mean state and the response to anthropogenic climate change (Orr et al., 2001, Doney et.
al., 2004). However, the OCMIP – II model simulations comes with a penalty of higher seasonal
biases when compared with observations (Orr et al., 2003). In this protocol the light limitation is
formulated as a bulk quantity with the notion that the minimum light irradiance at which
phytoplankton photosynthesis is sufficient to balance the community respiration, Ic, is the
compensation irradiance (Sarmiento et al., 2006) and the depth at which the photosynthesis
equals respiration is the compensation depth $Z_c$ (Smetacek and Passow, 1990), which is clearly
different from the conventional euphotic zone depth (Morel., 1988). If the irradiance is below Ic,
phytoplankton growth will be suppressed. If the irradiance is above this, the planktonic
photosynthesis will exceed the community respiration and production will increase (Parsons et.
al., 1984, Sarmiento et. al., 2006). Therefore the compensation depth represents the oceanic
production zone in this approach.

However, $Z_c$ was held constant in time and space in OCMIP-II models (Raymond Najjar

and James Orr, 1998, Matsumoto et. al., 2008) though in reality $Z_c$ varies in space and time
(Najjar and Keeling, 1997) just as the euphotic zone depth does as documented in ship
measurements (Qasim, 1977, 1982). The variation in compensation depth indicates the
seasonality of the production zone itself. Availability of light and nutrients at an optimum level
is clearly essential for primary production.
Most of the biophysical models prescribe a constant value for compensation depth (e.g., $Z_c$ =
75m in OCMIP –II protocol (Raymond Najjar and James Orr, 1998), $Z_c$ = 100m for Minnestoa



Earth System Model (Matsumoto et. al., 2008) although in reality it is not a constant. Depending on the latitude, compensation depth varies between 50m and 100m in the real world (Najjar and Keeling, 1997). In our study we have attempted a novel biological parameterization scheme for spatially and temporally varying compensation depth in the OCMIP – II framework by representing it as a function of optimum solar radiation (Parsons et. al., 1984) and Chl-a availability. In this hypothesis, the minimum solar radiation required for photosynthesis is taken as 10 W m$^{-2}$ below which the production reduces to 20% (Parsons et. al., 1984) and the Chl-a concentration which determines the attenuation of solar radiation with depth in the production zone is also assumed to vary to yield the spatio-temporal variability of Zc. The basic currency of phosphate will act as limiting factor for biological production within this varying compensation depth. This spatially and temporally varying compensation depth represents the seasonality in the production zone.

Regions of sustained upwelling like the eastern equatorial Pacific are well understood in terms of the role of upwelling in increasing the surface water $pCO_2$ to drive an outgassing of $CO_2$ into the atmosphere (Feely et al., 1999, Valsala et al., 2014). The Indian Ocean on the other hand only experiences seasonal upwelling which is relatively weak in the deep tropics but stronger off the coasts of Somalia and Oman and in the SLD region (Valsala et al., 2013). The relative importance of the solubility vs. biological pump is not well understood. Our focus here on implementing seasonality in the compensation depth of OCMIP models nonetheless leads to new insights on the impact of improved biological production on surface water $pCO_2$ and air-sea $CO_2$ fluxes. The largely positive effects of the variable compensation depth over the Indian Ocean and the sensitivity experiments where upwelling is muted strongly imply that the biological pump may play as much of role as the solubility pump in determining surface $pCO_2$ and $CO_2$ fluxes.



The  paper is organized as follows. Model, Data and Methodology are detailed in Section
2. The spatially inhomogeneous Zc derived out of the new parameterization and its impact in
simulated seasonality of biology and carbon cycle are detailed in Section 3. Further results and
discussion are followed in Section 4 and a conclusion is given in Section 5.
**2. Model, Data and Methods**
**2.1. Model**
The study utilizes the Offline Ocean Tracer Transport Model (OTTM) (Valsala et al., 2008)
coupled with OCMIP biogeochemistry model (Raymond Najjar and James Orr, 1998). OTTM
does not compute currents and stratifications (i.e., temperature and salinity) on its own. It is
capable of accepting any ocean model or data-assimilated product as physical drivers. The
physical drivers prescribed include 4-dimensional currents (u,v), temperature, salinity, and 3-
dimensional mixed layer depth,  surface freshwater and heat fluxes, surface wind stress and sea
surface height. The resolution of the model setup is similar to the parent model from which it
borrows the physical drivers. With the given input of Geophysical Fluid Dynamics Laboratory
(GFDL) reanalysis data, the zonal and meridional resolutions are $1^{o}$ with 360 grid points
longitudinally and $1^{o}$ at higher latitudes but having a finer resolution of $0.8^{o}$ in the tropics, with
200 grid points, respectively. The model has 50 vertical levels with 10m increment in the upper
225m and stretched vertical levels below 225m. The horizontal grids are formulated in spherical
co-ordinates and vertical grids are in z levels. The model employs a B-grid structure in which the
velocities are resolved at corners of the tracer grids. The model uses a centered-in-space and
centered-in- time (CSCT) numerical scheme along with an Asselin-Robert filter (Asselin., 1972)
to control the ripples in CSCT.


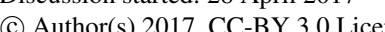


The tracer concentration (C) evolves with time as
$\frac{\partial C}{\partial t} + U.\nabla_H C + W\frac{\partial C}{\partial z} = \frac{\partial}{\partial z}K_z\frac{\partial}{\partial z}C + \nabla_H.(K_h\nabla_H C) + \Phi$        (1)
where $\nabla_H$ is the horizontal gradient operator, U and W are the horizontal and vertical velocities
respectively. $K_z$ is the vertical mixing coefficient, and $K_h$ is the two-dimensional diffusion
tensor.  $\Phi$ represents any sink or source due to the internal consumption or production of the
tracer as well as the emission or absorption of fluxes at the ocean surface. Here, the source and
sink term are provided through the biogeochemical model. Vertical mixing is resolved in the
model using K- profile parameterization (KPP) (Large et al., 1994).
In addition to KPP, the model uses a background vertical diffusion reported by Bryan and Lewis
(Bryan and Lewis., 1979) in order to represent the convention and mixing that happens in a time
scale of a few days. For horizontal mixing model incorporates Redi fluxes (Redi., 1982) and GM
fluxes (Gent and Mcwilliams., 1990) which accommodate the eddy induced variance from the
mean in the tracer transport. A weak Laplacian diffusion is also included in the model for
computational stability where sharp gradient of concentration occurs.

195       The biogeochemical model used in the study is based on the OCMIP – II protocol as

stated above. The main motivation of OCMIP–II model design is to simulate the ocean carbon
cycle with reductionist approach to ocean biology using appropriate biogeochemical
parameterizations. The major advantage of the OCMIP – II protocol is (i) it reproduces the first
order carbon cycle and the associated elemental cycles in the ocean reasonably well and (ii) it is
much easier to implement and computationally efficient than the explicit ecosystem models.  The
present version of the model has five prognostic variables coupled with the circulation field, viz.,
inorganic phosphate ($PO_4^{3-}$), dissolved organic phosphorous (DOP), oxygen ($O_2$), dissolved



inorganic carbon (DIC) and alkalinity (ALK). In order to retrieve the accurate spatial and
temporal distribution of $CO_2$ flux and $pCO_2$, the model uses a "nutrient restoring" approach
(Najjar et. al., 1992, Anderson and Sarmiento., 1995) for biological production. The basic
currency for biological production in the model is phosphate because of the availability of a
more extensive database and to eliminate the complexities associated with nitrogen fixation and
denitrification. The biogeochemical dynamics implemented in the model are given Appendix-A
The air – sea $CO_2$ flux in the model is estimated by,
$F = K_w \, \Delta \, pCO_2$     (2)
where $K_w$ is gas transfer velocity and $\Delta \, pCO_2$ is the difference in partial pressure of carbon
dioxide between the ocean and atmosphere. The design and validation of the physical model is
reported by Valsala et. al.,(2008, 2010) and biogeochemical design by Najjar and Orr (1998).
**2.2. Data**

The present setup of the model uses ocean reanalysis products based on MOM–4

(Modular Ocean Model) developed by GFDL (Chang et. al., 2012). Monthly data from 1961 to
2010 were utilized in the present study. For validating the results observational datasets of $CO_2$
flux and $pCO_2$ were taken from Takahashi et al., (2009). Satellite derived Net Primary
Production data were taken from Sea-viewing Wide Field of view sensor (SeaWiFS) Chl-a
product, calculated using Vertically Generalized Production Model (VGPM) (Behrenfeld and
Falkowski, 1997). The initial conditions for $PO_4$ and $O_2$ were taken from World Ocean Atlas
(Conkright et al., 1994). Initial conditions for DIC and ALK were taken from the Global Ocean
Data Analysis Project (GLODAP) dataset. The data sources and citations are given in the
Acknowledgement.





A spin-up for 50 years from the given initial conditions are performed with the
climatological physical drivers. Because the initial conditions were provided from a mean state
observed climatology this duration of spin-up is sufficient to reach statistical equilibrium in the
upper 1000 m (Le Quere et al., 2000). Atmospheric $pCO_2$ has been set to a value from the 1950s
in the spin-up run for calculating the air-sea $CO_2$ exchange. A seasonal cycle of atmospheric
$pCO_2$ has been prescribed.
After the spin-up, an interannual simulation for 50 years from 1961 to 2010 has been
carried out with the corresponding observed atmospheric $pCO_2$ described in Keeling et al,
(1995). The first five years of the interannual run were looped five times through the physical
fields of 1961 repeatedly for a smooth merging of the spin-up restart to the interannual physical
variables. Since the study is focused only on bias corrections to seasonal cycle with a variable
$Zc$, a model climatology has been constructed from 1990 to 2010. This includes the
anthropogenic increase of oceanic DIC in the climatological calculation and is comparable with
the Takahashi et al. (2009) observations.
Additional two sensitivity experiments have been performed separately by providing
annual mean currents or temperatures as drivers over selected regions of the basin for
segregating the role of varying compensation depth (varZc) in improving the seasonality of
carbon cycle and biological production. The model driven with annual mean currents suppress
the effect of upwelling by muting the ekman divergence over the region of interest. On the other
hand, the model forced with annual mean temperatures suppresses the cooling effect of
upwelling. This will highlight the effect of new parameterization in simulating seasonality of
carbon cycle and biological production. A smoothing technique with linear interpolation
$(u = u(1-x) + \bar{u}x)$ is applied to the offline-data in order blend the annual mean fields $(\bar{u})$



given to the selected region with the rest of the domain (u) in order to reduce a sudden transition
at the boundaries.

**2.3. Compensation depth ($Z_c$) parameterization**


The OCMIP – II simulation protocol separates the production and consumption zones by
a depth termed as compensation depth ($Z_c$); the depth at which photosynthesis is equal to
respiration of the photosynthetic community (Smetacek and Passow., 1990). The light intensity
at compensation depth is compensation irradiance ($I_c$) with larger values at higher temperatures
since respiration is temperature dependent (Parsons et. al., 1984, Ryther, 2003). We define a
spatially and temporally varying compensation depth (hereinafter varZc) as a depth where solar
radiation (attenuated by surface Chl-a, Jerlov et al., 1976) reaches a minimum value of 10 W m$^{-2}$.
In this way the varZc has both spatio-temporal variability of light as well as Chl-a data. The Chl-
a is given as monthly climatology as constructed from the satellite data. Observations show that
the primary production reduces rapidly to 20% or less of the surface value below a threshold of
10 W m$^{-2}$ (Parsons et.al., 1984, Ryther, 2003). Figure 2 compares the scatter of average relative
photosynthesis within varZc as a function of  solar radiation for the Indian Ocean. This
encapsulate the corresponding curve from the observations for the major phytoplankton species
in the ocean such as diatoms, green algae and dinoflagellates (Ryther et al., 1956, 2003, Parsons
et al., 1984). The model permits 100% relative photosynthesis for radiation above 50 W m$^{-2}$.
However the availability of phosphate concentration in the model act as an additional  limiter for
production which indirectly represents the photoinhibition at higher irradiance, for example
oligotrophic gyres.



3. **Results and Discussions**

The inclusion of seasonality in Zc by way of parameterizing varZc leads to a remarkable

spatio-temporal variability in compensation depth (Figure 3). The compensation depth over the
Arabian Sea varies from 10m to 25m during DJF and deepens up to 45m during MAM in par
with incoming solar radiation. During the monsoon season (JJAS), the compensation depth again
shoals to 10m-35m due to the attenuation of solar radiation by the increased biological
production (Chl-a). During Oct–Nov the Zc slightly deepens as compared to JJAS.
The Bay of Bengal compensation depth deepens from 35m to 40m during DJF and further
deepens to 50m during MAM when the solar radiation is maximum and biological production is
minimum (Prasannakumar et al., 2002). Further reduction of compensation depth can be seen
through JJAS as a result of reduction in solar radiation during monsoon cloud cover. The Zc
during Oct – Nov is on an average of 35m. However, caution is needed since the Bay of Bengal
is dominated by freshwater forcing from rivers and precipitation and temperature inversions
occur routinely (Howden and Murtugudde., 2001, Vinayachandran et. al., 2013). The impact of
these factors on compensation depth variability is not clear and is not addressed here.
The equatorial Indian Ocean can be seen as a belt of 40m-45m deep compensation depth
throughout the season except for JJAS. During JJAS, a shallow compensation depth is seen near
the coastal Arabian Sea (around 10m to 35m) presumably due to the coastal Chl-a blooms. Deep
compensation depth off the coast of Sumatra  (~ 40m to 50m) is found during JJAS. Java-
Sumatra coastal upwelling is centered on September-Nov (Susanto et al., 2001) and upwelling
originates at around 100m deep (Xing et al., 2012).



Southward of 10$^o$S in the oligotrophic gyre region, the compensation depth varies from 40m to
more than 60m throughout the year. A conspicuous feature observed while parameterizing the
solar radiation and Chl-a dependent Zc is that its maximum value never crosses 75 m especially
in the Indian Ocean which is the value specified in OCMIP-II models. The cutoff depth of 75 m
in OCMIP-II is obtained from observing the seasonal variance in the oxygen data (Najjar and
Keeling, 1997) as an indicator of production zone. However, parameterizing a production zone
based on optimum solar radiation and Chl-a (Parsons et al., 1984) predicts a production zone
and its variability that is largely less than 75 m. The consequence of this in the seasonality of the
modeled carbon cycle is illustrated as follows.
**3.1. Simulated seasonal cycle of pCO$_2$ and CO$_2$ fluxes**
The annual mean biases in simulated CO$_2$ fluxes and pCO$_2$ were evaluated by comparing
with Takahashi et al., (2009) observations (Figure 4). The model biases are significantly reduced
with the implementation of varying Z$_c$ compared to that of the constant Z$_c$. A notable reduction in
pCO$_2$ bias (by ~ 10μatm) is observed along the WAS region [Figure (4d)].
In order to address the role of the new biological parameterization of a variable
compensation depth, we extended our study by choosing four key regions where the biological
production and CO$_2$ fluxes are prominent in the Indian Ocean with additional sensitivity
experiments (see Introduction and references therein). The boxes we considered are, (1) Western
Arabian Sea (WAS) [40$^o$E:65$^o$E, 5$^o$S:25$^o$N] (2) Sri Lanka Dome (SLD) [81$^o$E:90$^o$E, 0$^o$:10$^o$N] (3)
Seychelles-Chagos Thermocline Ridge (SCTR) [50$^o$E:80$^o$E, 5$^o$S:10$^o$S] (4) Sumatra Coast (SC)
[90$^o$E:110$^o$E, 0$^o$:10$^o$S; Figure 1]. The seasonal variations of Zc over these selected key regions





are shown in Figure 5. A detailed analysis of $CO_2$ fluxes, $pCO_2$, biological export and new
production for these key regions are presented below.

**3.2. Western Arabian Sea (WAS) region**
The WAS $Z_c$ has a double peak pattern over the annual cycle. Over the February-March
period $Z_c$ deepens up to a maximum of $43.85 \pm 2.3$ m into March and then shoals to $25.75 \pm 1.5$
m (Fig 5) during the monsoon period (uncertainty represents the interannual standard deviations
of monthly data from 1990-2010). This shoaling of compensation depth during the monsoon
indicates the potential ability of the present biological parameterization to capture the wind
driven upwelling related production in the WAS. During the post monsoon period, the second
deepening of compensation depth occurs during November with a maximum depth of $34.91 \pm$
2.2 m. The ability to represent the seasonality of biological production zone renders a unique
improvement in $CO_2$ flux variability especially in the WAS region in comparison to the OCMIP-
II experiments (Orr et al, 2003, Figure (6a)).
OCMIP –II simulations with a constant $Z_c$ of 75 m underestimate the $CO_2$ flux when
compared to the observations of Takahashi et al. (2009). This underestimation is clearly visible
during monsoon period. Our simulations with the present biological parameterization having a
spatially and temporally varying compensation depth results in a better seasonality of $CO_2$ flux
when compared with Takahashi et al. (2009) observations (Figure 6a). The improvement brought
about the varying Zc scheme is able to represent the seasonality of $CO_2$ flux especially during the
monsoon period, when wind driven upwelling is dominant. Obviously the role of the biological
and solubility pumps have to be deciphered in this context.

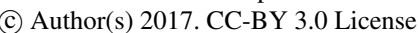


The $CO_2$ flux during July from observations, constZc simulations and varZc simulations are
3.09 mol m$^{-2}$ yr$^{-1}$, 1.82 ± 0.4 mol m$^{-2}$ yr$^{-1}$ and 3.10 ± 0.5 mol m$^{-2}$ yr$^{-1}$, respectively. Southwesterly
wind-driven upwelling over the WAS especially off Somali coast (Smith & Codispoti, 1980,
Schott, 1983, Smith, 1984) and Oman (Bruce, 1974, Smith &Bottero, 1977, Swallow, 1984,
Bauer et al, 1991), pulls nutrient-rich subsurface waters closer to the surface while the available
turbulent energy due to the strong winds leads to mixed layer entrainment of the nutrients
resulting in a strong surface phytoplankton bloom (Krey & Babenerd, 1976, Banse, 1987, Bauer,
1991, Brock et al, 1991). This regional bloom extends over 700 km offshore from the Omani
coast due to upward Ekman pumping driven by strong, positive wind-stress curl to the northwest
of the low level jet axis and the offshore advection (Bauer et. al., 1991, Brock et al., 1991, Brock
& McClain, 1992a, b, Murtugudde and Busalacchi., 1999) resulting in strong outgassing of $CO_2$
flux and an enhanced $pCO_2$ in the western Arabian Sea region (Valsala and Maksyutov, 2013,
Sarma et al., 2002). The seasonal mean $CO_2$ flux during the southwest monsoon period (JJAS)
for constZc simulations and varZc simulations are 1.44 ± 0.2 mol m$^{-2}$ yr$^{-1}$ and 2.31 ± 0.4 mol m$^{-2}$
yr$^{-1}$, respectively. The biological parameterization of varying compensation depth considerably
improves the average $CO_2$ flux during the monsoon period by 0.86 ± 0.1 mol m$^{-2}$ yr$^{-1}$. The annual
mean $CO_2$ flux from observations, constZc simulations and varZc simulations are 0.94 mol m$^{-2}$
yr$^{-1}$, 0.80 ± 0.17 mol m$^{-2}$ yr$^{-1}$ and 1.07 ± 0.2 mol m$^{-2}$ yr$^{-1}$, respectively. The annual mean $CO_2$ flux
improved by 0.27 ± 0.05 mol m$^{-2}$ yr$^{-1}$.
Seasonality in $pCO_2$ also shows a remarkable improvement during the southwest monsoon
period [Figure (6b)]. The $pCO_2$ with ConstZc is considerably lower at a value of 385.22 ± 3.5
µatm during June compared to observational values of 392.83 µatm. However, varZc simulations
perform better in terms of  $pCO_2$ variability. The peak value of $pCO_2$ reaches up to 405.42 ± 5.8



µatm. The seasonal mean $pCO_2$ during the southwest monsoon period from observations, constZc simulations and varZc simulations are 397.58 µatm, 389.18 ± 3.6 µatm and 399.95 ± 5.0 µatm, respectively. The improvement in $pCO_2$ brought about by varZc simulations is 10.76 ± 1.3 µatm compared to the constZc simulations. This inherently says that constZc simulations fail to capture the $pCO_2$ driven by upwelling during southwest monsoon, meanwhile varZc simulations are demonstrably better in representing this seasonal increase. The annual mean $pCO_2$ from observations, constZc and varZc simulations are 394.69 µatm, 389.62 ± 3.9 µatm and 391.19 ± 4.7 µatm, respectively. However it is worth mentioning that there are parts of the year where the constant Zc performs better compared to varying Zc. For instance during MAM as well as in November, the constZc simulations yielded a better comparison with the observed $pCO_2$ whereas varZc simulations yield a reduced magnitude of $pCO_2$. This may well indicate the biological vs solubility pump controls on $pCO_2$ during the intermonsoons. The role of mesoscale variability in the ocean dynamics may also play a role (Valsala and Murtugudde, 2015) Nevertheless during the most important season (JJAS) when the $pCO_2$, $CO_2$ fluxes and biological production are found to be dominant in the Arabian Sea, the varying Zc produces a better simulation.

The improvement shown by the implementation of new biological parameterization in the simulations of $CO_2$ flux and $pCO_2$ can be answered by further analysis of the model biological production. Figure 7 shows the comparison of model export production and new production with observational export production from satellite-derived NPP for constZc and varZc simulations. The model export production in the constZc simulations is much weaker when compared to varZc simulations. The varZc simulations have improved the model export production. Theoretically, the new and export productions in the model should be in balance with each other



(Eppley and Peterson, 1979). ConstZc export production is much weaker than new production
and it is not in balance. In contrast the varZc simulation yields a nice balance among them.
Comparing with the observational export production which peaks in August at a value of
154.78 g C m$^{-2}$ yr$^{-1}$, the varZc simulated export and new productions peak at a value of 160.44 ±
20.4 g C m$^{-2}$ yr$^{-1}$ and 167.18 ± 24.0 g C m$^{-2}$ yr$^{-1}$, respectively but in July. A similar peak can be
observed in constZc simulated new production as well at a value of 178.19 ± 28.0 g C m$^{-2}$ yr$^{-1}$.
This apparent shift of one month during JJAS in the model export production as well as in the
new production is noted as a caveat in the present set up which will need further investigation.
Arabian Sea production is not just limited by nutrients but also the dust inputs (Wiggert and
Murtugudde., 2006). The dust induced primary production in the WAS especially over the Oman
coast is noted during August (Liao et. al., 2016). The mesoscale variability in the circulation and
its impact on production and carbon cycle are also a limiting factor in this model as noted above.
The seasonal mean export production during the southwest monsoon period from satellite-
derived estimate is 123.57 g C m$^{-2}$ yr$^{-1}$, whereas for constZc and varZc simulations it is 84.81 ±
16.0 g C m$^{-2}$ yr$^{-1}$ and 147.19 ± 23.8 g C m$^{-2}$ yr$^{-1}$, respectively. The new biological
parameterization strengthened the model export production by 62.38 ± 7.8 g C m$^{-2}$ yr$^{-1}$ for the
southwest monsoon period, which is over a 70% increase. This indicates a considerable impact
of the biological pump in the model simulated $CO_2$ flux and $pCO_2$ over the WAS. For constZc
simulations, the computed new production is slightly higher (150.84 ± 27.9 g C m$^{-2}$ yr$^{-1}$) than that
of varZc (133.03 ± 19.5 g C m$^{-2}$ yr$^{-1}$). The  annual mean export production from observations,
constZc and varZc simulations are 94.31 g C m$^{-2}$ yr$^{-1}$, 77.41 ± 15.1 g C m$^{-2}$ yr$^{-1}$ and 122.54 ± 25.2
g C m$^{-2}$ yr$^{-1}$, respectively.



To understand how the varying compensation depth parameterization strengthened the export
production in the model, we analyzed the phosphate profiles. It appears that the varZc
parameterization allows more phosphate concentration (Figure 8a,b) in the production zone and
thereby increases the corresponding biological production (Figure 8c, d). The net export
production in the model  during JJAS is consistent with the satellite data (Figure 8d, see also
Figure 7a). However, in the constZc case the exports are rather 'flat' throughout the season with
imperfect representation of seasonal biological export. The Table 1-4 summarizes all the values
discussed here.

**3.3  SriLanka Dome (SLD) Region**

The seasonal variation in the compensation depth for the SLD has a similar pattern as that of
the WAS. The compensation depth deepens to its maximum during March up to $45.23 \pm 0.3$ m
and reaches its minimum during the following monsoon period at $30.79 \pm 1.5$ m [Figure (5)].
The similarities of varZc between WAS and SLD indicate that they both are under similar cycles
of solar influx and biological production. The SLD chl-a dominates only up to July
(Vinayachandran et al., 2004) which explains why production with varZc increase  earlier
compared to the WAS which occurs during August-October.
The seasonality in $CO_2$ flux and $pCO_2$ were compared with Takahashi et al., (2009)
observations [Figure (9)]. The varZc results in a slight improvement in $CO_2$ flux when compared
with constZc [Figure (9a)]. However both constZc and varZc simulations underestimate the
magnitude of $CO_2$ flux when compared with observations. The seasonal mean $CO_2$ flux during




the monsoon period is 1.79 mol m$^{-2}$ yr$^{-1}$ from observations, which means SLD region is a source
of $CO_2$. But the mean values of constZc and varZc simulations yield flux values of -0.008 ± 0.2
mol m$^{-2}$ yr$^{-1}$ and 0.24 ± 0.2 mol m$^{-2}$ yr$^{-1}$, respectively. The constZc simulations misrepresent the
SLD region as a sink of $CO_2$ during monsoon period which is opposite to that of observations.
The varZc simulations correct this misrepresentation to a source albeit at a smaller magnitude by
0.24 ± 0.09 mol m$^{-2}$ yr$^{-1}$ for the monsoon period. Compared to observations, the varZc case
underestimates the magnitude of JJAS mean by 1.55 mol m$^{-2}$ yr$^{-1}$.
The annual mean $CO_2$ fluxes for constZc and varZc simulations are -0.02 ± 0.1 mol m$^{-2}$ yr$^{-1}$
and 0.10 ± 0.2 mol m$^{-2}$ yr$^{-1}$, respectively. The varZc parameterization leads to an improvement of
0.13 ± 0.1 mol m$^{-2}$ yr$^{-1}$ in the annual mean $CO_2$ flux when compared with constZc simulations.
The observational annual mean of $CO_2$ flux is 0.80 mol m$^{-2}$ yr$^{-1}$ which is highly underestimated
by both simulations. This indicates a regulation of biological production of the region by varZc
which makes this region a source of $CO_2$ during monsoon. The role of the solubility pump may
also be underestimated due to the biases in the physical drivers and the lack of mesoscale eddy
activities in these simulations (Prasanna Kumar et. al., 2002, Valsala and Murtugudde., 2015).
The seasonality of pCO$_2$ [Figure 9(b)] especially in the monsoon period has significantly
improved. The mean pCO$_2$ during the monsoon season from observation over the SLD region is
382.44 µatm. The seasonal mean pCO$_2$ during monsoon period for constZc and varZc
simulations are 371.67 ± 6.04 µatm and 379.24 ± 8.9 µatm, respectively. The annual mean pCO$_2$
from observations, constZc and varZc simulations are 380.21 µatm, 370.76 ± 6.1 µatm and
374.94 ± 9.6 µatm, respectively. varZc simulations improve the JJAS mean pCO$_2$ by 7.56 ± 2.8
µatm and the annual mean pCO$_2$ by 4.18 ± 3.5 µatm, which is reflected in $CO_2$ flux as well. This





is likely due to the impact of new biological parameterization in capturing the episodic upwelling
in the SLD region which is further investigated by looking at its biological production.

The SLD biological production is highly exaggerated by the model for both constZc and

varZc simulations (Figure 10a,b). The seasonal mean biological export for the monsoon period is
51.54 g C m$^{-2}$ yr$^{-1}$ as per satellite-derived estimates. However, the constZc  and varZc simulations
overestimate it by 167.71 ± 59.04 g C m$^{-2}$ yr$^{-1}$ and 151.51 ± 46.4 g C m$^{-2}$ yr$^{-1}$, respectively. This
exaggerated export is visible in climatological annual means where for constZc and varZc
simulations they are 144.43 ± 49.8 g C m$^{-2}$ yr$^{-1}$ and 156.08 ± 43.8 g C m$^{-2}$ yr$^{-1}$, respectively.

For constZc simulations, new production is overestimated from March to Oct when

compared to the observations and second peak is observed in November (Figure 10a). But the
overestimate in new production with varZc is observed only during JJAS period by a value of
26.23 g C m$^{-2}$ yr$^{-1}$. For the SLD region the varZc parameterization overestimates the export
production but minimizes the excess new production, especially in the monsoon period by 64.15
± 36.4 g C m$^{-2}$ yr$^{-1}$. This indicates that the varZc parameterization is somewhat successful in
capturing the upwelling episode during monsoon over SLD. All values are summarized in Table

1-4.

**3.4 Sumatra Coast (SC) region**

The seasonal variation in the compensation depth over the SC region lies between 40 m and

46 m [Figure (5)]. The seasonal maximum occurs during JFM, especially in March with a depth
of 45.5 m. During the monsoon period the compensation depth shoals slightly, with a minimum
of 41.1 m in July. The variation in $Z_c$ is relatively small as compared to the other regions which
is consistent with its relatively low production throughout the year.



The seasonality of $CO_2$ flux and $pCO_2$ captured by constZc and varZc simulations are shown
in Figure 11(a, b). The varZc simulations overestimate both $CO_2$ flux and $pCO_2$ especially during
the monsoon. It is found that the constZc simulations are better compared to varZc simulations.
The varZc simulations overestimate the seasonal mean $CO_2$ flux and $pCO_2$ by 1.19 mol m$^{-2}$ yr$^{-1}$
and 29.61 µatm, respectively, compared to observations (Table 1). However, constZc produces a
better estimate compared with observations for $CO_2$ flux and $pCO_2$. The constZc simulations
deliver a better annual mean than varZc (Table 1,2). The annual mean bias in constZc and varZc
simulations for $CO_2$ flux is -0.0033 mol m$^{-2}$ yr$^{-1}$ and 0.31 mol m$^{-2}$ yr$^{-1}$, respectively. Similarly
$pCO_2$ bias is 1.95 µatm and 9.07 µatm for constZc and varZc simulations.
Biological production simulated by the model in the SC explains the overestimation of $CO_2$
flux and $pCO_2$ (Figure 12). Both constZc and varZc simulations greatly overestimates export
production in the model. But a small enhancement in the new production during JJAS in constZc
case is an indicator of upwelling episodes. The seasonal mean new production during the
monsoon from constZc and varZc are 63.64 ± 30.9 g C m$^{-2}$ yr$^{-1}$ and 78.11 ± 29.1 g C m$^{-2}$ yr$^{-1}$,
respectively (Table 4). The seasonal mean export production during the monsoon from
observation is 58.87 g C m$^{-2}$ yr$^{-1}$ (Table 3). ConstZc simulations represent a better new
production, which is seen as a relatively small exaggeration of $CO_2$ flux and $pCO_2$. The
biological response off SC is found to be better with constZc which is in contradiction to a
general improvement found with varZc in the other regions examined here. Such discrepancies
over the SC could be due to the effect of Indonesian Throughflow (Bates et al., 2006) which is
not completely resolved in the model due to coarse spatial resolution.



**3.5 Seychelles-Chagos Thermocline Ridge (SCTR) region**

The SCTR is a unique upwelling region with a prominent variability in air-sea interactions (Xie et al., 2002). Wind-driven mixing and upwelling of subsurface nutrient rich water play a major role in biological production of this region (Dilmahamod et al., 2016). The seasonal cycle in the compensation depth is shown in figure 5. The maximum compensation depth occurs in November at about 44.94 m and the minimum at 33.2 m in July. The shoaling of compensation depth during the monsoon period shows that the biological parameterization captures the upwelling response over this region.

The seasonality of $CO_2$ flux and $pCO_2$ are shown in Figure 13. The Takahashi observations of $CO_2$ flux shows a peak in June with outgassing of $CO_2$ during the upwelling episodes. However, both constZc and varZc simulations underestimate this variability. The seasonality of $CO_2$ flux in varZc shows a significant improvement when compared to constZc simulations, but underestimated when compared to observations. The seasonal mean $CO_2$ flux during the monsoon for constZc and varZc simulations are 0.82 mol $m^{-2}$ $yr^{-1}$, -0.32 ± 0.3 mol $m^{-2}$ $yr^{-1}$ and -0.05 ± 0.4 mol $m^{-2}$ $yr^{-1}$, respectively. This represents a reduction in seasonal mean sink of $CO_2$ flux in the SCTR region during the monsoon by 0.27 ± 0.1 mol $m^{-2}$ $yr^{-1}$ bringing it closer to a source region (see Table 1 for details).

The improvement brought about in $CO_2$ flux is supported by the seasonal cycle in $pCO_2$ . Based on observations, seasonal mean of $pCO_2$ with constZc during JJAS is underestimated by 11.47 µatm, varZc simulations underestimate it by 6.45 µatm. So it is evident that varZc simulations capture the upwelling episodes better, marked by a greater $pCO_2$ during JJAS period. However, the magnitude of $pCO_2$ is still underestimated compared to observations (Table 2).



Figure 14 shows the biological production of constZc and varZc simulations for SCTR. It is
clear that both simulations overestimate the export production and underestimate the new
production. The JJAS mean export production from observations, constZc and varZc are 51.08 g
C m$^{-2}$ yr$^{-1}$, 57.39 ± 14.2 g C m$^{-2}$ yr$^{-1}$ and 99.23 ± 29.8 g C m$^{-2}$ yr$^{-1}$, respectively. The varZc
simulations exaggerate the model export production by 48.14 g C m$^{-2}$ yr$^{-1}$.The varZc simulations
improve the JJAS mean new production by 1.14 ± 2.2 g C m$^{-2}$ yr$^{-1}$ (Table 4). This slight
improvement in the model new production especially during the monsoon period signals that the
spatially and temporally varying compensation depth better captures the upwelling over SCTR.
Considering the annual mean values of model export and new production, constZc simulations
are reasonably faithful to observations.
The underestimation of $CO_2$ and $pCO_2$ as well as the exaggeration of model export
production and a slight overestimate in model new production may be due to two reasons. (1)
SCTR is a strongly coupled region with remote forcing of the mixed layer – thermocline
interactions (Zhou et al., 2008) which can affect the seasonality in biological production that the
model may not be resolving reasonably (2) The bias associated with physical drivers, especially
wind stress may underestimate the $CO_2$ flux as well biological production. A similar
overestimation of biological production was also reported in a coupled biophysical model
(Dilmahamod et. al., 2016).
Table 1 – 4 shows the entire summary of seasonal and annual mean $CO_2$ flux, $pCO_2$ and
biological production reported in Section 3.




**4. Sensitivity Simulations**

From the analysis of four major upwelling regions over Indian Ocean, it is evident that the biological parameterization of spatio-temporally varying compensation depth better captures upwelling episodes and thus it enhances the model export production. This is clearly visible over the WAS. In order to quantify how much the varZc parameterization contributed to seasonality of carbon cycle, two additional sensitivity simulations were carried out; (1) with annual mean offline currents and (2) annual mean offline temperatures with the notion of suppressing the dynamical and thermodynamical effects of seasonal upwelling over the WAS (see Section 2 for details). The focus on this region is motivated by its prominence as the most productive zone of the Indian Ocean. Moreover, the improvement in the biological processes in the model by the varZc parameterization is best captured in this region. The results are discussed in detail in the following subsections.

**4.1 Impact of varZc parameterization in seasonality of carbon cycle with annual mean currents.**

To quantify the impact of varZc parameterization, the model is forced with annual mean currents only over the WAS with unaltered currents in the rest of the ocean. The hypothesis is that the muting of the seasonal variability of Ekman divergence removes the upwelling and the biological pump contribution to production and carbon cycle. The comparison of constZc and varZc then allows us to decipher the impact of varZc on capturing the impacts of upwelling on biological production and the carbon cycle. The smooth blending of currents at the boundary of the WAS domain is achieved by a linear smoothing function as given in Section 2.



The model biological responses (inferred by comparing with the control run) in terms of the
$CO_2$ flux shows a flat pattern over the monsoon period for constZc simulations [Figure 15(a)].
While the varZc simulations forced with the annual mean currents shows an enhanced $CO_2$ flux
indicating the outgassing of $CO_2$ flux in the WAS due to wind-driven upwelling (Figure 15(b)).
This qualitatively shows that the spatially and temporally varying compensation depth itself has
improved the seasonality in the biological processes (export and new production) and captured
the upwelling episodes during the monsoon. The varZc parameterization is responsible for
improvement of $0.48 \pm 0.04$ mol m$^{-2}$ yr$^{-1}$ and $0.13 \pm 0.02$ mol m$^{-2}$ yr$^{-1}$during JJAS seasonal and
annual mean $CO_2$ flux, respectively. This improves the overall model $CO_2$ flux in the control run
especially in July (Figure 15(b)).
Similar improvements were also noticed in $pCO_2$ (Figure 16). In the constZc simulations
with annual mean currents, the $pCO_2$ dips down during JJAS monsoon period which indicates
the inadequacy of constZc in capturing the upwelling enriched $pCO_2$ difference (Figure 16a, b).
The varZc simulation slightly modifies the $pCO_2$ in the 'right' direction during JJAS despite the
annual mean currents.
The export production and the new production in the model explain the modification of
$CO_2$ flux and $pCO_2$ by varZc parameterization. The biological export production is highly
underestimated in the constZc simulations forced with annual mean currents while the varZc
simulations captures the seasonal upswing in production (Figure 17). The improved JJAS mean
and annual mean export production by $43.51 \pm 8.6$ g C m$^{-2}$ yr$^{-1}$ and $30.28 \pm 13.7$ g C m$^{-2}$ yr$^{-1}$,
respectively is a clear indication of the positive impacts of a variable Zc. Similarly the
improvement in JJAS mean and annual mean new production (Figure 18) from varZc simulated
with annual mean currents were $17.39 \pm 0.8$ g C m$^{-2}$ yr$^{-1}$ and $14.81 \pm 0.1$ g C m$^{-2}$ yr$^{-1}$,





respectively. In short the varZc biological parameterization improves the export and new
productions in the model. This helps the model to capture the upwelling episodes over the study
regions. Table 5 summarizes all the results of biological sensitivity runs.

**4.2 Impact of varZc parameterization in seasonality of carbon cyle with annual**

**mean temperatures.**

Using the annual mean temperature over the WAS, we are suppressing the cooling effect of
temperature due to upwelling and quantifying how much the  model seasonality is improved by
means of varZc parameterization.  (see Section 2 for details).  The varZc simulations forced with
annual mean SST has greater JJAS mean and annual mean $CO_2$ flux by $0.88 \pm 0.1$ mol m$^{-2}$ yr$^{-1}$
and $0.28 \pm 0.07$ mol m$^{-2}$ yr$^{-1}$, respectively (Figure 19 and Table 6). For a given annual mean SST
the solubility pump largely controls the $CO_2$ emission during JJAS if a variable Zc is prescribed,
likely by the enrichment in the DIC (inferred from Figure 8b). Similarly the improvement in
$pCO_2$ (Figure 20) with varZc simulation is also remarkable. The JJAS mean and annual mean
improvements from the implementation of varZc are $11.05 \pm 1.9$ µatm and $1.91 \pm 1.4$ µatm,
respectively. The detailed quantification of $CO_2$ and $pCO_2$ responses for this experimental setup
is given in Table 6. The above analysis adds supporting evidence that the varZc simulations
strengthen the seasonality of the model compared to the constZc case. This is presumably
accomplished by the more accurate compensation depth and production zone implied with a
variable Zc.




**5. Summary and Conclusions**
A spatially and temporally varying compensation depth parameterization as a function of
solar radiation and Chl-a is implemented in the biological pump model of OCMIP-II for a
detailed analysis of biological fluxes in the upwelling zones of the Indian Ocean. The varZc
parameterization improves the seasonality of model $CO_2$ flux and $pCO_2$ variability, especially
during the monsoon period. Significant improvement is observed in the WAS where monsoon
wind-driven upwelling dominates biological production. The magnitude of $CO_2$ flux matches
with observations, especially in July when monsoon winds are at their peak. Monsoon triggers
upwelling in SLD as well which acts as a source of $CO_2$ to the atmosphere. The seasonal and
annual mean are underestimated with constZc and the SLD is reduced to a sink of $CO_2$ flux. The
varZc simulations modify the seasonal and annual means of $CO_2$ flux of SLD and depict it as a
source of $CO_2$ especially during the monsoon, but the magnitude is still underestimated
compared to Takahashi et al., (2009) observations. The SCTR variability is underestimated by
both constZc and varZc simulations, portraying it as a $CO_2$ sink region whereas observations
over the monsoon period indicate that the thermocline ridge driven by the open ocean wind-
stress curl is in fact an oceanic source of $CO_2$. However, the varZc simulation reduced the
magnitude of the sink in this region bringing it relatively closer to observations.
VarZc biological parameterization strengthens the export and new productions in the model,
which allows it to a represent better seasonal cycle of $CO_2$ and $pCO_2$ over the study regions. The
WAS export production is remarkably improved by $62.37 \pm 7.8$ g C m$^{-2}$ yr$^{-1}$ compared to
constZc. This supports our conclusion that the varZc parameterization increases the strength of
biological export in the model. Over the SLD, the JJAS seasonal mean export and new
productions are underestimated in varZc compared to constZc simulations, but the annual mean



export production is improved. Export production at the SC and SCTR are highly exaggerated
and there is hardly any improvement in new production with a variable Zc especially over the
monsoon period. The inability of varZc parameterization to improve the seasonality of SC and
SCTR may be due to the interannnual variability of biological production associated with the
Indonesian throughflow and remote forcing of the mixed layer-thermocline interactions and the
effect of biases in the windstress data used as a physical driver in the model.
Further sensitivity experiments carried out with providing annual mean currents or
temperatures in selected subdomains reveal that the varZc retains the seasonality of carbon
fluxes, $pCO_2$, export and new productions in the right direction as in the observations. This
strongly supports our contention that varZc parameterization improves the export and new
productions and it is also efficient in capturing upwelling episodes of the study regions. This
points out the significant role of having a proper balance in seasonal biological export and new
production in models to capture the seasonality in carbon cycle. This also confirms the role of
biological and solubility pumps in producing the seasonality of carbon cycle in the upwelling
zones.
However the underestimation of seasonality of $CO_2$ flux over the SLD and overestimation
over the SC as well as the SCTR is a cautionary flag for the study. This uncertainty poses an
important scientific question as to whether the model biology over the SC and SCTR region is
not resolving the seasonality in $CO_2$ flux and $pCO_2$ properly or whether the seasonality in the
compensation depth is not able to fully capture the biological processes.
To address these questions we have used inverse modeling methods (Bayesian inversion) in
order to optimize the spatially and temporally varying compensation depth using surface $pCO_2$ as




the observational constraints and computed the optimized biological production. The results will
be reported elsewhere.




















**Appendix - A**

For $Z < Z_c$,

$$J_{prod} = \frac{1}{\tau}([PO_4] - [PO_4^*]), \qquad [PO_4] > [PO_4^*] \tag{A1}$$

$$J_{DOP} = \sigma J_{prod} - \kappa[DOP] \tag{A2}$$

$$J_{PO4} = -J_{prod} + \kappa[DOP] \tag{A3}$$

$$J_{ca} = R r_{C:P}(1-\sigma)J_{prod} \tag{A4}$$

$$J_{DIC} = r_{C:P}J_{PO4} + J_{ca} \tag{A5}$$

$$J_{ALK} = -r_{N:P}J_{PO4} + 2J_{ca} \tag{A6}$$

For $Z > Z_c$,

$$J_{prod} = 0, \qquad\qquad [PO_4] \leq [PO_4^*] \tag{A7}$$

$$J_{DOP} = -\kappa[DOP] \tag{A8}$$

$$J_{PO4} = -\frac{\partial F}{\partial Z} + \kappa[DOP] \tag{A9}$$

$$F(Z) = F_c\left(\frac{Z}{Z_c}\right)^{-a} \tag{A10}$$

$$F_c = (1-\sigma)\int_0^{Z_c} J_{prod}\, dZ \tag{A11}$$

$$J_{ca} = -\frac{\partial F_{Ca}}{\partial Z} \tag{A12}$$

$$F_{Ca} = R r_{C:P} F_C e^{-(z-Z_c)/d} \tag{A13}$$



Where Z is the depth and $Z_c$ is the compensation depth in the model. $J_{prod}$ , $J_{DOP}$, $J_{PO4}$, $J_{Ca}$ are the
biogeochemical sources and sinks. Within the compensation depth ($Z_c$), the biological production
in the model $J_{prod}$ is calculated using equation A1. $[PO_4]$ is the model phosphate concentration
and $[PO_4^*]$ is observational phosphate. $\tau$ is the restoration timescale taken as 30 days. Whenever
the model phosphate exceeds the observational phosphate, it allows production.The
observational phosphate data were taken from the World Ocean Atlas (*WOA*) 1994 [*Conkright et*
*al.,* 1994]. During the biological production a fixed fraction ($\sigma J_{prod}$ ) of phosphate is converted
into Dissolved Organic Phosphate (DOP) which is a source for $J_{DOP}$ [equation A2] and
remaining $-\kappa[DOP]$ is exported downward below the compensation depth, which is further
remineralized into inorganic phosphate and made available for further biological production
[equation A3]. The downward flux of phosphate which is not converted into DOP within the
compensation depth is given by equation A11. The decrease of flux with depth due to
remineralization is shown by equation A10. The values of the constants a, $\kappa$, $\sigma$ are 0.9, (0.2 year)$^-$
$^1$ to (0.7 year)$^{-1,}$ 0.67, respectively. The rate of production is used to explain the formation of
calcium carbonate cycle in the surface waters [equation A4] and its export is given by equation
A12. Where R is the rain ratio, a constant molar ratio of exported particulate organic carbon to
the exported calcium carbonate flux at compensation depth. The exponential decrease of calcium
carbonate flux with scale depth d is given by equation A13. The biological source or biological
sink of dissolved inorganic carbon (DIC) and alkalinity (ALK) is explained through equations
A5  and A6, respectively. Where the values of rain ratio (R) is taken as 0.07 and the Redfield
ratio, $r_{C:P} = 106$, $r_{N:P} = 16$ and scale depth d is chosen as 3500m.





**Acknowledgement**
The OCMIP-II routines were taken from (http://ocmip5.ipsl.jussieu.fr/OCMIP/). GFDL data for
OTTM is taken from (http://data1.gfdl.noaa.gov/nomads/forms/assimilation.html).Takahashi
data is taken from (http://www.ldeo.columbia.edu/res/pi/CO2/).The computations were carried
out in High Performance Computing (HPC) of Ministry of Earth Sciences (MoES), IITM. Ms.
Shikha Singh, Ms. Anju M (IITM) and Mr. Saran Rajendran (CUSAT) are thanked for initial
helps and discussions.
















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

Preliminary forecasts of Pacific bigeye tuna population trends under the A2 IPCC scenario, Prog
in Oceanography., 86, 302 – 315, doi:10.1016/j.pocean.2010.04.021, 2010.
Levy, M., Shankar, D., Andre, J. M., Shenoi, S., Durand, F., Montegut, C. B., 2007. Basin-wide
seasonal evolution of the Indian Ocean's phytoplankton blooms, J. Geophy. Res. Oceans.,
112(C12), 1978 – 2012, doi: 10.1029/2007JC004090, 2007.
Liao, X., Zhan, H., Du, Y.: Potential new production in two upwelling regions of the Western
Arabian    Sea:    Estimation    and    comparison,    J.    Geophy.    Res.    Oceans.,    121,
doi:10.1002/2016JC011707, 2016.





Matsumoto K., Tokos. K. S., Price., A. R., Cox. S. J.: First description of the Minnesota Earth
System Model for Ocean biogeochemistry (MESMO 1.0), Geosci. Model Dev., 1, 1-15,
doi:10.5194/gmd-1-1-2008, 2008.
McCreary, J., Murtugude, R., Vialard, J., Vinayachandran, P., Wiggert, J. D., Hood, R. R.,
Shankar, D., Shetye, S.: Biophysical processes in the Indian Ocean, Indian Ocean
Biogeochemical Processes and Ecological Variability., 9 – 32, doi: 10.1029/GM185, 2009.
Moisan, J. R., Moisan, A. T., Abbott, M. R.: Modelling the effect of temperature on the
maximum growth rates of phytoplankton populations, Eco. Modelling., 153, 197-215,
doi:10.1016/S0304-3800(02)00008, 2002.
Morel, A.: Optical modeling of the upper ocean in relation to its biogenous matter content (Case
1 Waters), J. Geophys. Res., 93, 10479-10, 768, doi: 10.1029/JC093iC09p10749, 1988.
Murtugudde R., McCreary J. P., Busalacchi, A. J.: Oceanic processes associated with anomalous
events in the Indian Ocean with relevance to 1997-1998, J. Geophy. Res., 105, 3295-3306, doi:
10.1029/1999JC900294, 2000.
Murtugudde, R., Busalacchi, A. J.: Interannual variability of the dynamics and thermodynamics
of the tropical Indian Ocean, J. Clim. 12, 2300-2326, doi:10.1175/1520-0442, 1999.
Murtugudde, R., Seager, R., Thoppil, P.: Arabian Sea response to monsoon variations,
Paleoceanography., 22, PA4217, doi:10.1029/2007PA001467, 2007.
Najjar, R. G., Orr, J. C.: Design of OCMIP-2 simulations of chlorofluorocarbons, the solubility
pump and common biogeochemistry, http://www.ipsl.jussieu.fr/OCMIP/., 1998.





Najjar, R. G., Keeling, R. F.: Analysis of the mean annual cycle of the dissolved oxygen
anomaly in the world ocean, J. Mar. Res., 55, 117 − 151, doi:10.1357/0022240973224481, 1997.
Najjar, R. G., Sarmiento, J. L., Toggweiler, J. R.: Downward transport and fate of organic matter
in the ocean: simulations with a general circulation model, Global biogeochem. Cycles., 6, 45-
76, doi/10.1029/91GB02718, 1992.
Naqvi, S. W. A., Moffett, J. W., Gauns, M. U., Narvekar, P. V., Pratihary, A. K., Naik, H.,
Shenoy, D. M., Jayakumar, D. A., Goepfert, T. J., Patra, P. K., Al-Azri, A., and Ahmed, S. I.:
The Arabian Sea as a high-nutrient, low-chlorophyll region during the late Southwest Monsoon,
Biogeosciences., 7, 2091-2100, doi:10.5194/bg-7-2091-2010, 2010.
Naqvi, S., Naik, H., Narvekar, P.: The Arabian Sea, in Biogeochemistry, edited by K. Black and
G. Shimmield, pp. 156 − 206, Blackwell, Oxford, 2003.
Orr, J. C., Aumont, O., Bopp, L., Calderia, K.,  Taylor, K., et. al.: Evaluation of seasonal air-sea
CO2 fluxes in the global carbon cycle models, International open Science conference (Paris, 7-
10 Jan. 2003), 2003.
Orr, J. C. and co-authors.: Estimates of anthropogenic carbon uptake from four three-
dimensional global ocean models, Glob. Biogeochem. Cycles., 15, p43 − 60, doi:
10.1029/2000GB001273, 2001.
Osawa, T., Julimantoro, S.: Study of fishery ground around Indonesia archipelago using remote
sensing data, International archives of the Photogrammetry, Remote sensing and spatial
information science., vol XXXVIII, part-8, 2010.



Parsons, T. R., Takahashi, M., Habgrave, B.: In Biological Oceanographic Processes, 3$^{rd}$ ed.,
330pp., Pergamon Press, New York, doi: 10.1002/iroh.19890740411, 1984.
Prasanna Kumar, .S., Muraleedharan, P. M., Prasad, T. G., Gauns, M., Ramaiah, N., de Souza, S.
N., Sardesai, S., Madhupratap, M.: Why is the Bay of Bengal less productive during summer
monsoon    compared    to    the    Arabian    Sea?,    Geophys.    Res.    Lett.,    29(24),    2235,
doi:10.1029/2002GL016013, 2002.
Prasanna Kumar, S., Roshin, P. R., Narvekar, J., Dinesh Kumar, P., Vivekanandan, E.: What
drives the increased phytoplankton biomass in the Arabian Sea?, Current Science, 99(I), 101 –

868    106, 2010.

Prassana Kumar. S, Ramaiah. N, Gauns. M., Sarma V. V. S. S., Muraleedharan. P. M.,
RaghuKumar. S., Dileep Kumar., Madhupratap. M.: Physical forcing of biological productivity
in the Northern Arabian Sea during the Northeast Monsoon, Deep Sea Res. Pt. II., 48, 1115-
1126, doi:10.1016/S0967-0645(00)00133-8, 2001.
Praveen, V., Ajayamohan, R. S., Valsala, V., Sandeep, S.: Intensification of upwelling along
Oman coast in a warming scenario, Geophys. Res. Lett., 43, doi:10.1002/2016GL069638, 2016.
Qasim, S. Z.: Biological productivity of the Indian Ocean, J. mar. Sci., 6, 122 – 137, 1977.
Qasim, S. Z.: Oceanography of Northern Arabian Sea, Deep Sea Res., 29(9A), 1041 – 1068,
doi:10.1016/0198-0149(82)90027-9, 1982.
Redi, M.: Oceanic isopycnal mixing by coordinate rotation, J. Phys. Oceanogr., 12, 1154 – 1158,
doi: 10.1175/1520-0485, 1982.




Roxy, M. K., Modi, A., Murtugudde, R., Valsala, V., Panickal, S., Prasanna Kumar, S.,
Ravichandran, M., Vichi, M., Levy, M.: A reduction in marine primary productivity driven by
rapid warming over the tropical Indian Ocean, 43, 826 – 833, J. Geophy. Res. Letters.,
doi:10.1002/2015GL066979, 2015.
Ryther, J.: Photosynthesis in the ocean as function of light Intensity, Limnol. Oceanogr., vol 1,
issue 1, doi: 10.4319/lo.1956.1.1.0061, 2003.
Ryther, J., Menzel, D.: On the production, composition, and distribution of organic matter in the
Western Arabian Sea, Deep Sea Research and Oceanographic Abstracts., 12(2), 199 -209.
doi:10.1016/0011-7471(65)90025-2, 1965.
Sarma V. V. S. S.: Net plankton community production in the Arabian Sea based on $O_2$ mass
balance model, Glob. biogeochem. Cycles., 18, GB4001, doi:10.1029/2003GB002198, 2004.
Sarma, V. V. S. S.: An evaluation of physical and biogeochemical processes regulating the
perennial suboxic conditions in the water column of the Arabian Sea, Global Biogeochem.
Cycles., 16, doi:10.1029/2001GB001461, 2002.
Sarmiento, J. L., and Gruber, N.: Ocean Biogeochemical Dynamics, Princeton University Press,
New Jersey, 2006.
Sarmiento, J. L., Monfray. P., Maier-Reimer., Aumont, O., Murnane, R. J., Orr, J. C.: Sea-air
$CO_2$ fluxes and carbon transport: A comparison of three ocean general circulation models,
Global Biogeochem. Cycles., 14, p1267 – 1281. doi: 10.1029/1999GB900062, 2000.
Schott, F.: Monsoon response of the Somali current and associated upwelling, Prog.Oceanogr.,
12, 357 – 381, doi:10.1016/0079-6611(83)90014-9, 1983.



Smetacek, V., and Passow, U.: Spring bloom initiation and Sverdrup's critical depth model,
Limnol. Oceanogr., 35, 228 – 234, doi: 10.4319/lo.1990.35.1.0228, 1990.
Smith, L. S.: Understanding the Arabian Sea: Reflections on the 1994-1996 Arabian Sea
Expedition, Deep Sea Res. Pt. II., 48, 1385-1402, doi:10.1016/S0967-0645(00)00144-2, 2001.
Smith, R. L., Bottero, L. S.: On upwelling in the Arabian Sea. In Angel, M (ed) A voyage of
Discovery. Pergammon Press, New York, p. 291 – 304, 1977.
Smith, S. L.: Biological indications of active upwelling in the northwestern Indian Ocean in 1964
and 1979, a comparison with peru and northwest Africa, Deep Sea Res., 31, 951 – 967,
doi:10.1016/0198-0149(84)90050-5, 1984.
Smith, S. L., Codistpoti, L. A.: Southwest monsoon of 1979: chemical and biological response of
Somali coastal waters. Science, 209, 597 – 600. doi:10.1126/science.209.4456.597, 1980.
Susanto. R., Gordon, A. L., Zheng. Q.: Upwelling along the coasts of Java and Sumatra and its
relation to ENSO, J. Geophy. Res. Lett., 28, 1599-1602, doi: 10.1029/2000GL011844, 2001.
Swallow, J. C.: Some aspects of the physical oceanography of the Indian Ocean, Deep Sea Res.,
31, 639 – 650, doi:10.1016/0198-0149(84)90032-3, 1984.
Takahashi, T., Sutherland, S. C., Wanninkhof, R., Sweeney, C., Feely, R. A., Chipman, D. W.,
Hales, B., Friederich, G., Chavez, F., Sabine, C., et al.: Climatological mean and decadal
changes in surface ocean $pCO_2$ and net sea-air $CO_2$ flux over the global oceans. Deep Sea Res.,
Pt. II., 56, 554 – 557, doi:10.1016/j.dsr2.2008.12.009, 2009.
Valsala V., Maksyutov, S.: Interannual variability of air-sea $CO_2$ flux in the north Indian Ocean,
Ocean Dynamics., 1 – 14, doi 10.1007/s10236-012-0588-7, 2013.



Valsala, K. V., Maksyutov, S., Ikeda, M.: Design and Validation of an offline oceanic tracer
transport model for a carbon cycle study, J. clim., 21, doi: 10.1175/2007JCLI2018.1, 2008.
Valsala, V., Murtugudde, R.: Mesoscale and Intraseasonal Air-Sea $CO_2$ Exchanges in the
Western Arabian Sea during Boreal Summer, Deep Sea Res. Pt. I, 103, 103-113,
doi:10.1016/j.dsr.2015.06.001, 2015.
Valsala, V., Roxy, M., Ashok, K., Murtugudde, R.: Spatio-temporal characteristics of seasonal to
multidecadal variability of $pCO_2$ and air-sea $CO_2$ fluxes in the equatorial Pacific Ocean, J.
Geophys. Res., 119, 8987 – 9012, doi:10.1002/2014JC010212, 2014.
Valsala, V., Maksyutov, S., Murtugudde, R.: Interannual to Interdecadal Variabilities of the
Indonesian Throughflow Source Water Pathways in the Pacific Ocean, J. Phys. Oceanogr., 41,
1921–1940, doi: 10.1175/2011JPO4561.1, 2011.
Valsala, V., Maksyutov, S.: Simulation and assimilation of global ocean $pCO_2$ and air-sea $CO_2$
fluxes using ship observations of surface ocean $pCO_2$ in a simplified biogeochemical model,
Tellus., 62B, doi: 10.1111/j.1600-0889.2010.00495, 2010.
Vialard,. J. P. and co-authors.: Air-Sea Interactions in the Seychelles-Chagos Thermocline Ridge
Region, BAMS, doi:10.1175/2008BAMS2499.1, 2009.
Vinayachandran P. N., Yamagata, T.: Monsoon Response of the Sea around Sri Lanka:
Generation of Thermal Domes and Anticyclonic Vortices, J. Phy. Oceano., 28, 1946 – 1960, doi:

940   10.1175/1520-0485, 1998.





Vinayachandran P. N., Shankar D., S. Vernekar, K. K. Sandeep, P. Amol, C. P. Neema and A.
Chatterjee.: A summer monsoon pump to keep the bay of Bengal salty, Geophys. Res. Lett., 40,
1777 – 1782, doi:10.1002/grl.50274, 2013.
Vinayachandran, P. N., Mathew, S.: Phytoplankton bloom in the Bay of Bengal during the
northeast monsoon and its intensification by cyclones, Geophy. Res. Lett., 30(11), 1572,
doi:10.1029/2002GL016717, 2003.
Vinayachandran, P. N., Chauhan, P., Mohan, M., Nayak, S.: Biological response of the sea
around    Sri    Lanka    to    summer    monsoon,    Geophys.    Res.    Lett.,    31,    L0I302,
doi:10.1029/2003GL018533, 2004.
Wang .X. J., Behrenfeld. M., Le Borgne .R., Murtugudde .R., and Boss. E.: Regulation of
phytoplankton carbon to chlorophyll ratio by light, nutrients and temperature in the equatorial
Pacific Ocean: a basin-scale model. Biogeosciences., 6, 391 – 404, doi:10.5194/bg-6-391-2009,

953    2009.

Wiggert J. D., Jones. B. H., Dickey .T D., Brink .K. H., Weller .R .A., Marra. J., Codispoti. L.
A.: The Northeast Monsoon's impact on mixing, phytoplankton biomass and nutrient cycling in
the Arabian Sea, Deep Sea Res. Pt. II, 47, 1353-1385, doi:10.1016/S0967-0645(99)00147-2,

957    2000.

Wiggert, J. D., Hood, R. R., Banse, K., Kindle, J. C.: Monsoon-driven biogeochemical processes
in the Arabian Sea, Progr. Oceanogr., 65, 176-213, doi:10.1016/j.pocean.2005.03.008, 2005.




Wiggert. J. D., Murtugudde, R. G., Christian J. R.: Annual ecosystem variability in the tropical
Indian Ocean: results of a coupled bio-physical ocean general circulation model, Deep Sea Res.
Pt. II., 53, 644-676, doi:10.1016/j.dsr2.2006.01.027, 2006.
Xie, S. P., Annamalai, H., Schott, F. A., McCreary Jr. J. P.: Structure and mechanism of south
Indian ocean climate variability, J. clim., 15, 864 – 878, doi: 10.1175/1520-0442, 2002.
Xing W., Xiaomei. L., Haigang Z., Hailong. L.: Estimates of potential new production in the
Java-Sumatra upwelling system, Chinese Journal of Oceanology and Limnology., 30, 1063-
1067, doi:10.1007/s00343-012-1281, 2012.



Yamanaka, Y., Yoshie, N, MasahikoFujii, Maka .N. Aita and Kishi. M. J.: An Ecosystem
coupled with Nitrogen-Silicon-Carbon cycles applied to station A7 in the Northwestern Pacific,
J. of Oceanogr., 60, p227-241, doi: 10.1023/B:JOCE.0000038329.91976.7d, 2004.
Zhou X., Weng. E., Luo., Y.: Modelling patterns of nonlinearity in the ecosystem responses to
temperature, $CO_2$ and precipitation changes, Eco. Appli., 18, 453 – 466, doi: 10.1890/07-0626.1,

973     2008.




**Table: 1** WAS = Western Arabian Sea, SLD = Sri Lanka Dome, SC = Sumatra Coast, SCTR = Seychelles-Chagos Thermocline Ridge. JJAS mean and Climatological annual mean of $CO_2$ flux from Takahashi observations, constZc and varZc simulations. Units are in mol m$^{-2}$ yr$^{-1}$.

| Regions | $CO_2$ flux (mol m$^{-2}$ yr$^{-1}$) | | | | | |
| --- | --- | --- | --- | --- | --- | --- |
| | JJAS Mean | | | Annual Mean | | |
| | OBS | constZc | varZc | OBS | constZc | varZc |
| **WAS** | 1.99 | 1.44 ± 0.2 | 2.31 ± 0.4 | 0.94 | 0.80 ± 0.1 | 1.07 ± 0.2 |
| **SLD** | 1.79 | -0.008 ± 0.2 | 0.24 ± 0.09 | 0.80 | -0.02 ± 0.1 | 0.10 ± 0.2 |
| **SC** | 0.31 | 0.60 ± 0.5 | 1.51 ± 1.01 | 0.21 | 0.21 ± 0.3 | 0.53 ± 0.5 |
| **SCTR** | 0.82 | -0.32 ± 0.3 | -0.05 ± 0.4 | 0.55 | -0.02 ± 0.1 | -0.07 ± 0.2 |

**Table: 2** Same as Table 1, but for $pCO_2$. Units are in µatm.

| Regions | $pCO_2$ (µatm) | | | | | |
| --- | --- | --- | --- | --- | --- | --- |
| | JJAS Mean | | | Annual Mean | | |
| | OBS | constZc | varZc | OBS | constZc | varZc |
| **WAS** | 397.58 | 389.18 ± 3.7 | 399.95 ± 5.01 | 394.69 | 389.62 ± 3.9 | 391.19 ± 4.7 |
| **SLD** | 382.44 | 371.67 ± 6.04 | 379.24 ± 8.9 | 380.21 | 370.76 ± 6.1 | 374.94 ± 9.6 |
| **SC** | 372.52 | 382.36 ± 12.7 | 402.14 ± 21.8 | 372.69 | 374.65 ± 9.3 | 381.76 ± 13.6 |
| **SCTR** | 377.18 | 365.71 ± 5.08 | 370.72 ± 7.4 | 379.89 | 372.69 ± 4.7 | 369.00 ± 5.4 |





**Table: 3** JJAS mean and Climatological annual mean of Export production from satellite derived Net Primary Production data, constZc and varZc simulations. Units are in g C m$^{-2}$ yr$^{-1}$.

| Regions | Export Production (g C m$^{-2}$ yr$^{-1}$) | | | | | |
|---|---|---|---|---|---|---|
| | JJAS Mean | | | Annual Mean | | |
| | OBS | constZc | varZc | OBS | constZc | varZc |
| WAS | 123.57 | 84.81 ± 16.04 | 147.19 ± 23.8 | 94.31 | 77.41 ± 15.1 | 122.54 ± 25.2 |
| SLD | 51.54 | 167.71 ± 59.04 | 151.51 ± 46.4 | 43.25 | 144.43 ± 49.8 | 156.08 ± 43.8 |
| SC | 58.87 | 260.11 ± 104.7 | 310.03 ± 99.5 | 54.53 | 172.52 ± 72.4 | 215.52 ± 70.8 |
| SCTR | 51.08 | 57.39 ± 14.2 | 99.23 ± 21.8 | 40.45 | 55.15 ± 17.9 | 80.35 ± 26.04 |

**Table: 4** Same as Table 3, but for New production.

| Regions | New Production (g C m$^{-2}$ yr$^{-1}$) | | | | | |
|---|---|---|---|---|---|---|
| | JJAS Mean | | | Annual Mean | | |
| | OBS | constZc | varZc | OBS | constZc | varZc |
| WAS | -- | 150.84 ± 27.9 | 133.03 ± 19.5 | -- | 108.43 ± 23.4 | 81.47 ± 15.7 |
| SLD | -- | 141.93 ± 64.1 | 77.78 ± 27.6 | -- | 111.05 ± 71.1 | 50.37 ± 26.3 |
| SC | -- | 63.64 ± 30.9 | 78.11 ± 29.1 | -- | 56.69 ± 43.3 | 54.58 ± 23.3 |
| SCTR | -- | 12.17 ± 16.3 | 13.32 ± 18.6 | -- | 13.74 ± 15.5 | 12.94 ± 13 |





**Table 5:** Table shows JJAS mean and climatological annual mean response from the model forced with annual mean currents.

| WAS region forced with Annual mean currents | JJAS mean | | | Climatological Annual mean | | |
|---|---|---|---|---|---|---|
| | constZc | varZc | Improvement | constZc | varZc | Improvement |
| $CO_2$ flux (mol m$^{-2}$ yr$^{-1}$) | 0.80 ± 0.2 | 1.29 ± 0.2 | 0.48 ± 0.04 | 0.65 ± 0.1 | 0.79 ± 0.1 | 0.13 ± 0.02 |
| pCO$_2$ (µatm) | 381.81 ± 3.4 | 387.24 ± 3.9 | 5.43 ± 0.5 | 388.68 ± 3.4 | 388.40 ± 3.6 | -0.28 ± 0.1 |
| Export production (g C m$^{-2}$ yr$^{-1}$) | 60.71 ± 4.7 | 104.22 ± 13.4 | 43.51 ± 8.6 | 74.30 ± 4.5 | 104.58 ± 18.3 | 30.28 ± 13.7 |
| New Production (g C m$^{-2}$ yr$^{-1}$) | 34.76 ± 2.3 | 52.16 ± 1.51 | 17.39 ± 0.8 | 29.91 ± 1.7 | 44.72 ± 1.6 | 14.81 ± 0.1 |

**Table 6** Same as Table 5 but from annual mean temperature simulations.

| WAS region forced with Annual mean temperature | JJAS mean | | | Climatological Annual mean | | |
|---|---|---|---|---|---|---|
| | constZc | varZc | Improvement | constZc | varZc | Improvement |
| $CO_2$ flux (mol m$^{-2}$ yr$^{-1}$) | 1.85 ± 0.2 | 2.74 ± 0.4 | 0.88 ± 0.1 | 0.81 ± 0.1 | 1.10 ± 0.2 | 0.28 ± 0.07 |
| pCO$_2$ (µatm) | 393.20 ± 3.01 | 404.26 ± 4.9 | 11.05 ± 1.9 | 384.61 ± 3.3 | 386.52 ± 4.8 | 1.91 ± 1.4 |





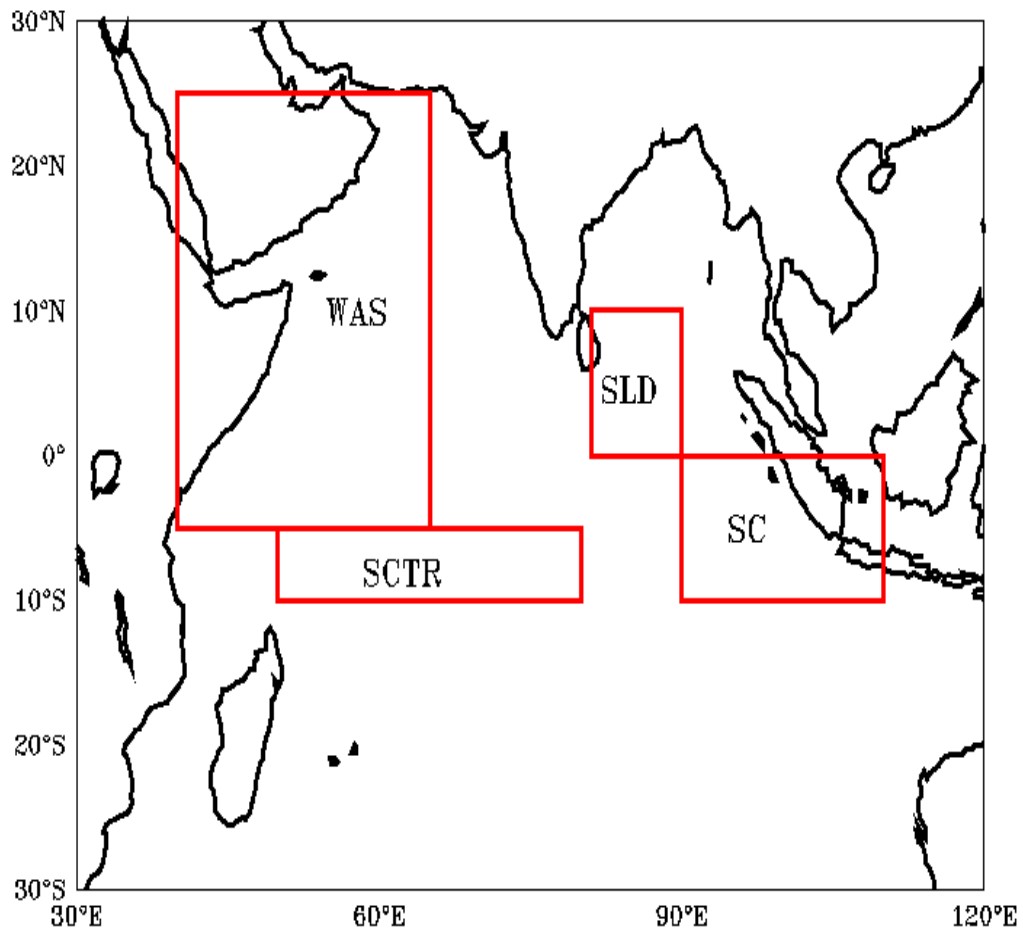

Fig (1) Red boxes shows the study regions (1) WAS (Western Arabian Sea, [40$^o$E:65$^o$E, 5$^o$S:25$^o$N]) (2) SLD (Srilankan Dome, [81$^o$E:90$^o$E, 0$^o$:10$^o$N]) (3) SCTR (Seychelles-Chagos Thermocline Ridge, [50$^o$E:80$^o$E, 5$^o$S:10$^o$S]) (4) SC (Sumatra Coast, [90$^o$E:110$^o$E, 0$^o$:10$^o$S]).



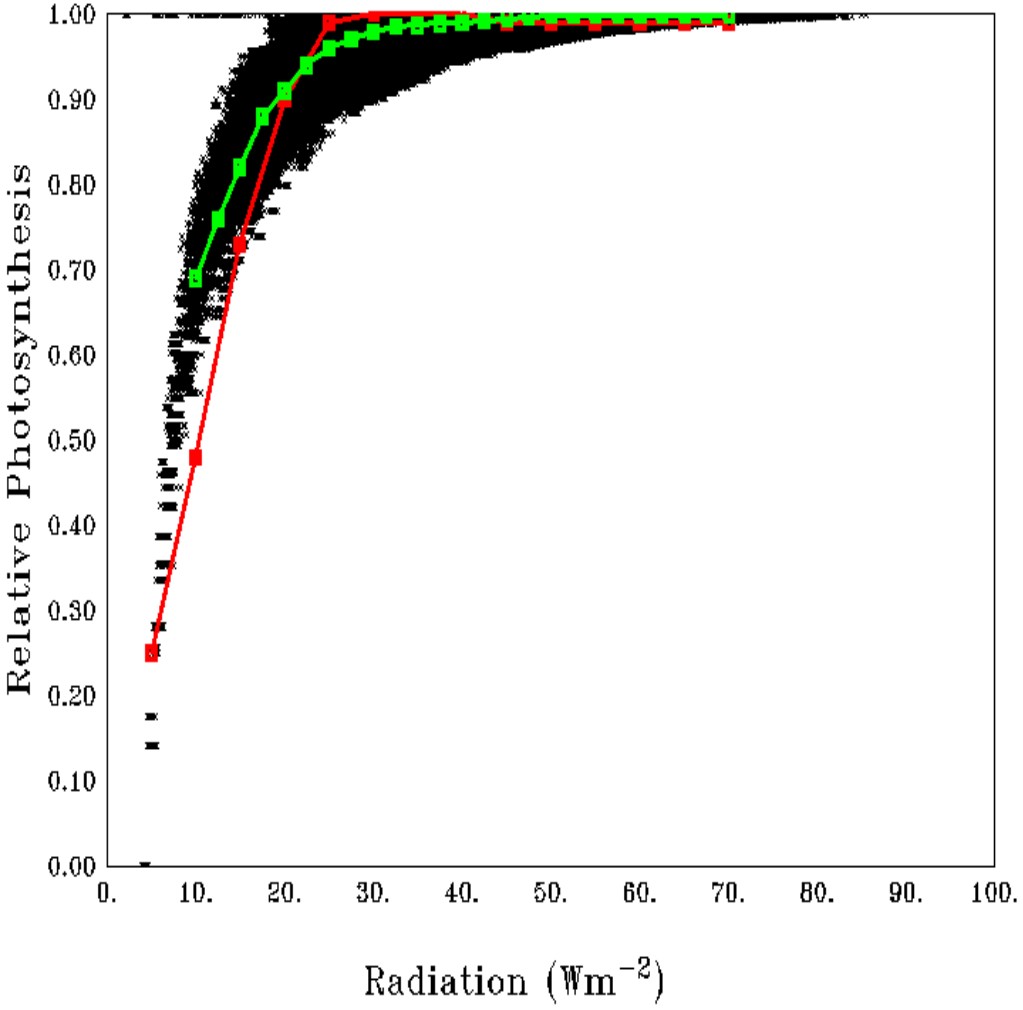

Fig (2) P − I curve, Scatter for average relative photosynthesis against different light intensities in the model. Red curve shows the theoretical P − I curve from Parsons et al., (1984). Green curve shows average of the scatter in the model.



Fig (3): Seasonal mean map of Compensation Depth (Zc) as a function of Chl-a and light (a) DJF, (b) MAM, (c) JJAS, (d) OCT-NOV. Units are in meters.







Fig (4): Annual mean biases in model evaluated against Takahashi observations for $CO_2$ flux
(a,b) and $pCO_2$ (c, d) with constant Zc (constZc) and Varying Zc (varZc). Units of $CO_2$ flux and
$pCO_2$ are mol m$^{-2}$ yr$^{-1}$ and µatm respectively.





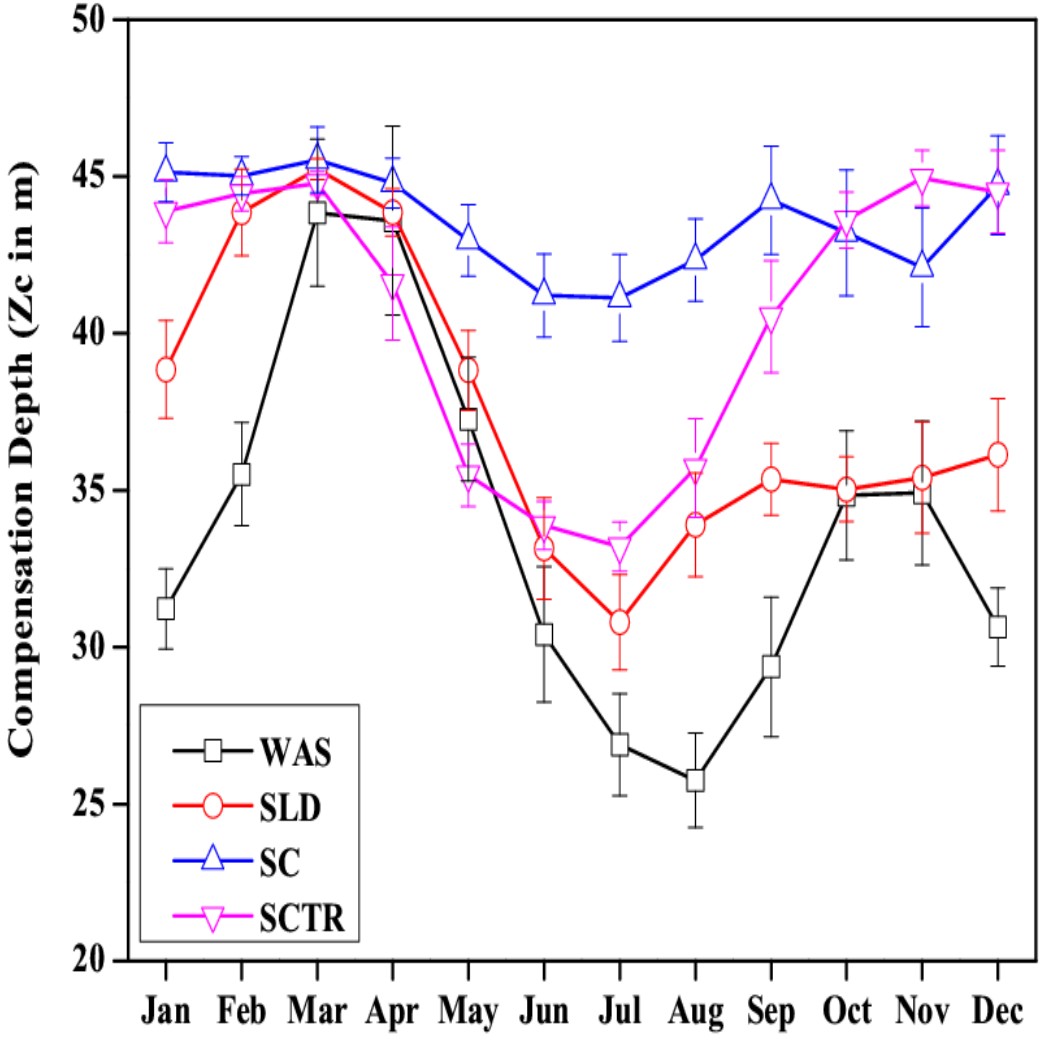

Fig (5): Seasonal variations in compensation depth (Zc) over the study regions as climatological state computed over 1990-2010. Error bar shows standard deviations of individual months over these years. Units are in meters.



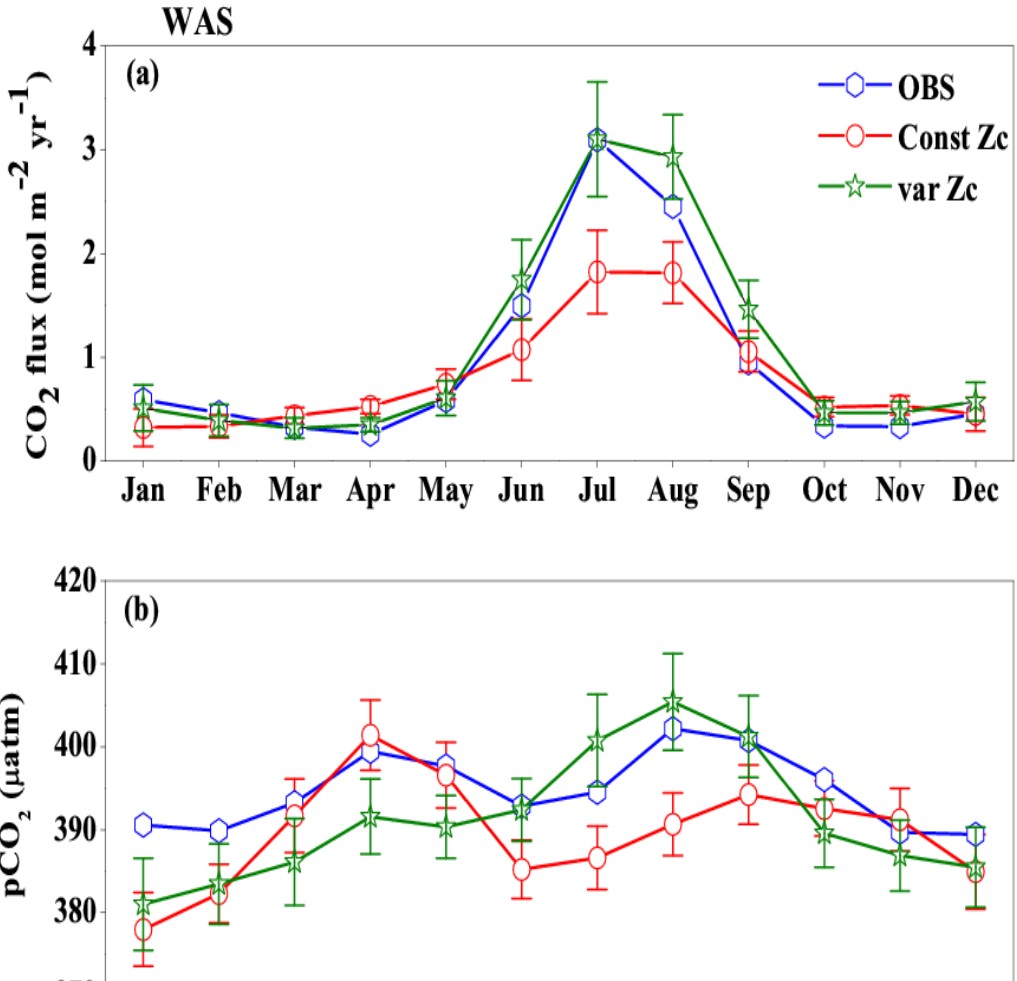

Fig (6): Comparison of model (a) $CO_2$ flux and (b) $pCO_2$ simulated with constant Zc (const Zc) and varying Zc (var Zc) with Takahashi Observations (OBS) over Western Arabian Sea (WAS) region as climatological state computed over 1990-2010. Error bar shows standard deviations of individual months over these years. Units of $CO_2$ flux and $pCO_2$ are mol m$^{-2}$ yr$^{-1}$ and µatm respectively. Legend is common for both graphs.





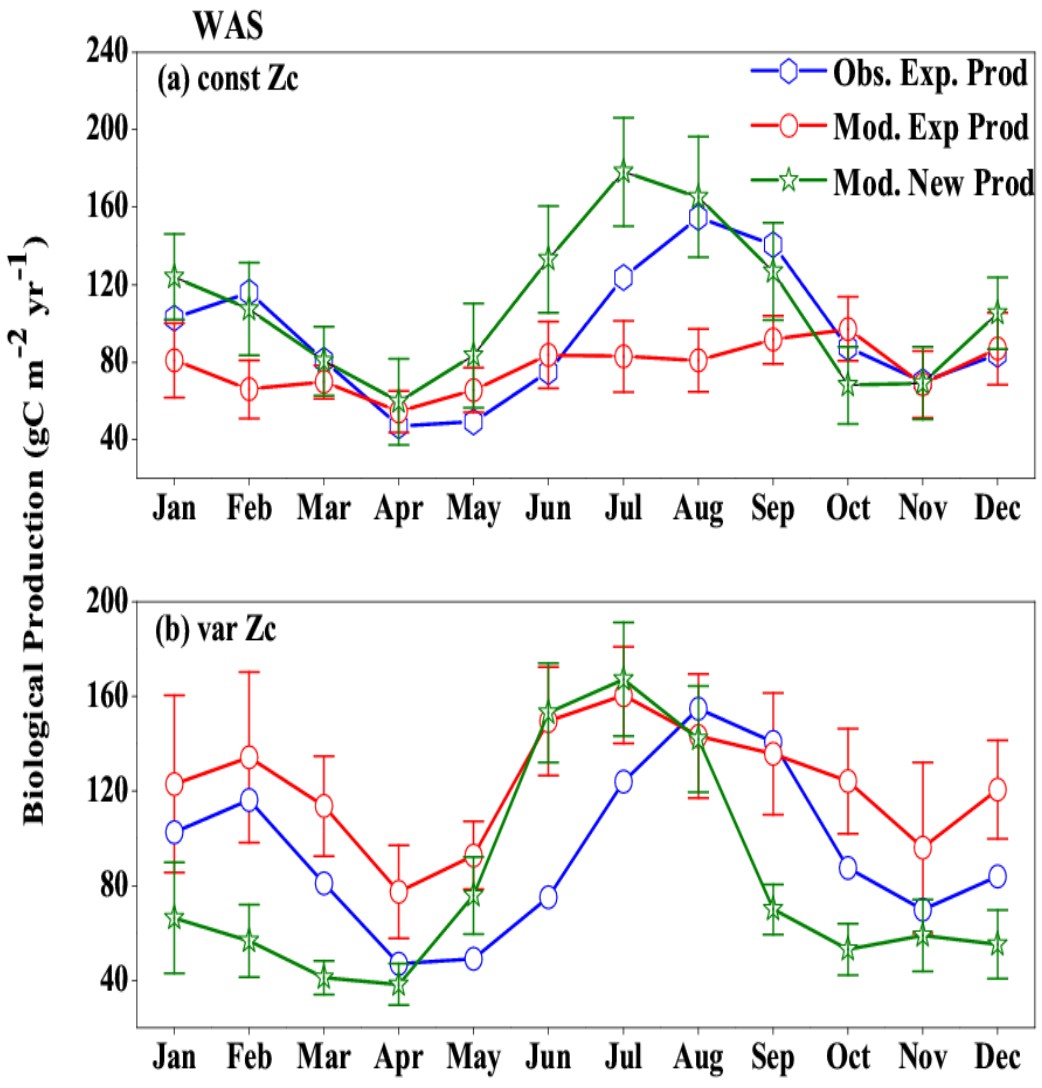

Fig (7): Comparison of model export production (Mod. Exp. Prod) and New production (Mod. New Prod) with satellite derived export production (Obs. Exp. Prod) for (a) Const Zc and (b) var Zc simulations for Western Arabian Sea (WAS) region. Units are in g C m$^{-2}$ yr$^{-1}$. Legends are common for both graphs.




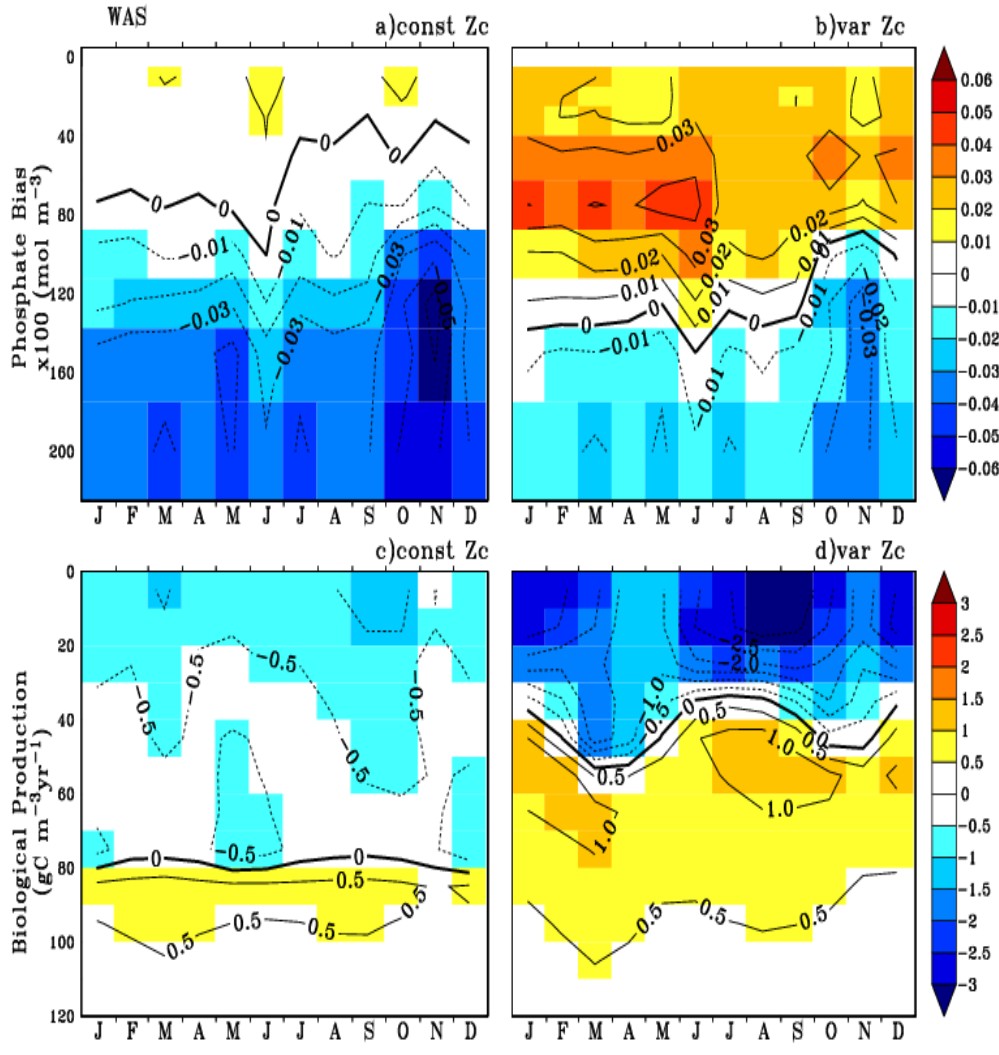

Fig (8): Bias estimation of Phosphate in the model with climatological observational data (a) const Zc and (b) Var Zc simulations and corresponding Biological production (c, d) in the model for Western Arabian Sea (WAS) region. Units of Phosphate (x 100 mol m$^{-3}$) and Biological Production (g C m$^{-3}$ yr$^{-1}$).





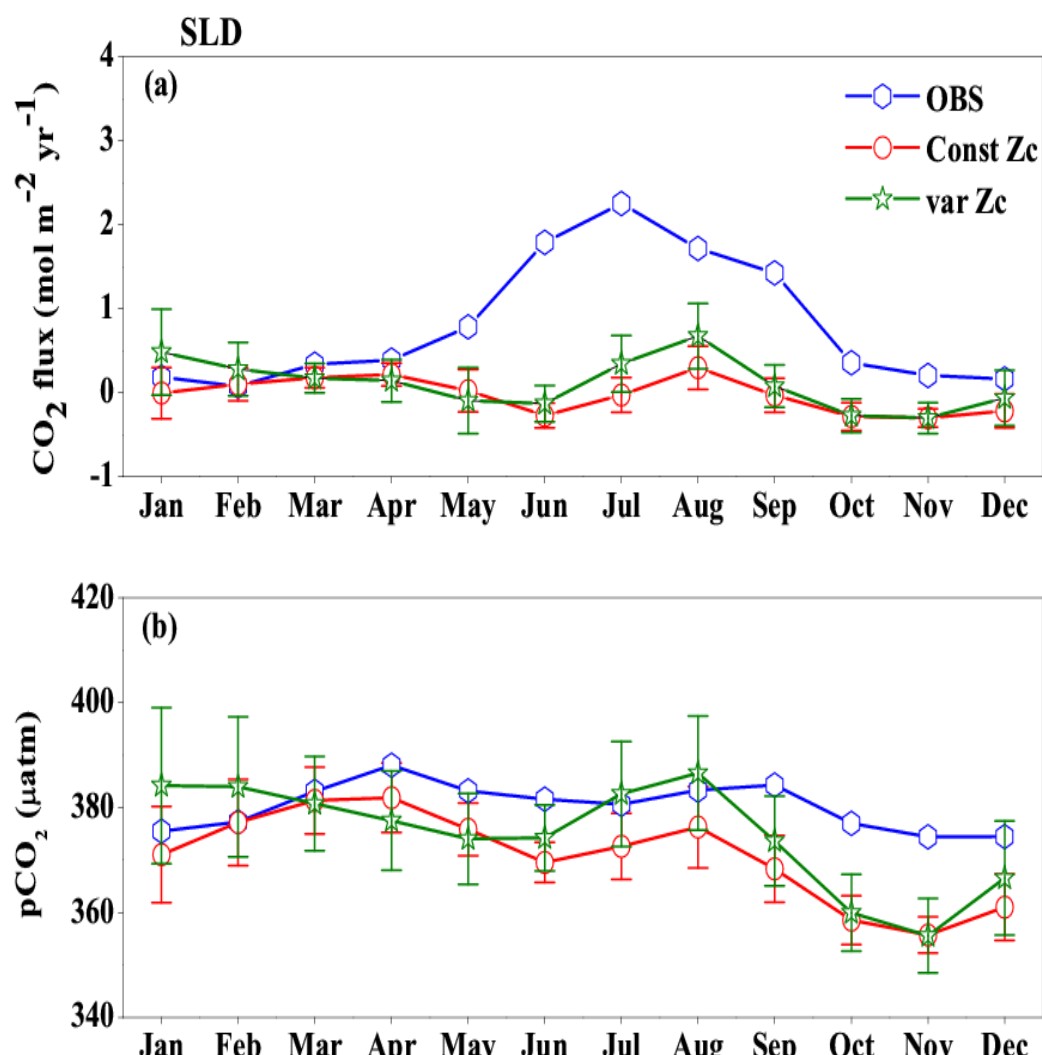

Fig (9): Same as fig (6), but for Srilankan Dome (SLD) region.





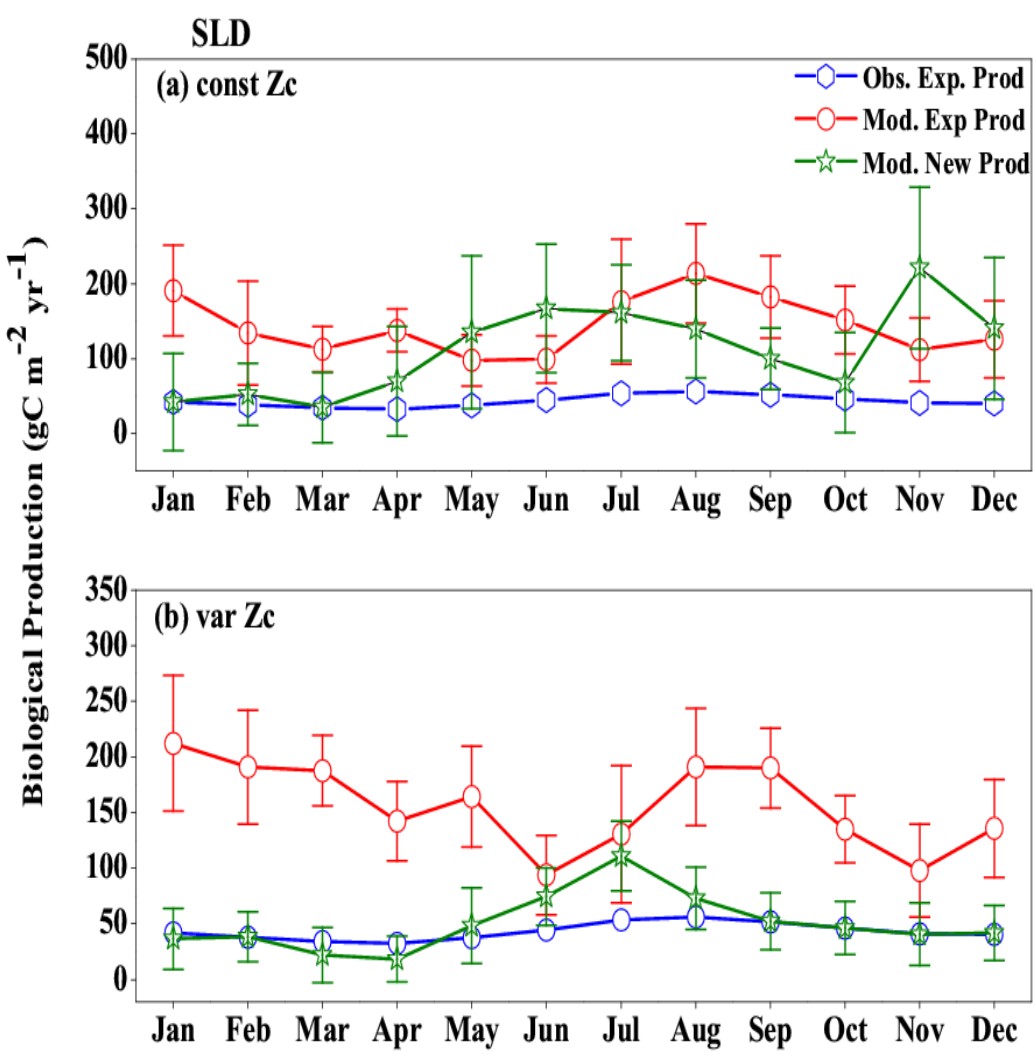

Fig (10): Same as fig (7), but for Srilankan Dome (SLD) region.





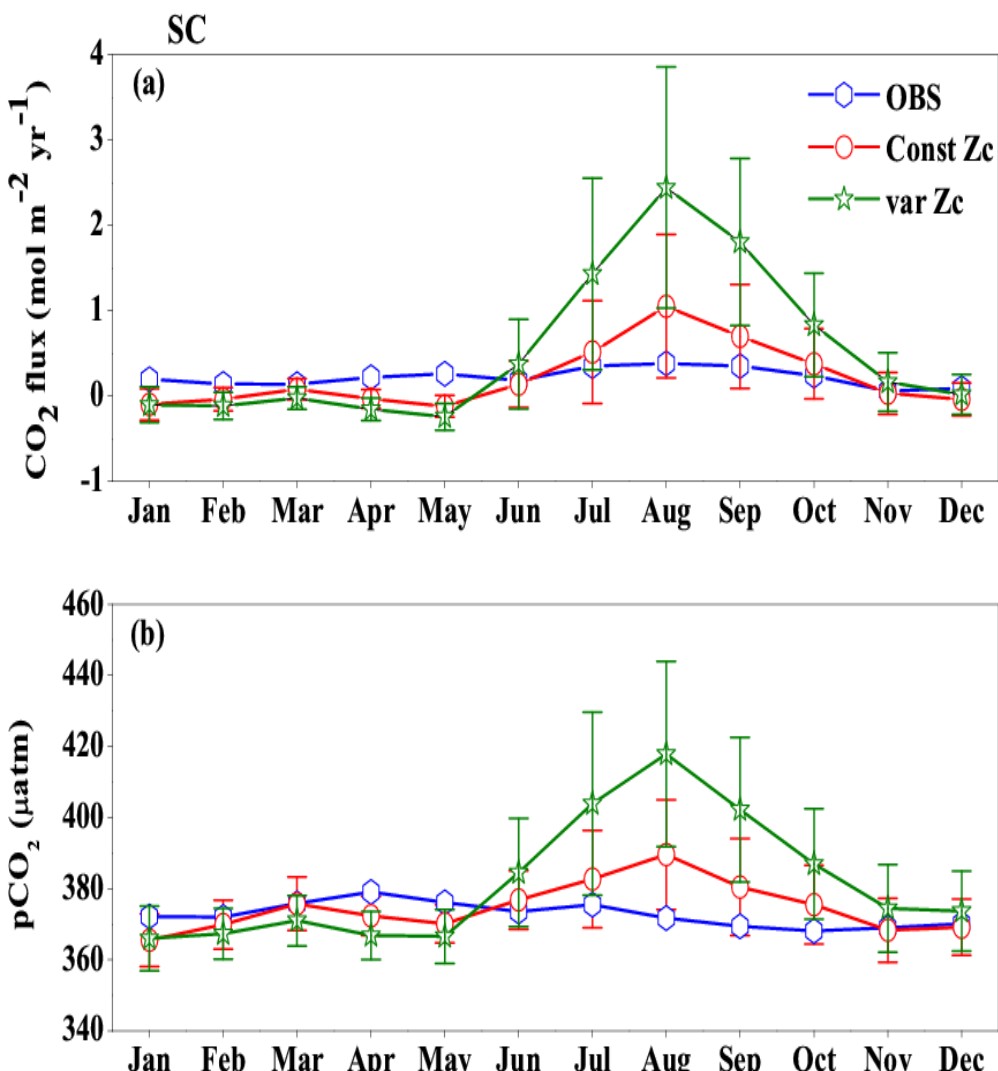

Fig (11): Same as fig (6), but for Sumatra Coast (SC) region.





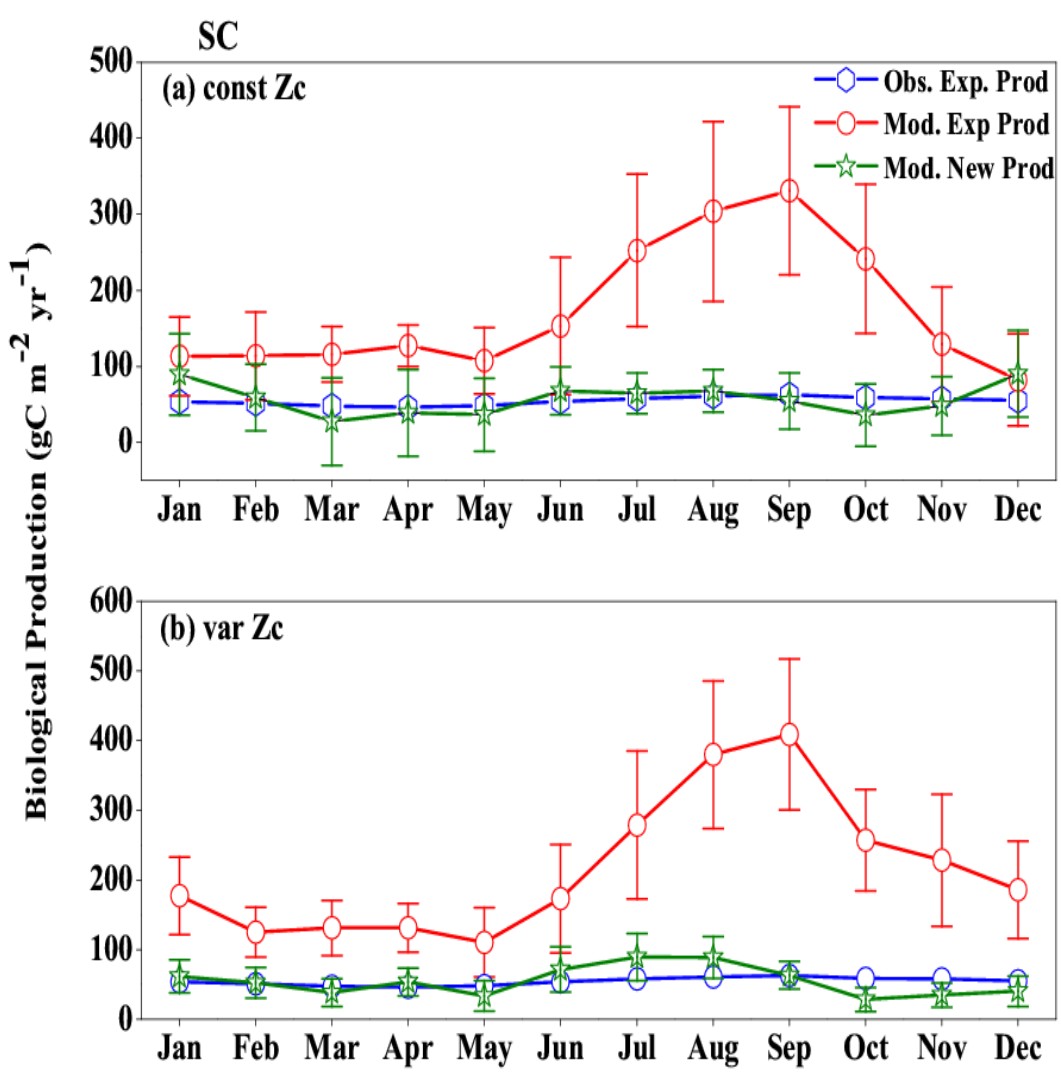

Fig (12): Same as fig (7), but for Sumatra Coast (SC) region.






Fig (13): Same as fig (6), but for Seychelles-Chagos Thermocline Ridge (SCTR) region.





Fig (14): Same as fig (7), but for Seychelles-Chagos Thermocline Ridge (SCTR) region.







Fig (15) Response of $CO_2$ flux from the model forced with annual mean currents over the Western Arabian Sea region (WAS) as climatological state computed over 1990-2010. Error bar shows standard deviations of individual months over these years. (a) const Zc and (b) var Zc. Units are in mol m$^{-2}$ yr$^{-1}$. Legends are same for both graphs.





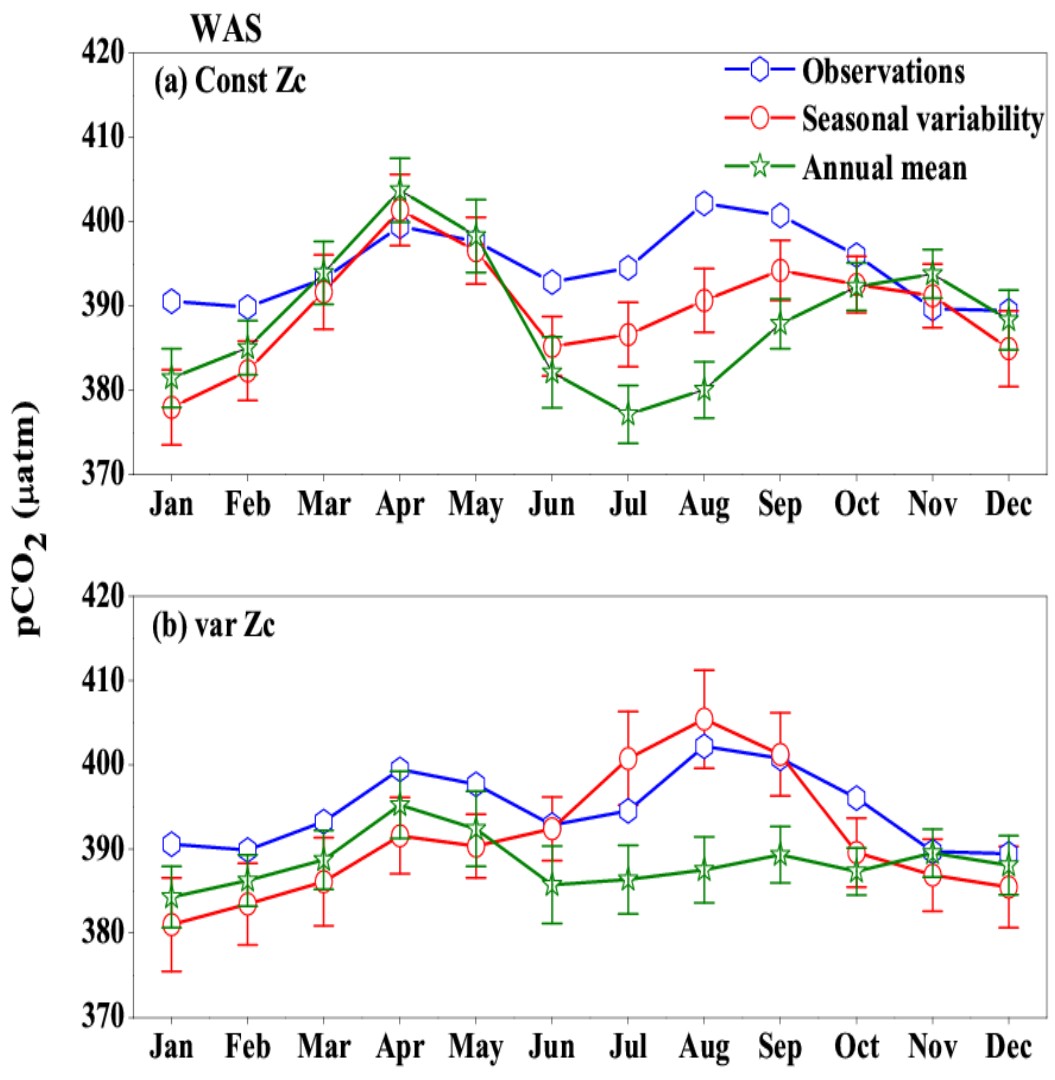

Fig (16) Same as Fig (15), but for pCO$_2$. Units are in µatm.



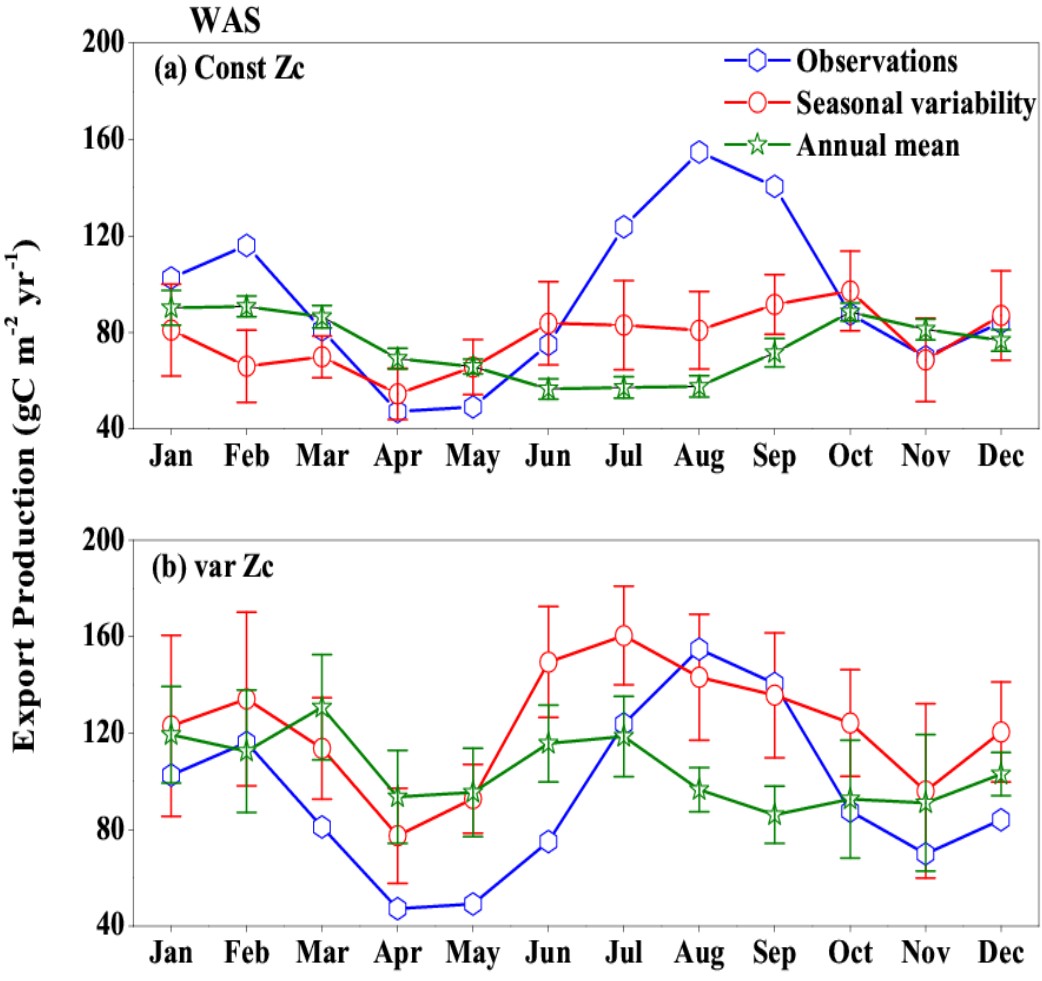

Fig (17) Response in Export Production of the model forced with annual mean currents in the Western Arabian Sea (WAS) region as climatological state computed over 1990-2010. Error bar shows standard deviations of individual months over these years. (a) Const Zc (b) varZc. Units are in g C m$^{-2}$ yr$^{-1}$. Legends are same for both graphs.





Fig (18) Same as fig (17), but for New Production. Units are in g C m$^{-2}$ yr$^{-1}$.




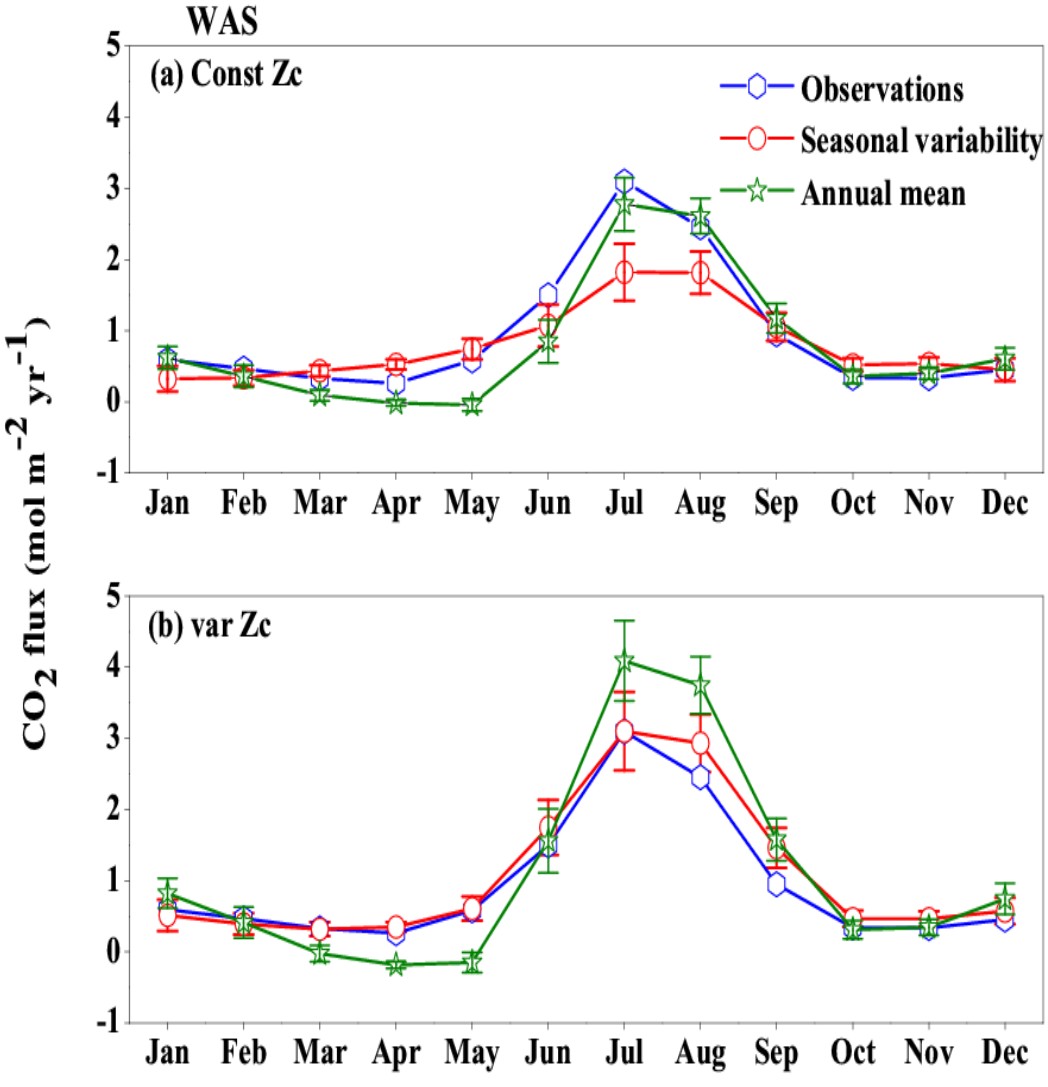

Fig (19) Response of $CO_2$ flux from the model forced with annual mean SST over the Western Arabian Sea region (WAS) as climatological state computed over 1990-2010. Error bar shows standard deviations of individual months over these years. (a) const Zc and (b) var Zc. Units are in mol $m^{-2}$ $yr^{-1}$. Legends are same for both graphs.



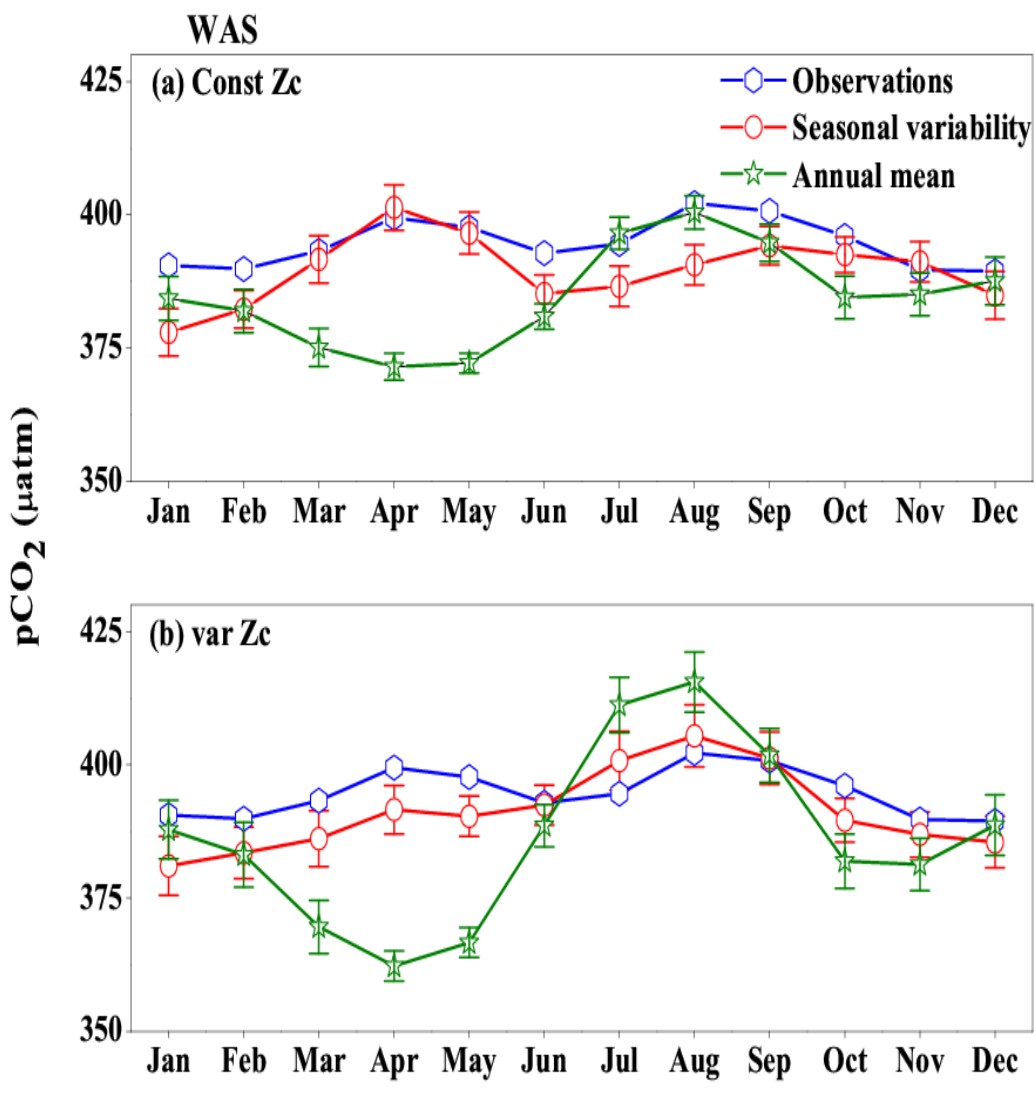

Fig (20). Same as Fig (19), But for $pCO_2$. Units are in µatm.