# Peer review of "Optimization of Biological Production for Indian Ocean upwelling zones: Part – I: Improving Biological Parameterization via a variable Compensation Depth"

_Biogeosciences, 2017_

## Referee Comment (RC1) · Anonymous Referee #1 · 4 Jul 2017

The paper introduces a new methodology to optimize a key biological parameter, the so called compensation depth ($Z_c$), which is defined as the depth at which the photosynthesis equals the respiration and is a key variable in large scale biogeochemical modeling, especially in nutrient restoring models. Usually this is set as a constant value in models, however observational evidences show a spatio-temporal variability and seasonality in $Z_c$. In this paper a spatio-temporal varying $Z_c$ is parameterized/derived from conventional wisdom of light attenuation at depths, minimum radiation required for production and optimal growth rate of biomass and came up with a spatio-temporally

varying Zc. This new parameterization has improved the model export and new production especially in the upwelling areas of the the Indian Ocean. This resulted a reduced seasonal bias in pCO2 and CO2 fluxes in the model (which is a caveat usually found in the constant Zc models) by the new parameterization.

I recommend the manuscript be published but after addressing the following concerns. 1. The introduction is too long with too many unnecessary narrations. I generally have a feeling after reading Introduction several times, the paragraphs are not carrying a 'specific message per paragraphs'. Introductions required to be synchronized. 2. The biogeochemical model used in the study requires a little more details. 3. Author(s) may explore the possibility of quantifying both the biological and solubility pumps which play an important role in the Indian Ocean upwelling zones.

---

## Referee Comment (RC2) · Anonymous Referee #2 · 21 Aug 2017

My problems with Sreeush et al. have mostly to do with their definition of the compensation depth.

1. Line 121. I disagree with citing "Sarmiento et al. 2006." First, it should actually be "Sarmiento and Gruber. 2006" since that is the only citation for 2006. Sarmiento and Gruber is a book, with only one mention of the compensation depth; hardly justifying a citation when there are whole contributions dealing with it (e.g., Marra et al., 2014, DSR 83:45-50). Better would be Ryther, from L&O, 1956, but which in the references is listed as "2003."

[Figure]

2. Line 120. They use the old symbol for irradiance. Use 'E.'

3. line 124: "Suppressed"? Not suppressed, but growth will be negative, phytoplankton will decline through respiration.

4. line 125: Here is the crux of the matter. The authors continually confuse the community compensation depth with the autotrophic compensation depth. I have argued that the latter is more appropriate, since if autotrophic production is negative, the community compensation depth will be 0 m: at the ocean's surface. The compensation irradiance is not where "planktonic photosynthesis" equals respiration, it is where GROSS photosynthesis = autotrophic respiration.

It seems as if the authors want the community compensation depth (see papers by Carlos Duarte, e.g.), and that's ok. They just have to define their parameters. Najjar and Keeling (1997), based on oxygen distributions, can give only the community compensation depth.

5. line 141: work in units of quanta, not energy. I've mad the conversion and it appears that that is equivalent to 1.7 mol photons/m2/d, or about 6% of total daily surface irradiance. For a community compensation irradiance, that might be ok, but I don't agree that that is the right parameter.

A better way to get the compensation depth is to use the base of the chlorophyll-a maximum as the bottom of the euphotic zone. There is justification for this experimentally (Marra et al., 2014), and also intuitively, in that it captures all the autotrophic biomass. This of course is the autotrophic compensation depth, which I argue is better for modeling purposes than a community compensation depth.

6. line 252: Again, there is a confusion about which compensation depth the authors are referring to. My guess is that Smetacek and Passow (1990) are talking about the community compensation depth, whereas what is mentioned here is the autotrophic compensation depth.

7. line 262: Ryther (2003)??

The rest of the ms is the working out of the model results, which I can't really comment on. But the results all stem from the compensation depth. It is not clear to me whether the model currency is oxygen or carbon.

───────────────────────────

---

## Author Comment (AC1) · 17 Sep 2017

We are thankful to the reviewer for acknowledging the importance of our work and highlighting the point that spatially and temporally varying compensation depths in the surface restoration models are indeed important. The reviewer also gave important comments to further improve the manuscript. We sincerely thank the reviewer for recommending our paper for accepting but with minor modifications. As per his/her comments, modifications/revisions have been and more analysis have been done and reported as below. We have revised the manuscript by taking into account all the com-

ments by the reviewer. A point-by-point reply to reviewer's comment is as follows. For clarity the comments are shown in blue fonts.

1. The introduction is too long with too many unnecessary narrations. I generally have a feeling after reading introduction several times, the paragraphs are not carrying a 'specific message per paragraphs'. Introductions required to be synchronized.

We thank the reviewer for a thorough reading and understanding of the manuscript. We agree to some extend to the reviewer's opinion that the introduction could be trimmed a bit for more clarity and avoid too long unnecessary narrations. This has been resolved in the revised manuscript.

2. The biogeochemical model used in the study requires a little more details.

We kindly request the reviewer to go through the appendix –A of the manuscript where we have given the entire details of the model.

3. Author(s) may explore the possibility of quantifying both the biological and solubility pumps which play an important role in the Indian Ocean upwelling zones.

This was a very valuable suggestion, to further highlight our claim that the biological pumps are better represented by the new parameterization. Though this was already given in the earlier version of the manuscript an explicit quantification and narration was missing. In the revised form we have resolved this issue.

As per the reviewer's suggestions we have conducted two additional simulations to quantify the impact of varying compensation depth in the biological and solubility pumps over the upwelling zones. The simulations were carried out from 1961 to 2010, however for further analysis; the data from 1990 to 2010 is utilized as done in the previous version of this manuscript. In these new simulations more model diagnostics were saved such as explicit profiles of biological pump in terms of DIC and calcite. We have adopted the methodology of Louanchi et. al., 1996 for the computation of the dissolved inorganic carbon tendency caused by the biological and solubility pumps.

The biological effect on dissolved inorganic carbon is calculated from the biomass production and calcite formation in the production zone expressed as below:

$(\delta DIC/\delta t)\_b = ((\delta PO\_4)/\delta t)\_b * R\_(C:P) - J\_Ca$ (1)

The total tendency due to DIC in the mixed layer depth is the sum of both the pumps (Louanchi et al., 1996).

$(\partial DIC/\partial t)\_total = (\partial DIC/\partial t)\_b + \int \_x \int \_y \Phi dx\ dy$ ãĂŮ (2)

Where $(\delta DIC/\delta t)\_b$ is evolution of dissolved inorganic carbon due to the impact of biology. $((\delta PO\_4)/\delta t)\_b$ is the rate of change of phosphate which represents the biological production in the model multiplied by the Redfield ratio ($R\_(C:P)$ = 117:1) calculated in terms of carbon and $J\_Ca$ represents the calcite formation in the model. The solubility pump is calculated by integrating the surface fluxes. Results are discussed as below.

Effect of varZc parameterization in strengthening the pump intensity over the selected upwelling regions are shown in Figure 1 (a-d) of this response note. The spatially and temporally varying compensation depth (varZc) strengthened the biological pump and solubility pump in the model as compared to constant Zc simulations.

Figure 1a shows the comparison of both solubility and biological pump over the western Arabian Sea (WAS). The analysis proves that draw down of dissolved inorganic carbon(DIC) from the production zone due to biological effect is increased by the varZc thereby strengthening the biological pump in the model.

The annual mean DIC variation due to biological effect in constZc simulation is 45.49 ± 14.3 g C m-2 yr-1. However varZc parameterization increased the DIC variation due to biological effect to 126.6 ± 24.3 g C m-2 yr-1. This is clear that the varZc has increased the strength of biological pump as evidenced by the increase in DIC variations of the production zone due to biological effects.

The above analysis clearly shows that the spatially and temporally varying compensation depth significantly affects both solubility and biological pumps in the upwelling

zones.

Further, the results are consistent with the export production profile which is indirectly a measure of biological pump in the model. These are added to the revised manuscript.

Please also note the supplement to this comment:
https://www.biogeosciences-discuss.net/bg-2017-114/bg-2017-114-AC1-supplement.pdf

The table 1 summarizes the results of biological pump impact over DIC in the model due to constZc simulations and varZc simulations.

| Biological Pump (gC m$^{-2}$ yr$^{-1}$) | constZc | | varZc | |
|---|---|---|---|---|
| | JJAS Mean | Annual Mean | JJAS Mean | Annual Mean |
| WAS | 45.18 ± 14.8 | 45.49 ± 14.38 | 151.7 ± 23.8 | 126.67 ± 24.3 |
| SLD | 89.39 ± 58.1 | 108.65 ± 48.6 | 156.07 ± 48.4 | 161.15 ± 43.5 |
| SC | 235.54 ± 95.4 | 155.21 ± 67.4 | 319.16 ± 94.9 | 222.92 ± 68.7 |
| SCTR | 30.49 ± 13.4 | 26.81 ± 16.8 | 103.13 ± 19.6 | 83.98 ± 23.6 |

Table 2 summarizes the impact of varZc over the solubility pump in the model

| Solubility Pump (gC m$^{-2}$ yr$^{-1}$) | constZc | | varZc | |
|---|---|---|---|---|
| | JJAS Mean | Annual Mean | JJAS Mean | Annual Mean |
| WAS | 17.29 ± 3.5 | 9.63 ± 2.1 | 27.72 ± 4.8 | 12.92 ± 2.7 |
| SLD | -0.09 ± 2.4 | -0.32 ± 2.3 | 2.9 ± 3.5 | 1.31 ± 3.5 |
| SC | 7.22 ± 6.9 | 2.56 ± 3.8 | 18.17 ± 12.1 | 6.43 ± 6.0 |
| SCTR | -3.95 ± 3.7 | -0.35 ± 2.3 | -0.61 ± 5.3 | -0.86 ± 2.8 |

**Fig. 1.**

[Figure]

**Fig. 2.**

[Figure]

**Fig. 3.**

[Figure]

**Fig. 4.**

[Figure]

Fig. 5.

**Supplement:**

**Reply to the comments from Reviewer-1**

We are thankful to the reviewer for acknowledging the importance of our work and highlighting the point that spatially and temporally varying compensation depths in the surface restoration models are indeed important. The reviewer also gave important comments to further improve the manuscript. We sincerely thank the reviewer for recommending our paper for accepting but with minor modifications. As per his/her comments, modifications/revisions have been and more analysis have been done and reported as below.

We have revised the manuscript by taking into account all the comments by the reviewer. A point-by-point reply to reviewer's comment is as follows. For clarity the comments are shown in blue fonts.

1. The introduction is too long with too many unnecessary narrations. I generally have a feeling after reading introduction several times, the paragraphs are not carrying a 'specific message per paragraphs'. Introductions required to be synchronized.

We thank the reviewer for a thorough reading and understanding of the manuscript. We agree to some extend to the reviewer's opinion that the introduction could be trimmed a bit for more clarity and avoid too long unnecessary narrations. This has been resolved in the revised manuscript.

2. The biogeochemical model used in the study requires a little more details.

We kindly request the reviewer to go through the appendix –A of the manuscript where we have given the entire details of the model.

3. Author(s) may explore the possibility of quantifying both the biological and solubility pumps which play an important role in the Indian Ocean upwelling zones.

This was a very valuable suggestion, to further highlight our claim that the biological pumps are better represented by the new parameterization. Though this was already given in the earlier

version of the manuscript an explicit quantification and narration was missing. In the revised form we have resolved this issue.

As per the reviewer's suggestions we have conducted two additional simulations to quantify the impact of varying compensation depth in the biological and solubility pumps over the upwelling zones. The simulations were carried out from 1961 to 2010, however for further analysis; the data from 1990 to 2010 is utilized as done in the previous version of this manuscript. In these new simulations more model diagnostics were saved such as explicit profiles of biological pump in terms of DIC and calcite. We have adopted the methodology of Louanchi et. al., 1996 for the computation of the dissolved inorganic carbon tendency caused by the biological and solubility pumps.

The biological effect on dissolved inorganic carbon is calculated from the biomass production and calcite formation in the production zone expressed as below:

$$\left(\frac{\delta DIC}{\delta t}\right)_b = \left(\frac{\delta PO_4}{\delta t}\right)_b * R_{C:P} - J_{Ca} \qquad (1)$$

The total tendency due to DIC in the mixed layer depth is the sum of both the pumps (Louanchi et al., 1996).

$$\left(\frac{\partial DIC}{\partial t}\right)_{total} = \left(\frac{\partial DIC}{\partial t}\right)_b + \int_x \int_y \Phi dx\, dy \qquad (2)$$

Where $\left(\frac{\delta DIC}{\delta t}\right)_b$ is evolution of dissolved inorganic carbon due to the impact of biology. $\left(\frac{\delta PO_4}{\delta t}\right)_b$ is the rate of change of phosphate which represents the biological production in the model multiplied by the Redfield ratio ($R_{C:P} = 117:1$) calculated in terms of carbon and $J_{Ca}$ represents the calcite formation in the model. The solubility pump is calculated by integrating the surface fluxes. Results are discussed as below.

Effect of varZc parameterization in strengthening the pump intensity over the selected upwelling regions are shown in Figure 1 (a-d) of this response note. The spatially and temporally varying compensation depth (varZc) strengthened the biological pump and solubility pump in the model as compared to constant Zc simulations.

Figure 1a shows the comparison of both solubility and biological pump over the western Arabian Sea (WAS). The analysis proves that draw down of dissolved inorganic carbon(DIC) from the production zone due to biological effect is increased by the varZc thereby strengthening the biological pump in the model.

The annual mean DIC variation due to biological effect in constZc simulation is $45.49 \pm 14.3$ g C m$^{-2}$ yr$^{-1}$. However varZc parameterization increased the DIC variation due to biological effect to $126.6 \pm 24.3$ g C m$^{-2}$ yr$^{-1}$. This is clear that the varZc has increased the strength of biological pump as evidenced by the increase in DIC variations of the production zone due to biological effects.

The table 1 summarizes the results of biological pump impact over DIC in the model due to constZc simulations and varZc simulations.

| Biological Pump (gC m$^{-2}$ yr$^{-1}$) | constZc | | varZc | |
|---|---|---|---|---|
| | JJAS Mean | Annual Mean | JJAS Mean | Annual Mean |
| WAS | $45.18 \pm 14.8$ | $45.49 \pm 14.38$ | $151.7 \pm 23.8$ | $126.67 \pm 24.3$ |
| SLD | $89.39 \pm 58.1$ | $108.65 \pm 48.6$ | $156.07 \pm 48.4$ | $161.15 \pm 43.5$ |
| SC | $235.54 \pm 95.4$ | $155.21 \pm 67.4$ | $319.16 \pm 94.9$ | $222.92 \pm 68.7$ |
| SCTR | $30.49 \pm 13.4$ | $26.81 \pm 16.8$ | $103.13 \pm 19.6$ | $83.98 \pm 23.6$ |

Table 2 summarizes the impact of varZc over the solubility pump in the model

| Solubility Pump (gC m$^{-2}$ yr$^{-1}$) | constZc | | varZc | |
|---|---|---|---|---|
| | JJAS Mean | Annual Mean | JJAS Mean | Annual Mean |
| WAS | $17.29 \pm 3.5$ | $9.63 \pm 2.1$ | $27.72 \pm 4.8$ | $12.92 \pm 2.7$ |
| SLD | $-0.09 \pm 2.4$ | $-0.32 \pm 2.3$ | $2.9 \pm 3.5$ | $1.31 \pm 3.5$ |
| SC | $7.22 \pm 6.9$ | $2.56 \pm 3.8$ | $18.17 \pm 12.1$ | $6.43 \pm 6.0$ |
| SCTR | $-3.95 \pm 3.7$ | $-0.35 \pm 2.3$ | $-0.61 \pm 5.3$ | $-0.86 \pm 2.8$ |

The above analysis clearly shows that the spatially and temporally varying compensation depth significantly affects both solubility and biological pumps in the upwelling zones.

Further, the results are consistent with the export production profile which is indirectly a measure of biological pump in the model. These are added to the revised manuscript.

**Reference**

Louanchi. F., N. Metzl., and Alain Poisson: Modelling the monthly sea surface $f_{CO_2}$ fields in the Indian Ocean, Marine Chemistry, 55, 265 – 279, 1996.

[Figure]

[Figure]

b) SLD

[Figure]

c) SC

[Figure]

Figure 1(a-d): The strength of biological pump (black lines) and solubility pump (red lines) as calculated from the Laounchi et al., 1996 equation (Eq-1 and 2 of this response note). The left axis shows the biological pump and the right axis shows the solubility pump. The units are in gC m$^2$ yr$^{-1}$.

---

## Author Comment (AC2) · 17 Sep 2017

Reviewer-2 gave very valuable comments especially for clarifying certain wording issues we had in the original manuscript. We sincerely thank the reviewer for enlightening us with the idea of "autotrophic compensation depth" which otherwise will not have been contrasted explicitly versus the community compensation depth. As per the definition we learnt from Marra et al., (2014) (thank you for pointing out this paper, this clarified us certain wording issues we had), the autotrophic compensation depth is the depth where the gross primary production (GPP) balances the autotrophic respiration (Ra) or in other words the depth at which the Net Primary Production (NPP) equals zero. (ie. NPP = GPP – Ra). We admit that there were some wording issues in the original manuscript about the definition of compensation depth which created the confusion whether the compensation depth defined here is autotrophic compensation depth or community compensation depth, However, here we note that, the Ocean Carbon-cycle Model Intercomparison Project (OCMIP – II) protocol recommends the community compensation depth adapted from (Smetacek and Passow., 1990). According to this definition the community compensation depth is the depth at which the phytoplankton photosynthesis is great enough to balances the community respiration (ie. both the autotrophic and heterotrophic respiration). At the community compensation depth, the Net Community Production (NCP) is Zero. i.e. NCP = NPP - Rh (Marra et. al., 2014, Regaudie-de-Gioux, A., and C. M. Duarte, 2010, Gattuso et. al., 2006). As per the OCMIP-II protocol we have to stick to the community compensation depth, as our intention here is to suggest a spatio-temporal variability to it through Compensation Irradiance (Ecom). Defining so, the Ecom is again the minimum light irradiance at which phytoplankton photosynthesis is great enough to balance the community respiration and the depth at which the irradiance is equal to the compensation irradiance is referred as community compensation depth (Sarmiento and Gruber, 2006). Therefore there is absolutely no error in this hypothesis as per Sarmiento and Gruber, (2006) and also after reading Marra et al., (2014) and Regaudie-de-Gioux, A., and C. M. Duarte, (2010). Again highlighting from the paper of Regaudie-de-Gioux, A., and C. M. Duarte, (2010) the compensation irradiance for the community metabolism is the irradiance at which gross community primary production balances respiratory carbon losses for the ENTIRE community (Gattuso et. al., 2006, Regaudie-de-Gioux, A., and C. M. Duarte, 2010). This means that the OCMIP-II protocol suggests the compensation depth as the depth where the gross primary production is equal to the community respiration. Therefore we are not falsified in our definitions of compensation depth. At the same time we admit that in the original manuscript it was mentioned as 'compensation depth is the depth at which the photosynthesis equal to planktonic respiration' (line-214). This

we corrected in the revised manuscript.

With the literature survey suggested by the Reviewer-2 we are clear in the definition and concept of autotrophic compensation depth and community compensation depth. However the present study focus to parameterize a spatio-temporally varying compensation depth (i.e., Community compensation depth [Smetacek and Passow., 1990, Najjar and Orr, 1998]) in the OCMIP –II protocol in order see whether it reduces the seasonal biases in the carbon cycle which is reported as a caveat in model simulations by Orr et. al., 2003.

Another important point raised by the Reviewer-2 is that whether our choice of 10 w/m2 cut-off as compensation irradiance (Ecom) is justified when converted it to mole photon/m2/day. We argue that this choice is indeed justified especially in the view of following points.

1. The observations show that the primary production reduces rapidly to 20% or less of the surface value below threshold of 10 W m-2 (Parsons et. al., 1984, Ryther, 1956).

2. Higher ocean temperature (those in the tropics) enhances the respiration rates resulting in high compensation irradiance (Parsons et. al., 1984, Ryther, 1956, Lopez-Urrutia et. al., 2006, Regaudie-de-Gioux, A., and C. M. Duarte, 2010).

3. The Table-1 of Regaudie-de-Gioux, A., and C. M. Duarte, (2010) says that the 0.4±0.2 mole photon/m2/day in case of Arabian Sea which is close to what 10Wm-2.

We have revised the manuscript by taking into account all the comments by the reviewer. A point-by-point reply to reviewer's comment is as follows. For clarity the comments are shown in blue fonts.

1. Line 121. I disagree with citing "Sarmiento et. al. 2006." First, It should actually be "Sarmiento and Gruber. 2006" since that is the only citation for 2006. Sarmiento and Gruber is a book, with only one mention of the compensation depth; hardly justifying a citation when there are whole contributions dealing with it (e.g., Marra et. al., 2014,

DSR 83:45-50). Better would be Ryther, from L&O, 1956, but which in the references is listed as "2003."

We apologize for the citation error in the manuscript. We corrected the citations, Sarmiento and Gruber, 2006 and Ryther, 1956. As per your suggestion a literature review has been conducted on Marra et. al., 2014 and further added discussions based on this paper.

2. Line 120. They use the old symbol for irradiance. Use 'E'.

Modified the symbol for compensation irradiance as 'Ecom'.

3. Line 124: "Suppressed"? Not Suppressed, but the growth will be negative, phytoplankton will decline through respiration.

Corrected as suggested.

4. Line 125: Here is the crux of the matter. The authors continually confuse the community compensation depth with the autotrophic compensation depth. I have argued that the latter is more appropriate, since if autotrophic production is negative, the community compensation depth will be 0m: at the ocean's surface. The compensation irradiance is not where "planktonic photosynthesis" equals respiration, it is where GROSS photosynthesis = autotrophic respiration.

It seems as if the authors want the community compensation depth (See papers Carlos Duarte, e.g.), and that's ok. They just have to define their parameters. Najjar and Keeling (1997), based on oxygen distributions can give only the community compensation depth.

5. Line 141: work in units of quanta, not energy. I've made the conversion and it appears that is equivalent to 1.7 mol photons/m2/d. or about 6% of the total daily surface irradiance. For a community compensation irradiance, that might be ok, but I don't agree that that is the right parameter.

A better way to get the compensation depth is to use the base of the chlorophyll-a maximum as the bottom of the euphotic zone. There is justification for this experimentally (Marra et. al., 2014), and also intuitively, in that it captures all the autotrophic biomass. This of course is the autotrophic compensation depth, which I argue is better for modeling purposes than a community compensation depth.

Reply 4 & 5

Thank you for pointing out the wording confusion in the manuscript writing regarding the definition of compensation depth and educating us about the autotrophic compensation depth. Here we note that, through this study we are trying to parameterize a spatio-temporally varying compensation depth in the OCMIP –II protocol which is community compensation depth, not autotrophic compensation depth. The definition is made as "the depth at which the phytoplankton photosynthesis is great enough to balance the community respiration or the depth at which compensation irradiance for community metabolism is received (the irradiance at which gross community primary production balances respiratory carbon losses for the entire community) [Gattuso et. al., 2006, Regaudie-de-Gioux, A., and C. M. Duarte, 2010, Sarmiento and Gruber, 2006]. As per the protocol of OCMIP-II we stick to the community compensation depth because of following reasons:

(a) The OCMIP –II protocol defines it clearly as a community compensation depth above which is the production zone and below is the consumption zone (Najjar and Orr, 1998). And the OCMIP –II models are very successful in simulating the annual mean state of the carbon cycle.

(b) If we introduce the autotrophic compensation depth, which is depth at which phytoplankton photosynthesis equal to the autotrophic respiration, we will lose the contribution of inorganic carbon sources from the heterotrophic respiration (Regaudie-de-Gioux, A., and C. M. Duarte, 2010) and there is possibility that this will affect the annual mean carbon cycle which is net effect of both autotrophic as well as heterotrophic respiration.

However we do agree that in the original manuscript there were confusions in the wording of our definitions, which we revised.

6. Line 252: Again, there is a confusion about which compensation depth the authors are reffering to. My guess is that Smetacek and Passow (1990) are talking about the community compensation depth, whereas what is mentioned here is the autotrophic compensation depth.

Corrected accordingly.

7. Line 262: Ryther (2003) ?? The rest of the ms is the working out of the model results, which I can't really comment on. But the results all stem from the compensation depth. It is not clear to me whether the model currency is oxygen or carbon.

Corrected Ryther (2003) as Ryther (1956). The model currency for OCMIP –II protocol is Phosphate and Dissolved Inorganic Carbon (Najjar and Orr, 1998).

References:

Regaudie-de-Gioux, A., and C. M. Duarte (2010), Compensation irradiance for planktonic community metabolism in the ocean, Global Biogeochem. Cycles, 24, GB4013, doi:10.1029/2009GB003639.

Gattuso, J. P., B. Gentili, C. M. Duarte, J. A. Kleypas, J. J. Middelburg, and D. Antoine (2006), Light availability in the coastal ocean: Impact on the distribution of benthic photosynthetic organisms and their contribution to primary production, Biogeosciences, 3, 489 – 513, doi:10.5194/bg-3-489-2006.

Lopez-Urrutia, A., E. San Martin, R. P. Harris, and X. Irigoien (2006), Scaling the metabolic balance of the oceans, Proc. Natl. Acad. Sci. U.S.A., 103, 8739-8744,doi:10.1073/pnas.0601137103.

Ryther, J. H. (1956), Photosythesis in the ocean as a function of the light intensity,

limnol. Oceanogr., 1, 61 -70, doi:10.4319/lo.1956.1.1.0061.

Marra, J. F., Veronica P. Lance, Robert D. Vaillancourt, Bruce R. Hargreaves (2014), Resolving the ocean's euphotic zone, Deep Sea. Res. pt. I., 83, 45 -50, doi:10.1016/j.dsr.2013.09.005. Parsons, T. R., Takahashi, M., Habgrave, B.: In Biological Oceanographic Processes, 3rd ed., 330pp., Pergamon Press, New York, doi: 10.1002/iroh.19890740411, 1984

Smetacek, V., and Passow, U.: Spring bloom initiation and Sverdrup's critical depth model, Limnol. Oceanogr., 35, 228 – 234, doi: 10.4319/lo.1990.35.1.0228, 1990.

Najjar, R. G., Orr, J. C.: Design of OCMIP-2 simulations of chlorofluorocarbons, the solubility pump and common biogeochemistry, http://www.ipsl.jussieu.fr/OCMIP/., 1998.

Sarmiento, J. L., and Gruber, N.: Ocean Biogeochemical Dynamics, Princeton University Press, New Jersey, 2006

Orr, J. C., Aumont, O., Bopp, L., Calderia, K., Taylor, K., et. al.: Evaluation of seasonal air-sea CO2 fluxes in the global carbon cycle models, International open Science conference (Paris, 7-10 Jan. 2003), 2003.

---

## Author Response (AR1)

**Contents of the File**

1. **Reply to the comments from Reviewer-1**
2. **Reply to the comments from Reviewer-2**
3. **Revised manuscript with correction marked in red color.**

1. **Reply to the comments from Reviewer-1**

We are thankful to the reviewer for acknowledging the importance of our work and highlighting the point that spatially and temporally varying compensation depths in the surface restoration models are indeed important. The reviewer also gave important comments to further improve the manuscript. As per his/her comments, analysis has been carried out and the manuscript is revised as below.

A point-by-point reply to reviewer's comment is as follows. For clarity the comments are shown in blue fonts.

1. The introduction is too long with too many unnecessary narrations. I generally have a feeling after reading introduction several times, the paragraphs are not carrying a 'specific message per paragraphs'. Introductions required to be synchronized.

We thank the reviewer for a thorough reading and understanding of the manuscript. We agree to some extend to the reviewer's opinion that the introduction could be trimmed a bit for more clarity and avoid too long unnecessary narrations. This has been resolved in the revised manuscript.

2. The biogeochemical model used in the study requires a little more details.

We kindly request the reviewer to go through the appendix –A of the manuscript where we have given the entire details of the model. We feel that it is better to leave the details in the Appendix to avoid distraction from the main message and to keep the main manuscript more focused. Thank you.

3. Author(s) may explore the possibility of quantifying both the biological and solubility pumps which play an important role in the Indian Ocean upwelling zones.

This was a very valuable suggestion, to further highlight our claim that the biological pump is better represented by the new parameterization. Though this was already stated in the earlier version of the manuscript an explicit quantification and narration was missing. In the revised form we have resolved this issue.

As per the reviewer's suggestions we have conducted two additional simulations to quantify the impact of varying compensation depth in the biological and solubility pumps over the upwelling zones. The simulations were carried out from 1961 to 2010, however for further analysis; the data from 1990 to 2010 is utilized as done in the previous version of this manuscript. In these new simulations more model diagnostics were saved such as explicit profiles of biological pump in terms of DIC and calcite. We have adopted the methodology of Louanchi et. al., (1996) for the computation of the tendency of dissolved inorganic carbon (DIC) caused by the biological and solubility pumps.

The biological effect on DIC is calculated from the biomass production and calcite formation in the production zone expressed as below:

$$\left(\frac{\partial DIC}{\partial t}\right)_b = \left(\frac{\partial PO_4}{\partial t}\right)_b * R_{C:P} - J_{Ca} \qquad (1)$$

The total tendency due to DIC in the mixed layer depth is the sum of both the pumps (Louanchi et al., 1996).

$$\left(\frac{\partial DIC}{\partial t}\right)_{total} = \left(\frac{\partial DIC}{\partial t}\right)_b + \int_x \int_y \Phi dx\, dy \qquad (2)$$

Where $\left(\frac{\partial DIC}{\partial t}\right)_b$ is the evolution of DIC due to the impact of biology. $\left(\frac{\partial PO_4}{\partial t}\right)_b$ is the rate of change of phosphate which represents the biological production in the model multiplied by the Redfield ratio ($R_{C:P} = 117:1$) calculated in terms of carbon and $J_{Ca}$ represents the calcite formation in the model. The solubility pump is calculated by integrating the surface fluxes. Results are discussed as below.

Effect of varZc parameterization in strengthening the pump intensity over the selected upwelling regions are shown in Figure 1 (a-d) of this response note. The spatially and temporally varying compensation depth (varZc) strengthened the biological pump and solubility pump in the model as compared to constant Zc simulations.

Figure 1a shows the comparison of both solubility and biological pump over the western Arabian Sea (WAS). The analysis proves that draw down of dissolved inorganic carbon(DIC) from the production zone due to biological effect is increased by the varZc thereby strengthening the biological pump in the model.

The annual mean DIC variation due to biological effect in constZc simulation is $45.49 \pm 14.3$ g C $m^{-2}$ $yr^{-1}$. However varZc parameterization increases the DIC variation due to biological pump to $126.6 \pm 24.3$ g C $m^{-2}$ $yr^{-1}$. This is clear that the varZc has increased the strength of biological pump as evidenced by the increase in DIC variations of the production zone due to biological effects.

The table 1 summarizes the results of the impact of the biological pump over DIC in the model under constZc and varZc simulations.

| Biological Pump (gC $m^{-2}$ $yr^{-1}$) | constZc | | varZc | |
|---|---|---|---|---|
| | JJAS Mean | Annual Mean | JJAS Mean | Annual Mean |
| WAS | $45.18 \pm 14.8$ | $45.49 \pm 14.38$ | $151.7 \pm 23.8$ | $126.67 \pm 24.3$ |
| SLD | $89.39 \pm 58.1$ | $108.65 \pm 48.6$ | $156.07 \pm 48.4$ | $161.15 \pm 43.5$ |
| SC | $235.54 \pm 95.4$ | $155.21 \pm 67.4$ | $319.16 \pm 94.9$ | $222.92 \pm 68.7$ |
| SCTR | $30.49 \pm 13.4$ | $26.81 \pm 16.8$ | $103.13 \pm 19.6$ | $83.98 \pm 23.6$ |

Table 2 summarizes the impact of varZc over the solubility pump in the model

| Solubility Pump (gC m$^{-2}$ yr$^{-1}$) | constZc | | varZc | |
|---|---|---|---|---|
| | JJAS Mean | Annual Mean | JJAS Mean | Annual Mean |
| WAS | 17.29 ± 3.5 | 9.63 ± 2.1 | 27.72 ± 4.8 | 12.92 ± 2.7 |
| SLD | -0.09 ± 2.4 | -0.32 ± 2.3 | 2.9 ± 3.5 | 1.31 ± 3.5 |
| SC | 7.22 ± 6.9 | 2.56 ± 3.8 | 18.17 ± 12.1 | 6.43 ± 6.0 |
| SCTR | -3.95 ± 3.7 | -0.35 ± 2.3 | -0.61 ± 5.3 | -0.86 ± 2.8 |

The above analysis clearly shows that the spatially and temporally varying compensation depth significantly affects both solubility and biological pumps in the upwelling zones.

In addition, the results are consistent with the export production profile which is indirectly a measure of the biological pump in the model. These additional findings are now added to the revised manuscript.

**Reference**

Louanchi. F., N. Metzl., and Alain Poisson: Modelling the monthly sea surface $f_{CO2}$ fields in the Indian Ocean, Marine Chemistry, 55, 265 – 279, 1996.

[Figure]

a) WAS

[Figure]

b) SLD

[Figure]

[Figure]

Figure 1(a-d): The strength of the biological pump (black lines) and solubility pump (red lines) as calculated based on Laounchi et al., (1996) equation (Eqs-1 and 2 above). The left axis shows the biological pump and the right axis shows the solubility pump. The units are in gC m$^2$ yr$^{-1}$.

**Reply to the comments from Reviewer-2**

Reviewer-2 offers very valuable comments especially for clarifying certain wording issues we had in the original manuscript. We sincerely thank the reviewer for enlightening us with the idea of "autotrophic compensation depth" which otherwise will not have been contrasted explicitly versus the community compensation depth. As per the definition from Marra et al., (2014) (thank you for pointing out this paper, this certainly clarified the wording issues we had), the autotrophic compensation depth is the depth where the gross primary production (GPP) balances the autotrophic respiration ($R_a$). In other words, it is the depth at which the Net Primary Production (NPP) equals zero. (ie. NPP = GPP – $R_a$). We admit that there were some wording issues in the original manuscript about the definition of compensation depth which created the confusion whether the compensation depth defined here is autotrophic compensation depth or community compensation depth, However, here we note that, the Ocean Carbon-cycle Model Intercomparison Project (OCMIP – II) protocol recommends the community compensation depth adapted from (Smetacek and Passow, 1990). According to this definition the community compensation depth is the depth at which the phytoplankton photosynthesis is great enough to balances the community respiration (i.e., both the autotrophic and heterotrophic respiration). At the community compensation depth, the Net Community Production (NCP) is zero. i.e., NCP = NPP - $R_h$ (Marra et al., 2014; Regaudie-de-Gioux and Duarte, 2010; Gattuso et al., 2006). As per the OCMIP-II protocol we have to stick to the community compensation depth, as our intention here is to suggest a spatio-temporal variability to it through Compensation Irradiance ($E_{com}$). In this framework, $E_{com}$ is again the minimum light irradiance at which phytoplankton photosynthesis is great enough to balance the community respiration and the depth at which the irradiance is equal to the compensation irradiance is referred as community compensation depth (Sarmiento and Gruber, 2006). Therefore there is absolutely no error in this hypothesis as per Sarmiento and Gruber, (2006) as well as Marra et al., (2014) and Regaudie-de-Gioux and Duarte, (2010). Again highlighting the definition of Regaudie-de-Gioux and Duarte (2010), the compensation irradiance for the community metabolism is the irradiance at which gross community primary production balances respiratory carbon losses for the **ENTIRE** community (Gattuso et al., 2006, Regaudie-de-Gioux and Duarte, 2010). This means that the OCMIP-II protocol assumes the compensation depth as the depth where the gross primary production is equal to the community respiration. Therefore we are accurate in our definitions of compensation depth as per OCMIP-II protocol. At the same time we admit that in the original manuscript it was mentioned as 'compensation depth is the depth at which the photosynthesis equals planktonic respiration' (line-214). We corrected this error in the revised manuscript. Thank you.

With the literature survey suggested by Reviewer-2 we are now aware of the alternative definition and concept of autotrophic compensation depth and community compensation depth. However, the present study focuses on parameterizing the spatio-temporally varying compensation depth (i.e., **Community compensation depth** [Smetacek and Passow, 1990; Najjar and Orr, 1998]) as per the OCMIP –II protocol in order to quantify how it impacts the seasonal biases in the carbon cycle since biases were reported as a caveat in model simulations by Orr et al., (2003).

Another important point raised by Reviewer-2 is whether our choice of 10 w/m$^2$ cut-off as compensation irradiance ($E_{com}$) is justified when converted to mole photon/m$^2$/day. We argue that this choice is indeed justified especially in view of the following points.

1. Observations show that primary production reduces rapidly to 20% or less of the surface value below threshold of 10 W m$^{-2}$ (Parsons et al., 1984; Ryther, 1956).
2. Higher ocean temperature (those in the tropics) enhances respiration rates resulting in high compensation irradiance (Parsons et al., 1984; Ryther, 1956; Lopez-Urrutia et al., 2006; Regaudie-de-Gioux and Duarte, 2010).
3. Table-1 of Regaudie-de-Gioux and Duarte, (2010) reports 0.4±0.2 mole photon/m2/day in case of Arabian Sea which is close to what 10 W m$^{-2}$.

We have revised the manuscript by taking into account all the comments by the reviewer. A point-by-point reply to reviewer's comment is as follows. For clarity the comments are shown in blue fonts.

1. Line 121. I disagree with citing "Sarmiento et. al. 2006." First, It should actually be "Sarmiento and Gruber. 2006" since that is the only citation for 2006. Sarmiento and Gruber is a book, with only one mention of the compensation depth; hardly justifying a citation when there are whole contributions dealing with it (e.g., Marra et. al., 2014, DSR 83:45-50). Better would be Ryther, from L&O, 1956, but which in the references is listed as "2003."

We apologize for the citation error in the manuscript. We corrected the citations, Sarmiento and Gruber, (2006) and Ryther, (1956). As per your suggestion a literature review has been conducted on Marra et al., (2014) and further added discussions based on this paper. Please see Page-6, Line-115 and Page-12, Line 257 of the revised manuscript.

2. Line 120. They use the old symbol for irradiance. Use 'E'.

Modified the symbol for compensation irradiance as '$E_{com}$'.

3. Line 124: "Suppressed"? Not Suppressed, but the growth will be negative, phytoplankton will decline through respiration.

Corrected as suggested. Please see Page-6, Line-113.

4. Line 125: Here is the crux of the matter. The authors continually confuse the community compensation depth with the autotrophic compensation depth. I have argued that the latter is more appropriate, since if autotrophic production is negative, the community compensation depth will be 0m: at the ocean's surface. The compensation irradiance is not where "planktonic photosynthesis" equals respiration, it is where GROSS photosynthesis = autotrophic respiration.

It seems as if the authors want the community compensation depth (See papers Carlos Duarte, e.g.), and that's ok. They just have to define their parameters. Najjar and Keeling (1997), based on oxygen distributions can give only the community compensation depth.

Responses to comment #4 and 5 are combined below.

5. Line 141: work in units of quanta, not energy. I've made the conversion and it appears that is equivalent to 1.7 mol photons/m2/d. or about 6% of the total daily surface irradiance. For a community compensation irradiance, that might be ok, but I don't agree that that is the right parameter.

A better way to get the compensation depth is to use the base of the chlorophyll-a maximum as the bottom of the euphotic zone. There is justification for this experimentally (Marra et. al., 2014), and also intuitively, in that it captures all the autotrophic biomass. This of course is the autotrophic compensation depth, which I argue is better for modeling purposes than a community compensation depth.

Reply 4 & 5

Thank you for pointing out the wording confusion in the manuscript writing regarding the definition of compensation depth and educating us about the autotrophic compensation depth. We want to emphasize that we are parameterizing a **spatio-temporally varying compensation depth** in the OCMIP –II protocol which is **community compensation depth,** not autotrophic compensation depth. The definition is adopted as "the depth at which the phytoplankton photosynthesis is great enough to balance the community respiration or the depth at which compensation irradiance for community metabolism is satisfied [the irradiance at which gross community primary production balances respiratory carbon losses for the entire community] (Gattuso et al., 2006; Regaudie-de-Gioux and Duarte, 2010; Sarmiento and Gruber, 2006). As per the protocol of OCMIP-II we adopt the community compensation depth because of the following reasons:

(a) The OCMIP –II protocol defines it clearly as a community compensation depth above which is the production zone and below is the consumption zone (Najjar and Orr, 1998). And the OCMIP –II models are very successful in simulating the annual mean state of the carbon cycle. This also allows us a direct comparison with previous results.

(b) If we introduce the autotrophic compensation depth, which is depth at which photosynthesis is equal to autotrophic respiration, we will lose the contribution of inorganic carbon sources from the heterotrophic respiration (Regaudie-de-Gioux and

Duarte, 2010) and there is a possibility that this will affect the annual mean carbon cycle which is net effect of both autotrophic as well as heterotrophic respiration.

However we do agree that in the original manuscript there were confusions in the wording of our definitions, which we have revised. Please see Page-5, Line-106-110 and Page-11, Line-242-243.

6. Line 252: Again, there is a confusion about which compensation depth the authors are reffering to. My guess is that Smetacek and Passow (1990) are talking about the community compensation depth, whereas what is mentioned here is the autotrophic compensation depth.

   Corrected accordingly.

7. Line 262: Ryther (2003) ??
   The rest of the ms is the working out of the model results, which I can't really comment on. But the results all stem from the compensation depth. It is not clear to me whether the model currency is oxygen or carbon.

   Corrected Ryther (2003) as Ryther (1956).
   The model currency for OCMIP –II protocol is Phosphate and Dissolved Inorganic Carbon (Najjar and Orr, 1998).

[revised manuscript text omitted]

---

## Author Response (AR2)

**Comments from the Editor**

We sincerely thank the Editor for taking such a commendable effort to allow us to clarify many points in the manuscript. We have taken utmost care in revising ALL the points raised by the editor. Action taken for each of the comment is elaborated here. We also indicate the relevant lines in the revised manuscript where the corrections are incorporated. A mark-up version also provides to easy locate the corrections made.

*All comments are given in blue fonts and replies in black fonts.*

I have read the revised version of your manuscript and based on that I find that major revisions are needed for the following reasons.

In its present form, the manuscript is still difficulty understandable for a general reader not experienced with the type of models used for OCMIP. Since the beginning, I was wondering why the compensation depth is not dynamically computed by the model then I had to read Najjar and Orr, 1992 to understand the approach. Therefore, I suggest that you clarify since the beginning the general approach of OCMIP modeling e.g. a simple biological model with a reduced number of state variables without an explicit representation of chlorophyll like in mechanistic models because …). It can be done briefly already in the introduction and then clearly in the methods section.

After reading the above comment, we read the manuscript with this view point in mind and agree with the Editor that some clarification in the introduction will indeed help the general audience who are not familiar to OCMIP-II protocols. To answer the question, 'Why the OCMIP-II by default did not include Chl-a dependent light module for biological modeling' can be answered as follows. OCMIP-II is designed for global simulations of carbon cycle for very long time scales by a family of models under a common protocol. The models were generally coarse in resolutions both horizontally and vertically. The intention was to 'inter-compare' the models run under a common protocol. The detailed light penetration and satellite Chl-a dependent modeling of biological production are more suitable in fine resolution models with more vertical levels such as the one used in the present study. Moreover, the implementation of Chl-a modules also would bring additional computational load in those days computing (late 1990s). Although all these details are not necessary in our paper, by respecting the Editor's view, we have added few sentences in the Introduction (see lines 113 - 117 of revised manuscript).

"However, Zc was held constant in time and space in OCMIP-II models (Najjar and Orr, 1998; Matsumoto et al., 2008) because the OCMIP-II protocol takes the minimalistic approach to biology and simplifies the model calculations with a very limited set of state variables suitable for long term simulations when casted in coarse resolution models (Orr et al., 2005)".

In addition to the general approach, I also find that the description of the model is insufficient. The information provided does not allow to understand the approach (even the basics). I would start the material and methods section, by informing the reader on the structure of the biogeochemical model (a scheme would be helpful), listing the state variables and interactions between them, mentioning that these variables are transported by a 3D physics. For instance, a synthesis of section 7 of the paper by Najjar and Orrr, 1992 would really help the general reader to understand the modelling approach (see also my detailed comments below).

ALL these have been carefully rectified in the revised manuscript. The detailed reply below locates them with line-numbers.

I still have concerns about the definition of the compensation depth made in the manuscript and how it translates in terms of model formulation and validation. You decided to define the compensation depth as the depth at which the community respiration equals photosynthesis but then after I noted some confusions that need to be clarified :

1) you mention that at the compensation depth, phytoplankton production is zero, which is not true (lines 112-115 ),
2) in the model formulation, you consider that below Zc, there is no biological production (Jprod=0),
3) model estimations of biological production are compared with vertically integrated primary production over the euphotic zone as proposed by Behrenfeld and Falkowski (1997). It would be helpful finally that you clearly define as early as possible the biological production (most of the time it is photosynthesis which is not the case here).

These three points raised by the Editor are extremely valuable and we are grateful to the Editor for pointing that out. The paper, in the present form, was not clear enough for general as well as expert readers. The usage of "phytoplankton production" confuses the reader in this context. In the OCMIP-II, the definition says that 'at the compensation depth, the photosynthesis is equal to total community respiration'. The phytoplankton production is not zero in this case but it represents a balance between phytoplankton production and community respiration. Therefore we have revised the manuscript relevant sections as follows.

"In this protocol, the community compensation depth (hereinafter Zc) is defined as the depth at which photosynthesis equals entire community respiration and the irradiance at which this balance is achieved is the compensation irradiance ($E_{com}$). Note that Zc is clearly different from the conventional euphotic zone depth (Morel, 1988). At Zc, the Net Community Production is zero, i. e., when the Net Primary Production (NPP) balances the community respiration and above Zc the NPP exceeds the community respiration and the ecosystem will grow (Smetacek and Passow, 1990; Gattuso et al., 2006; Sarmiento and Gruber, 2006; Regaudix-de-Gioux and Duarte, 2010; Marra et al., 2014)".

We have also removed the wording of 'negative growth' and irradiance which might have added to the confusion.

For the comparison of satellite NPP with the model export production, we first converted the satellite NPP into Export production by scaling with an e-ratio. The suitable e-ratio was taken from available observations and literature for the Arabian Sea. We admit that the confusion might have risen from the Figure labeling as 'Biological Production', which has been corrected to 'Export and New Production' in the revised manuscript.

**Point by Point Reply to the comments:**

1. Abstract: Lines 27-28: "Utilizing the principle of minimum solar radiation for the production zone..." What is the principle of minimum solar radiation?

In the revised version it is corrected as "Utilizing the criteria of surface Chl-a based attenuation of solar radiation and the minimum solar radiation required for production, we have proposed a new parameterization for a spatially and temporally varying 'compensation depth' which captures the seasonality in the production zone reasonably well." Please refer to lines 27 – 30 in the revised manuscript. Thank you for noticing this wording issue.

2. Lines 112-115: I do not agree that if the irradiance is less than Ecom, phytoplankton growth will be negative and hence phytoplankton will decline due to respiration. This would be true if Ecom represents the irradiance at which photosynthesis equals autotrophic respiration. I would like that you clarify that point. Besides, clarify what you mean by "the growth will be negative". Growth of? (line 113).

We partially agree that the statement written perhaps was not clear, so it is rewritten in simple words with more clarity. And, the revised lines are consistent with Smetacek and Passow, 1990;

Gattuso et al., 2006; Sarmiento and Gruber, 2006; Regaudix-de-Gioux and Duarte, 2010; Marra et al., 2014. Please see lines 106 – 113 in the revised manuscript.

The corrected statement is as follows:

"In this protocol, the community compensation depth (hereinafter Zc) is defined as the depth at which photosynthesis equals entire community respiration and the irradiance at which this balance is achieved is the compensation irradiance ($E_{com}$). Note that Zc is clearly different from the conventional euphotic zone depth (Morel, 1988). At  Zc, the Net Community Production is zero, i. e., when the Net Primary Production (NPP) balances the community respiration and above Zc the NPP exceeds the community respiration and the ecosystem will grow (Smetacek and Passow, 1990; Gattuso et al., 2006; Sarmiento and Gruber, 2006; Regaudix-de-Gioux and Duarte, 2010; Marra et al., 2014)."

3.  Line 116: please define what you mean by "oceanic production zone".

Corrected as "production zone". OCMIP –II protocol clearly refers to the region above the community compensation depth as the production zone where the model protocol allows production and the region below the compensation depth as the consumption zone (please see Najjar and Orr, 1998, page-10, para-4). Here we refer to the same as "production zone". Please refer to lines 112 – 114.

4.  Line 121: please avoid parentheses insides parentheses.

Corrected accordingly. Kindly see lines 121-123 in the revised manuscript.

5.  Line 127: please define "optimum" solar radiation? What do you mean by optimum? How variable will be the surface light? And which data do you use? Please specify

Thank you for pointing out the wording issue. Corrected as "production as a function of solar radiation and Chl-a availability." The data for shortwave radiation, which is scaled to PAR as 55% of the total shortwave, is taken from the same atmospheric forcing as used for the GFDL Re-analysis (Chang et al., 2012). Please refer to lines 124 -127 in the revised manuscript.

6.  Lines 127-129: This sentence is not clear and needs to be reformulated. First a reference is needed for justifying the choice of 10 W/m2 for the value of the compensation irradiance. Then, the second part of the sentence "…and calculated its depth…" needs to be reformulated. Do you mean that " a spatially and temporally varying compensation depth is estimated from a vertical light profile taking into account the shading effect using a typical? Chlorophyll profile. Please specify how the chlorophyll profile varies in space and time?

As in the replies to the comments of reviewer-2 in the first round revision and also incorporated in the first revised version of the manuscript, our justification for the choice of 10 $Wm^{-2}$ is based on following references.

1. The observations show that the primary production reduces rapidly to 20% or less of the surface value below the threshold of 10 $Wm^{-2}$ (Parsons et. al., 1984, Ryther, 1956).
2. Higher ocean temperature (those in tropics) enhances the respiration rates resulting in high compensation irradiance (Parsons et. al., 1984, Ryther, 1956, Lopez-Urrutia et al., 2006, Regaudie-de-Gioux and Duarte, 2010).
3. The Table-1 of Regaudie-de-Gioux and Duarte (2010) infers that the 0.4±0.2 mole photon/$m^2$/day in case of the Arabian Sea which is close to 10 $Wm^{-2}$.

Therefore this choice is amply justified with reference to existing literature. The citation for chlorophyll data (SeaWiFS) is given in the Acknowledgement. We have not presented any figures for chlorophyll because that has been included in several of the previous papers (Wiggert et al., 2005; Levy et. al., 2007; Prakash et al., 2012; Resplandy et al., 2009). It may also be noted that the combination of Chl-a and SSW attenuation through Chl-a mimic a varying Zc which has the inherent property of Chl-a variation and therefore our Figure 3 in the revised manuscript is more meaningful in this context.

In order to answer the Editor's comment here we show the seasonal mean map of chlorophyll variability.

[Figure]

Figure 1. Seasonal mean map of Chlorophyll-a variability over the Indian Ocean from SeaWiFS data.

7. Line 141: please clarify what do you mean by "positive" effect? Better representation?

'Positive' effect means a better representation saying that the variable compensation depth has a significant impact on both solubility as well as the biological pump. This has been revised as "the improvements due to the effect of a variable Zc". Please refer to lines 140 – 143 in the revised manuscript.

8. Line 161: why "respectively"?

Corrected accordingly. Please refer to lines 160-161.

9. Lines 163-179: Please specify that all the details that you are giving concern the biogeochemical model and if the model is solved on the same grid as the physical model. At the beginning I thought that it was for the physics and was wondering why all these details are given since you did not run the physics but rather force it from another simulation.

Yes, the BGC model is implemented on the same grid as the physical model. As per your suggestions, a detailed description of the biogeochemical model design is provided in the Model, data and methods section and more details in Appendix A.

10. Lines 180 -: Conversely, I would like to have further specifications on the biogeochemical modeling approach since the OCMIP-II protocol is not known by everybody. I think that the description of the approach that you gave at lines 181-183 is not enough for understanding the methodology and in particular, how it differs from "classic" biogeochemical modelling approach. You mention a restoration to ocean biology (which biology) and appropriate biogeochemical parameterization. Please give more details and give a reference with the methods is described because as it is, this is not enough to understand what is done.

We do agree with the editor that some more details may clarify the scenario. Please refer to section 2.2 of revised manuscript as well as Appendix A.

The "restoration biology" is a terminology common to OCMIP-II protocol meaning that the biological production is calculated as a restoration of model $PO_4$ to observational $PO_4$ from the World Ocean Atlas. See equation (2) in the manuscript (Najjar et al., 1992; Najjar and Orr, 1998).

11. Line 184: what is the first order carbon cycle?

This word has been removed for avoiding confusion.

12. Line 185: "explicit ecosystem models" please specify. I guess that it is the classic modeling approach but as said before, due to the lack of details we cannot understand how you approach differ from this "explicit" approach. Do you mean models that explicitly solve the light profile? Your model is also explicit for certain variables.

As the editor pointed out "explicit ecosystem models" is indeed classical ecosystem models with multiple compartments of NPZD. Our model is rather a "nutrient restoration approach of OCMIP-II" (Najjar et. al., 1992, Anderson and Sarmiento, 1995, Najjar and Orr, 1998). The advantage of this approach is basically ensuring the correct spatial and temporal distribution of surface nutrients, which is necessary for modeling the correct spatial and temporal distribution of surface ocean $pCO_2$ and air-sea $CO_2$ transfer.

13. Line 191: phosphorus and not phosphate

Corrected accordingly. Thank you for pointing this out.

14. Line 202: Since as far as I understood you did not run the physics? If yes, I would move what concerns the physics to this section and leaves to the model section what concerns the biology. Besides, you mention that you use ocean reanalysis products, is it for the physics only or do you use them for restoring the biology? For the remaining of the manuscript, I guess that this reanalysis is only used for the physics and so I do not see why we need to describe twice. Please clarify

Yes, we use ocean reanalysis only for physics. Ocean Tracer Transport Model (OTTM) is an offline model which uses reanalysis data from a parent model (here GFDL – MOM) to drive the physics. However in the biogeochemical model the nutrient restoration approach (Najjar et al., 1992, Anderson and Sarmiento, 1995) is achieved using Phosphate observational data (World Ocean Atlas: WOA; Garcia et. al., 2014) not with reanalysis data (For biogeochemical model details please refer Section 2.2 and Appendix A of the revised manuscript).

The repeated sentence is removed and corrected as per your suggestion.

Please refer line 217-218 in the revised manuscript.

15. Lines 208-210: Please reformulate. The initial conditions for PO4 and O2 were taken from World Ocean Atlas (Garcia et al., 2014) while for DIC and ALK data from the Global Ocean Data Analysis Project (GLODAP; Key et al., 2004) are taken. What about

DOP? Oxygen is not listed as a state variable in the appendix so why do you need to initialize it? You mention a list of data for validation like satellite estimated NPP, how do you link with model?

Dissolved Organic Phosphorous is initialized with a constant value of 0.02 μmol kg$^{-1}$. Oxygen is removed from listing. Please refer to lines 225 – 227 in the revised manuscript.

Satellite-derived Net Primary Production (NPP) data were taken from Sea-viewing Wide Field of view Sensor (SeaWiFS) Chl-a product, calculated using Vertically Generalized Production Model (VGPM: Behrenfeld and Falkowski, 1997). The NPP data is approximated to export production EP = NPP * e-ratio, here the value of e-ratio for the Indian Ocean upwelling zones is adapted as 0.37 as per Sarmiento and Gruber, 2006; Laws et al., 2000; Falkowski et al., 2003 and then compared with the OTTM model export production profiles (See Model, Data and Methods for details).

A reference is added. Please see line 235 in the revised manuscript.

17. Line225-227: Please reformulate. " This includes anthropogenic increase…" what represent "This" this is not clear what includes anthropogenic DIC?

Restated as "Since the study is focused only on bias correction to the seasonal cycle with a variable Zc, a model climatology of carbon cycle has been constructed from 1990 to 2010, which includes the anthropogenic increase of oceanic DIC in the climatological calculation and is comparable with the Takahashi et al. (2009) observations."

The time period of 1990 – 2010 is taken for climatology since the Takahashi observations are available from 1990 onwards. This is a better way to validate the model against Takahashi observations.

Please refer to line 240 – 243 of the revised manuscript.

18. Lines 228-229: please clarify the aim of these sensitivity experiments directly (you want to remove the effect of upwelling). . You remove the seasonal cycle form the physics and only impose that from Zc? (we understand afterwards reading the results section ).

The aim of conducting these sensitivity experiments is to understand how much a varZc parameterization in the biogeochemical model is successful in capturing the upwelling related pCO2 and CO2 fluxes even though the seasonal cycle in physics of upwelling mechanisms is removed when compared with a constZc simulation. By removing the seasonal cycle in currents we are suppressing the Ekman divergence over the upwelling region. Similarly, by suppressing the seasonal cycle in the temperature we are removing the cooling effect due to upwelling in the solubility pump and quantifying how much the model seasonality is improved due to varZc parameterization alone.

Please refer to lines 246 – 250 in the revised manuscript.

> 19. Line 265-266: "the model permits 100 % relative photosynthesis for radiation above 50 W/M2" what do you mean by relative photosynthesis? from equations given in the appendix, no term related to photosynthesis is presented. Do you mean Jprod is not null when the irradiance is above 50W/m2(so when Z Line 268-269: clarify the link between phot inhibition and phosphate availability as it is not clear how the availability of phosphate can be linked as a general rule with photo inhibition.

The editor is right that we do not have any explicit term to calculate the photosynthesis alone. However, in order to compare our model production of organic phosphorous to the curve of Ryther et al., we have merely scaled our total production to "relative photosynthesis", which is, according to Ryther et al., (1956) is an index between 0 and 1 indicating the strength of production estimated as Pl/Pmax,where Pl is the photosynthesis at each intensity (of light) of different species and Pmax is the maximum photosynthesis observed in the same control experiment. The curve between relative photosynthesis and light intensity shows the relation between photosynthetic activity and light in marine phytoplankton. Since our method relates the biological production to a function of light (limitation) by Chl-a attenuation, it is the best curve to cross-compare our results. Therefore we also scaled our total biological production within Zc into relative values between 0-1 by $P_l/P_{max}$. In our case, $P_l$ is taken as the individual grid cell biological component of organic phosphorus production and $P_{max}$ is the maximum production available in the domain at any given instant. All the grid points align similarly to the curve of Ryther et al., (1956) which is quite encouraging.

We have included the above steps into the Appendix B of the manuscript.

The relation between photo-inhibition and phosphate availability can be explained by taking an example of a gyre region. Indian Ocean gyre is an oligotrophic region where the continuous downwelling water at the gyre centre does not allow the nutrients to come up into the production zone (above Zc). Due to the absence of chlorophyll over this region, solar radiation penetrates deeper (approximately Zc=60m see figure 3 in the revised manuscript). But the production of organic phosphorus ($J_{prod}$) in the model is allowed at radiation values above 50 W m$^{-2}$. Since there are no sufficient new nutrients above the Zc over this region, the production of organic phosphorus is limited. Thus the limited availability of phosphate for these gyre regions above the Zc will indirectly represent photoinhibition (eventhough solar radiation is available).

20. Lines 235-238: this part is not clear. What does "x" represent and reformulate.

A smoothing technique with linear interpolation ($u = u(1 - x) + \bar{u}x$) is applied to the offline-data in order to blend the annual mean fields ($\bar{u}$) provided to the selected region with the rest of the domain (u) to reduce the sudden transition at the boundaries. Here x represents an index which varies between 0 and 1 within a distance of $10^0$ from the boundaries of the region of interest to the rest of the model domain.

Reformulated as per your suggestion. Please refer to lines 252 – 256 in the revised manuscript.

21. Line 241: compensation depth is already previously defined. Once defined, Zc should be used throughout the manuscript.

Corrected throughout the manuscript.

22. Line 252: Same remark as for Zc for varZc

Corrected accordingly.

23. Line 262: the average relative photosynthesis needs to be defined and details on how it is computed need to be provided.

Kindly refer to the reply to comment 19. Please refer to Appendix B in the revised manuscript.

24. Line 272-273: Please reformulate, this sentence is heavy. Consider for instance: "The value of Zc estimated from equation… shows marked seasonal and spatial variability (Figure 3). ..
An equation stating clearly how Zc is estimated would help.

Zc is calculated as defined in section 2.5, lines 268 -271 in the revised manuscript.

25. Line 275: "..and deepens up to 45 m from March to May due to the increase in the surface solar radiation".

Corrected accordingly. Please refer to lines 291 – 295 in the revised manuscript.

Corrected accordingly. Please refer to line 290 - 296 in the revised manuscript.

27. Line 281: Please replace compensation depth by Zc and do it through the manuscript for consistency.

Revised throughout the manuscript.

Corrected accordingly. Please refer to lines 299 – 301 in the revised manuscript.

29. Lines 283-286: Please explain how these factors can affect the penetration of light. If it is through chlorophyll, this is taken into account since in your profile you use a chlorophyll profile from the observation. Turbidity? Please clarify the link or remove.

This sentence is removed.

30. Line 299: Please clarify what does mean "optimum" here. Surface value? If yes this is not optimum.

We have removed the word "optimum" here. Please refer to lines 313 - 316 in the revised manuscript.

31. Line 299-300: reference to Parsons et al 1984 is for the parameterization used for Zc? If Yes you do not need to remind it here again (better to do in the description of the equation above) because it is confusing as we can wonder whether this reference to Parson concerns the fact that the production zone is never larger than 75m.

The citation has been removed. Please see lines 313 – 316 of the revised manuscript

32. Line 300-301: "The consequence of this on the seasonality …". The consequence of What? A varying Zc? Or a Zc limited to 75m?

Reformulated as "The relevance of varZc in the seasonality of the modeled carbon cycle is illustrated as follows." Please refer to lines 315 – 316 in the revised manuscript.

33. Lines 306-307: How pCO2 is derived from your model state variables? Give a reference in the modeling section.

A detailed derivation of $pCO_2$ from model state variables are provided at the end of this report (See Section A) and the equation for the calculation of $pCO_2$ is provided in Model, data, and Methods section 2.2 in the revised manuscript. (See the equation 7 in the revised manuscript).

34. Line 316: please explain how the new production is estimated from the model? Jprod converted into carbon? This is confusing because at line 702-703, it is mentioned: " the biological production in the model Jprod is calculated using equation A1" then lines 730-731: "… the rate of change of phosphate which represents biological production in the model …". However, the rate of phosphate change (see eq A3) also incorporates some remineralization of DOP and hence is not only linked to "primary production". As far as I know new production, is the production based on "new nutrients" (and not remineralized nutrients).

This was a technical error while writing the appendix. The first term in the R.H.S of Equation A18 is the rate of change of phosphate resulting from photosynthesis and respiration in the model (i.e., $J_{po4}$ in this case) multiplied by the carbon to phosphorous Redfield ratio ($R_{C:P}$ = 117:1) and $J_{Ca}$ represents the calcite formation in the model (see Equation A8 & A16). This has been corrected in the manuscript (see lines 754-759).

The vertically integrated new production (g C m$^{-2}$ yr$^{-1}$) in the model is calculated as per Najjar et al., (1992) as follows.

$$New\ production = -\int_{zc}^{0} J_{prod}\ dz$$

(Najjar et al., 1992)

Please refer to the Biogeochemical model section 2.2 in the revised manuscript.

35. Line 378: Please explain how the export production is estimated from satellite estimated vertically integrated production over the euphotic zone and how this can compare with the model estimated production over the compensation depth.

By the relation,

$$e\ ratio = \frac{Export\ Production}{Primary\ Production}$$

$$Export\ Production = Primary\ Production * e\ ratio$$

Here the value of primary production is used from the satellite-derived Net Primary Production (NPP) and e- ratio, which is the ratio of export production to the primary production is taken as 0.37 (Sarmiento and Gruber, 2006, Laws et al., 2000, Falkowski et al. 2003). Thus the export production is computed from the satellite-derived NPP data.

However, in the model, the export production is calculated as

$$Export\ production, EP = (1 - \sigma)\int_{0}^{Z_c} J_{prod}\ dz$$

Where $\sigma = 0.67$ is the fraction of nutrients transferring to dissolved organic phosphorous and $J_{prod}$ represents the production of organic phosphorous (Yamanaka and Tajika, 1997; Najjar and Orr, 1998; Oka et. al., 2011)

This has been added in section 2.2 of the revised manuscript.

36. Line 375: what do you mean by "new biological parameterization"? Are they related to the model that you described in the appendix? Or is it the use of varZc? In that last case please mention the use of varZc as this is how it is referred in the text. I find that using biological parameterization is confusing.

As per your suggestion, the sentence is reformulated as "The improvements shown by the use of varZc in the simulation of $CO_2$ flux and $pCO_2$ can be elicited by further analysis of the model biological production." Please refer to lines 390 – 391 of the revised manuscript.

37. Line 363: fails or simulations

Corrected accordingly. Please refer to line 378 in the revised manuscript.

38. Line 380 simulations or has

Corrected accordingly. Please refer to line 395 in the revised manuscript.

39. Line 408: please explain how Fig 8d shows that the simulated export production is consistent with satellite data.

Thank you for pointing out the error. Corrected to Figure 7b. Please refer to line 421 in the revised manuscript.

40. Line 410-411: Tables 1-4 summarize

Corrected accordingly. Please refer to line 425 in the revised manuscript.

41. Line 413 strengthened

Corrected accordingly. Please see line 427 in the revised manuscript.

42. Lines 570-571: please explain how temperature affects your biological model? For appendix this is not explained.

The sensitivity of temperature is assessed for the carbon cycle but NOT for biology. The varZc alters the DIC gradients and impact the carbon cycle. The dynamic effects are also cross-checked along with temperature effect.

Temperature sensitivity of the carbon cycle comes into play during the estimation of $pCO_2$. Takahashi et al. (1993) provide a useful relationship that summarizes the temperature sensitivity of $pCO_2$ in a closed system.

$$\frac{1}{pCO_2}\frac{\partial pCO_2}{\partial T} = \frac{\partial lnpCO_2}{\partial T} \approx 0.0423^o C^{-1}$$

For example, if we take a water parcel with an initial $pCO_2$ of 300 µatm at 20°C and with a salinity of 35, a one-degree warming increases $pCO_2$ by approximately 13 µatm, whereas a salinity increase of 1 results in a $pCO_2$ increase of 9 µatm.

Thus the effect of seasonal changes in SST on $pCO_2$ is estimated simply by multiplying the observed seasonal SST changes, $\Delta$SST by

$$\Delta pCO_2|_{thermal} \approx pCO_2.0.0423^o C^{-1}.\Delta SST$$

For the estimation of model $pCO_2$, the solubility constant $K_o$, and dissociation constants $K_1, K_2$ are computed by using the temperature and salinity relations. This will indirectly affect the $pCO_2$ seasonal cycle in the biogeochemical model. (See section A given below)

43. Appendix: the description that is made is insufficient. Please provide the evolution equations for each of your state variables (i.e. DIC, ALK, PO4, DOP, O2?).

As per your suggestions, a detailed model description is provided in Model, Data and Methods (see Section 2.2) and Appendix A. Please refer to the revised manuscript.

Also, see the end of this report (Section B)

44. Line 701: As far as I understand, JDOP, JPO4 represents the balance between sources and sinks while Jprod and JCa represent biogeochemical flows with respectively production and calcification, so J is not always the balance between sink and sources.

Corrected accordingly. Please refer to lines 727 – 729 in the revised manuscript.

45. Line 703: please define precisely what is meant by biological production.

Corrected as "the production of organic phosphorous in the model." Please refer to lines 729-730 of the revised manuscript.

46. Line 706: please clarify in equation A1 that when the phosphate concentration is lower than the climatology, there is no production.

Clarified according to your suggestion. Please refer to lines 731 - 733 in the revised manuscript.

47. Line 707: "a fixed fraction of production" rather than " a fixed fraction of phosphate", "a source for DOP"

Corrected accordingly. Please refer to lines 734 – 737 in the revised manuscript.

48. Line 708: phosphorus

Corrected accordingly.

49. Line 709: remaining is (1-sigma) Jprod and not kappa*[DOP, Kappa*[DOP] is remineralization.

Corrected accordingly.

50. Line 728: What does phi represent? In the main text, eq 1 states that phi represents the production/destruction term? In the appendix, we are told that J is the production/destruction term and here what does phi represent?

Thank you for pointing out this error. In the main text phi is replaced with F. and restaed as

"J represents any sink or source due to the internal consumption or production of the tracer. F represents the emission or absorption of fluxes at the ocean surface." (See section 2.2 in the revised manuscript)

51. Line 732: the value of Rc:p is inconsistent: here Rc:P equals 117 while at line 720 this ratio equals 106.

Corrected as $R_{C:P} = 117$. Please refer to line 746 in the revised manuscript.

52. Line 733: Phi needs to be defined. At line 171, phi is defined as a source/sink term. Now J is used to define that term.

Kindly refer to the reply to comment 50.

53. Figure 3: Seasonal maps of the compensation depth estimated from equation (give the number of the equation) … …Please explain what is the blue rectangle in Figure 3a.

We could not locate the "blue rectangle" in figure 3.

54. Table 4: I guess that it is model derived values and not satellite derived values

Corrected accordingly.

**Section A: Derivation of pCO₂ from model state variables.**

In general, ocean carbon cycle models derive $pCO_2$ based on two approaches (1) Specifying DIC and ALK and then calculating $pCO_2$ (2) Specifying $pCO_2$ and ALK and computing DIC. In OCMIP –II model protocol $pCO_2$ is derived by specifying DIC and ALK. The details are given below.

The reactions which take place when carbon dioxide dissolves in water can be represented by the following series:

$$CO_2 + H_2O \overset{K_0}{\leftrightarrow} H_2CO_3^* \qquad\qquad (1)$$

$$H_2CO_3^* \overset{K_1}{\leftrightarrow} H^+ + HCO_3^- \qquad\qquad (2)$$

$$HCO_3^- \overset{K_2}{\leftrightarrow} H^+ + CO_3^{2-} \qquad\qquad (3)$$

$$H_2O \overset{K_w}{\leftrightarrow} H^+ + OH^- \qquad\qquad (4)$$

$$H_3BO_3 + H_2O \overset{K_B}{\leftrightarrow} H^+ + B(OH)_4^- \qquad (5)$$

Where $K_0$, $K_1$, $K_2$, $K_w$, $K_B$ are the equilibrium constants for dissociations respectively, which are given as below:

$$K_0 = \frac{[H_2CO_3^*]}{pCO_2} \qquad\qquad (6)$$

$$K_1 = \frac{[H^+][HCO_3^-]}{[H_2CO_3^*]} \qquad\qquad (7)$$

$$K_2 = \frac{[H^+][CO_3^{2-}]}{HCO_3^-} \qquad\qquad (8)$$

$$K_w = [H^+][OH^-] \qquad\qquad (9)$$

$$K_B = \frac{[H^+][B(OH)_4^-]}{[H_3BO_3]} \qquad\qquad (10)$$

From the concentration definition,

$$DIC = [H_2CO_3^*] + [HCO_3^-] + [CO_3^{2-}] \qquad\qquad (11)$$

$$ALK = [HCO_3^-] + 2[CO_3^{2-}] + [OH^-] - [H^+] + [B(OH)_4^-] \qquad (12)$$

$$TB = [B(OH)_4^-] + [H_3BO_3] = c.S \qquad\qquad (13)$$

Neglecting the minor contribution of other weak bases to alkalinity (ALK; Dickson and Goyet, 1994) and rewriting the equation for ALK in terms of the known parameters DIC and TB and then by solving the resulting equation for unknown $[H^+]$, pCO$_2$ can be calculated as

$$pCO_2 = \frac{[DIC]}{K_0} \frac{[H^+]^2}{[H^+]^2 + K_1[H^+] + K_1K_2} \qquad (14)$$

Where $[H^+]$ is calculated using Newton-Raphson iterative method (Press et. al., 1996, Najjar and Orr, 1998), [DIC] is calculated using equation (11) and the value of equilibrium constants are derived from temperature and salinity dependent equation as given below.

The solubility of CO$_2$ (mol kg$^{-1}$ atm$^{-1}$):

$$\ln K_0 = -60.2409 + 93.4517 \left(\frac{100}{T}\right) + 23.3585 \ln\left(\frac{T}{100}\right) + S\left(0.023517 - 0.023656\left(\frac{T}{100}\right) + 0.0047036 \left(\frac{T}{100}\right)^2\right) \qquad (15) \qquad \text{(Weiss, 1974)}$$

The dissociation constant of carbonic acid (mol kg$^{-1}$):

$$-\log K_1 = -62.008 + \frac{3670.7}{T} + 9.7944 \ln(T) - 0.0118S + 0.000116S^2 \qquad (16)$$

$$-\log K_2 = 4.777 + \frac{1394.7}{T} - 0.0184S + 0.000118S^2 \qquad (17)$$

(Mehrbach et al., 1973, Dickson and Millero, 1987)

The dissociation constant of water $K_w$ (mol kg$^{-1}$)

$$-\ln K_w = 148.96502 + \frac{-13847.26}{T} - 23.6521 \ln(T) + S^{\frac{1}{2}}\left(-5.977 + \frac{118.67}{T} + 1.0495 \ln(T)\right) - 0.01615S \qquad (18)$$

(Millero, 1995)

The dissociation constant of boric acid $K_B$ [(mol kg$^{-1}$)$^2$]

$$-\ln K_B = \frac{1}{T} \left(-8966.9 - 2890.53\, S^{0.5} - 77.942\, S + 1.728\, S^{1.5} - 0.0996\, S^2\right) 148.0248 \\ + 137.1942\, S^{0.5} + 1.62142\, S + 0.053105\, S^{0.5}\, T \\ + \ln(T)\left(-24.4344 - 25.085\, S^{0.5} - 0.2474\, S\right) \qquad (19)$$

(Dickson, 1990)

Where T is the temperature in [K] and S is the salinity on the practical salinity scale.

The total boron equation ($\mu$mol kg$^{-1}$; Uppström, 1974):

$$TB = 1.185\, S \qquad (20)$$

Thus using equation (14) pCO$_2$ is calculated in the biogeochemical model (Najjar and Orr, 1998; Sarmiento and Gruber, 2006).

**Section B**

**Biogeochemical model**

[revised manuscript text omitted]

Takahashi, T., J. Olafsson, J. G. Goodard, D. W.. Chipman, and S.C. Sutherland: Seasonal variation of $CO_2$ and nutrients in the high latitudes surface oceans: A comparative study, Global Biogeochem. Cycles, 7, 843 – 878.

Takahashi, T., Sutherland, S. C., Wanninkhof, R., Sweeney, C., Feely, R. A., Chipman, D. W., Hales, B., Friederich, G., Chavez, F., Sabine, C., et al.: Climatological mean and decadal changes in surface ocean $pCO_2$ and net sea-air $CO_2$ flux over the global oceans. Deep Sea Res., Pt. II., 56, 554 – 557, doi:10.1016/j.dsr2.2008.12.009, 2009.

Uppström, L. (1974), The boron/chlorinity ratio of deep sea water from the Pacific Ocean, Deep-Sea Res., 21, 161-162.

Valsala, K. V., Maksyutov, S., Ikeda, M.: Design and Validation of an offline oceanic tracer transport model for a carbon cycle study, J. clim., 21, doi: 10.1175/2007JCLI2018.1, 2008.

Valsala, V., Maksyutov, S.: Simulation and assimilation of global **oc**ean $pCO_2$ and air-sea $CO_2$ fluxes using ship observations of surface ocean $pCO_2$ in a simplified biogeochemical model, Tellus., 62B, doi: 10.1111/j.1600-0889.2010.00495, 2010.

Weiss, R. F. (1974), Carbon dioxide in water and seawater: The solubility of a non-ideal gas, Mar. Chem., 2, 203-215.

Wiggert, J. D., Hood, R. R., Banse, K., Kindle, J. C.: Monsoon-driven biogeochemical processes in the Arabian Sea, Progr. Oceanogr., 65, 176-213, doi:10.1016/j.pocean.2005.03.008, 2005.

Yamanaka, Y., and E. Tajika: Role of dissolved organic matter in the marine biogeochemical cycle: studies using an ocean biogeochemical general circulation model. Global Biogeochem. Cycles, 11, 599 – 612, 1997.

[revised manuscript text omitted]

---

## Author Response (AR3)

**Replies to the Editor comments**

1. I have read the revised version of your work as well as the answers you provided to my comments. I find the manuscript much improved but I still have comments.
I have problems with the title, "Optimization of Biological Production for Indian Ocean upwelling zones: Part – I: Improving Biological Parameterization via a variable Compensation Depth" because the paper is not dealing with optimization of biological production (better estimation but not optimization). I would suggest something like "Biological production in the Indian Ocean upwelling zones: Part –I: refined estimation via the use of a variable compensation depth in ocean carbon models. " it would more reflect the content of your paper.

Thank you for your suggestion. We agree that the word 'optimization' may be bit confusing in this context, and we accept the title suggested by the Editor. As per your advice we are restating the title of our paper as "Biological production in the Indian Ocean upwelling zones: Part-I: refined estimation via the use of a variable compensation depth in ocean carbon models".

2. I also find that the paper is very long with notably 8 tables and 21 figures which may render difficult to capture the main messages. I would suggest that you consider to put some figures as supplements and that you leave in the main text the most striking ones.

Thank you for your suggestion. In fact it is true that the number of figures are quiet large. We have decided to move Tables 7, 8, Figures 16 to 21 and relevant text (of the sensitivity experiments from section 4) to supplementary material.

3. In your answer to my comments, you mentioned that biological production will be defined in the text (you mentioned lines 729-730) but it is not the case. So please define biological production since the beginning (Introduction). For instance (Lines 123-), when you defined Zc, it will be appropriate that you inform the reader on what will be the biological production in this manuscript.

As per the Editor's comment we have explicitly stated the above lines (i.e. 729-730) in the Introduction part immediately after defining Zc. The revised line numbers are 109 - 114.

4.  Lines 154-155 : Please reformulate.

Reformulated as "The improvements due to the effect of a variable Zc over the Indian Ocean and the sensitivity experiments where upwelling is muted strongly imply that the biological pump may play as much of a role as the solubility pump in determining surface $pCO_2$ and $CO_2$ fluxes over the Indian Ocean". Please refer lines 143 – 146 in the revised manuscript.

5.  Line 221-223: Please refer appropriately to appendix for details on the model equations.

Citation to the Appendix is added properly to Section 2.2.

6.  Line 226 : Jprod has to be defined.

$J_{prod}$ represents the biogeochemical flows with respect to production of organic phosphorous. Please refer lines 200 – 201 in the revised manuscript.

7.  Lines 226-229: I suppose that Jprod =(2) if PO4>PO4* and Z<Zc While Jprod=0 if PO4Zc? Please clarify as it is not clear if the two conditions are needed or not. (it is needed at line 227 while at line 229 it is one of the two if I am right)

Thank you for pointing this out. This was a typo and we appreciate your eye to the detail. This has been rectified as below.

$$J_{prod} = \frac{1}{\tau}([PO_4] - [PO_4]^*) \; ; \qquad\qquad [PO_4] > [PO_4]^* \; ; \quad Z < Zc \qquad\qquad \text{Eq(1)}$$

$$J_{prod} = 0 \; ; \qquad\quad [PO_4] \leq [PO_4]^* \qquad\qquad\qquad\qquad\qquad\qquad \text{Eq(2)}$$

Please refer equations 2 and 3 in the revised manuscript.

8. Line 283: suppresses

Corrected accordingly. Please refer lines 251 - 252 in the revised manuscript.

9. Lines 294-304: Zc, NPP,.. have already been defined before in the introduction (lines 107-123)

Corrected accordingly, Please refer lines 264 - 268 in the revised manuscript.

10. Line 320: Why 50 W/m2? Equations (2) suggest a difference for Z

Thank you for pointing this out. Corrected as 10 W m$^{-2}$. Please refer to lines 286 – 287 in the revised manuscript.

11. Line 745: "are" the biological source/sink terms..

Corrected accordingly. Please refer to lines 638 - 639 in the revised manuscript.

12. Line 746 " due to evaporation.." do you mean air-sea exchange?

No, We do not mean air-sea exchange. It accounts for dilution and concentration of water by precipitation and evaporation, and corresponding changes in DIC, ALK etc. They are different from the air-sea exchange term which is $J_g DIC$.

Please refer to lines 639 to 641 in the revised manuscript.

13. Lines 750-753: you use JDOP, JPO4, JDIC, JALK while above you use JbPO4,JbDOP,.. please homogenize the notation for clarity.

Corrected accordingly. Please refer to lines 637 - 639 in the revised manuscript.

14. Line 764: JDOP, ..already defined above

Corrected accordingly. Please refer to lines 653-654 in the revised manuscript.

15. Line 801 remove "is" because it is written twice.

Corrected accordingly. Please refer to line 690 in the revised manuscript.

[revised manuscript text omitted]

---

## Author Response (AR4)

Dear Editor,

Thank you for accepting our work. As per your suggestions, I have addressed all your comments.

A point by point reply to the comments is provided below.

Sincerely,

Sreeush M. G.

**Reply to the editor comments:**

1. Line 158 U and W are the velocity (please add W as in the equations below)

   Corrected accordingly. Please refer lines 158 and 172 of the revised manuscript.

2. Lines 188-189: I would say "Inorganic phosphorus"

   Corrected accordingly. Please see lines 188 -189 in the revised manuscript.

3. Line 203 : how Jprod can give a new-production in gC/m2/yr ? you need to have a conversion factor from phosphorus to carbon.

   Using the Redfield ratio $R_{C:P}$ = 117:1. The phosphorus is converted into carbon. Please refer lines 202 – 203 in the revised manuscript.

4. Line 283 : why did you mention ("see Appendix B") ? Figure 2 is not in the appendix

   Appendix B explains the method of calculating average relative photosynthesis shown in figure 2. Please refer the lines 282 – 283 of the revised manuscript.

5. Lines 330-331 : the order of presentation of case study is first SC and then SCTR.

   Corrected accordingly. Please refer lines 328 – 331 of the revised manuscript.

[revised manuscript text omitted]